# Solving Discrete (Semi) Unbalanced Optimal Transport with Equivalent Transformation Mechanism and KKT-Multiplier Regularization

**Weiming Liu**[1], **Xinting Liao**[2]*, **Jun Dan**[2], **Fan Wang**[2], **Hua Yu**[3],

**Junhao Dong**[3], **Shunjie Dong**[4], **Lianyong Qi**[5,6], **Yew-Soon Ong**[3,7]

[1] ByteDance Inc., [2] Zhejiang University,
[3] College of Computing and Data Science, Nanyang Technological University, Singapore,
[4] Shanghai Jiao Tong University, [5] China University of Petroleum (East China),
[6]Shandong Key Laboratory of Intelligent Oil & Gas Industrial Software,
[7] Centre for Frontier AI Research, Institute of High Performance Computing,
Agency for Science, Technology and Research, Singapore
lwming95@gmail.com, {xintingliao, danjun, fanwang97}@zju.edu.cn,
{junhao003, yu_hua, asysong}@ntu.edu.sg, sjdong@sjtu.edu.cn, lianyongqi@gmail.com

## Abstract

Semi-Unbalanced Optimal Transport (SemiUOT) shows great promise in matching two discrete probability measures by relaxing one of the marginal constraints. Previous SemiUOT solvers often incorporate an entropy regularization term, inevitably resulting in inaccurate matching solutions. To address this issue, we propose an Equivalent Transformation Mechanism (ETM) approach to determine the marginal probability distributions of SemiUOT with KL divergence. Furthermore, we validate the generalization capability of ETM by exploiting the marginal probability distributions of Unbalanced Optimal Transport (UOT). ETM is able to determine the exact marginal probabilities of both SemiUOT and UOT, based on which we can transform the SemiUOT/UOT into classic Optimal Transport (OT) problem. Moreover, we propose a KKT-Multiplier regularization term combined with Multiplier Regularized Optimal Transport (MROT) to achieve more accurate matching results. We conduct extensive experiments to demonstrate the effectiveness of our proposed methods in addressing SemiUOT and UOT problems[2].

## 1 Introduction

Optimal Transport (OT) technique is a powerful tool for matching and discerning two distinct probability distributions. Nowadays, OT has multiple successful applications in traditional machine learning [37, 35, 112, 22, 85, 62], unsupervised clustering [4, 15], domain adaptation [25, 23, 84, 60, 61], diffusion [49, 56], generative modeling [50, 79, 100, 44] and many others. Nevertheless, directly solving OT distances could have relatively high computation cost with around super-cubic time. Although one can adopt entropy-based Sinkhorn algorithm [24] for solving OT efficiently, it still suffers from the dilemma of dense and inaccurate solutions [58, 63, 31]. Moreover, classic OT strictly assumes that the probability masses on both source and target domains should be equal. It further hurdles the generalization of OT when the data samples inherit noise or outliers.

---

*Corresponding author.
[2]The demo code is provided: https://github.com/XeniaLLL/ETM.git

39th Conference on Neural Information Processing Systems (NeurIPS 2025).

Recently, Unbalanced Optimal Transport (UOT) [6, 19, 93, 92, 96, 59] and Semi-Unbalanced Optimal Transport (SemiUOT) [52] have become more attractive in adapting outliers since they allow relaxing marginal constraints for transportation results. This advantage makes SemiUOT and UOT powerfully applicable in transfer learning [101, 73, 80, 29, 26–28], computer vision [9, 30, 21, 74, 16, 67, 110], structure data exploration [90], natural language processing [3], and other areas. SemiUOT and UOT relax the strict OT mass equality constraints by introducing relaxation terms defined by Kullback-Leibler (KL) divergence [83], $\ell_1$ norm [10], or $\ell_2$ norm [8], whose effects are controlled by a coefficient $\tau$. Meanwhile, KL divergence is the most commonly-used in real practice [95]. Previous solvers always involves extra regularization terms, i.e., entropy regularization term and proximal point term [34], for tackling Semi and UOT problems. Meanwhile adding additional entropy terms will lead to dense and inaccurate matching solutions. Latest, [17] and [78] further reconsider solving UOT problem with majority maximization algorithm without the requirements of regularization terms. However, these methods are sensitive to the choice of $\tau$, i.e., providing sparse and accurate matching solutions when $\tau$ is small, but unsatisfying solutions when $\tau$ is large. Therefore, it is quite challenging to efficiently achieve accurate matching solutions for both SemiUOT and UOT problems.

In this paper, we propose a new method, i.e., ***Equivalent Transformation Mechanism*** (ETM), which directly finds the exact marginal probabilities of discrete SemiUOT and UOT with KL divergence. Specifically, ETM first finds the marginal probability distributions for SemiUOT and UOT problems based on Karush-Kuhn-Tucker (KKT) conditions and their dual forms. This induces a new insight for understanding SemiUOT and UOT problems, i.e., *We can transform SemiUOT and UOT problems into classic OT problems based on adjusting initial marginal weights via ETM*. We propose ETM-Refine to achieve exact marginal probabilities without needing overwhelming computation. Specifically, ETM-Refine first seeks the approximate results of marginal distributions via the fixed-point iteration with the smoothness function, and applies quasi-Newton based iterative methods to obtain exact results within quite a few steps. This competitively reduces the computation burden to obtain accurate matching results on SemiUOT/UOT. Beyond solving the marginal distribution, *we also discover that the KKT multipliers provide valuable guidance for addressing the OT problem, which is transformed from the SemiUOT or UOT problem with adjusted marginal weights.* Therefore we further propose ***Multiplier Regularized Optimal Transport*** (MROT) for achieving more sparse and accurate OT matching solutions. We summarize our contributions: (1) To our best knowledge, we first propose both exact and approximate solutions for ETM on two problems, i.e., SemiUOT and UOT. After optimizing these problems, one can obtain the sample marginal probabilities and transfer SemiUOT/UOT into standard optimal transport problems. (2) We first innovatively propose multiplier constraint terms to establish MROT for achieving more accurate results. (3) We conduct extensive experiments on both synthetic and real-world datasets to evaluate the performance of proposed ETM.

## 2 Preliminary

We first provide a brief introduction of SemiUOT and UOT. Let us consider two sets of data samples $\boldsymbol{X} \in \mathbb{R}^{M \times D}$ and $\boldsymbol{Z} \in \mathbb{R}^{N \times D}$ in source and target domains, where $M$, $N$ denote the number of samples and $D$ denotes the data dimension. Samples in each domain have corresponding prior-given mass weights, i.e., $\boldsymbol{a} \in \mathbb{R}^{M \times 1}$ for source domain and $\boldsymbol{b} \in \mathbb{R}^{N \times 1}$ for target domain. The semi-unbalanced optimal transport problem is set to measure the minimum cost among data samples $\boldsymbol{X}$ and $\boldsymbol{Z}$, meanwhile filtering out the noise and outliers by relaxing one of marginal constraints:

$$\min_{\pi_{ij} \geq 0} J_{\text{SemiUOT}} = \langle \boldsymbol{C}, \boldsymbol{\pi} \rangle + \tau \text{KL}\left(\boldsymbol{\pi} \mathbf{1}_N \| \boldsymbol{a}\right), \quad \text{s.t. } \boldsymbol{\pi}^\top \mathbf{1}_M = \boldsymbol{b}. \tag{1}$$

where $\boldsymbol{C} \in \mathbb{R}^{M \times N}$ denotes the pairwise distance matrix. Meanwhile $\boldsymbol{\pi} \in \mathbb{R}^{M \times N}$ denotes the coupling matching matrix among the data samples $\boldsymbol{X}$ and $\boldsymbol{Z}$. $\tau$ denotes the hyper parameter and KL $(\cdot)$ denotes the commonly-used KL divergence. SemiUOT relaxes the constraint $\boldsymbol{\pi} \mathbf{1}_N \neq \boldsymbol{a}$ and keep the constraint $\boldsymbol{\pi}^\top \mathbf{1}_M = \boldsymbol{b}$. Likewise, the unbalanced optimal transport problem [83] further relaxes both two marginal constraints, i.e., $\boldsymbol{\pi} \mathbf{1}_N \neq \boldsymbol{a}$ and $\boldsymbol{\pi}^\top \mathbf{1}_M \neq \boldsymbol{b}$ as shown:

$$\min_{\pi_{ij} \geq 0} J_{\text{UOT}} = \langle \boldsymbol{C}, \boldsymbol{\pi} \rangle + \tau_a \text{KL}\left(\boldsymbol{\pi} \mathbf{1}_N \| \boldsymbol{a}\right) + \tau_b \text{KL}(\boldsymbol{\pi}^\top \mathbf{1}_M \| \boldsymbol{b}), \tag{2}$$

where $\tau_a$ and $\tau_b$ denote the balanced hyper parameters. Previous researches always add an entropy regularization term to enhance the scalability of solving $\boldsymbol{\pi}^*$ of SemiUOT and UOT. However, it still suffers from the dense and inaccurate solution dilemma in real practice.

## 3 Methodology

In this section, we will first introduce *Equivalent Transformation Mechanism* (ETM), which investigates the problem of SemiUOT/UOT from the perspective of marginal probability distribution. Then we illustrate the effect of *Multiplier Regularized Optimal Transport* (MROT) that finds out the accurate solutions of $\pi^*$ for SemiUOT and UOT, with the guidance of *KKT-Multiplier Regularization*.

### 3.1 Equivalent Transformation Mechanism

ETM aims to seek the exact marginal distributions with satisfyingly efficient computation. Previous methods [83, 20] always directly adopted entropy-based regularization term into tackling SemiUOT and UOT problems. Although such approaches can provide fast computation speed, it will lead to relatively ambiguous and dense solutions that does not match most of the situations in real practice [55, 91]. In this section, we propose ETM-based methods to determine the marginal probabilities of source data samples in SemiUOT, accompanied by detailed illustrations. We then extend the ETM-based method to address the more complex UOT problem.

**ETM for SemiUOT.** To start with, we illustrate the ETM for transforming SemiUOT to classic OT, and introduce three types of ETM-based methods, i.e., ETM-Exact, ETM-Approx, and ETM-Refine, for determining the marginal probability distributions. Specifically, ETM-Exact directly computes the dual variables via iterative methods, e.g., LBFGS, achieving the exact results with an overwhelming computation burden. While ETM-Approx is a variant of ETM for SemiUOT by replacing the infimum with its smoothness approximation. Then we newly propose a fixed-point iteration method to solve the optimization problem efficiently. To further figure out the exact results, ETM-Refine applies ETM-Exact with quite a few iterations to solve the exact SemiUOT, by taking the approximate results as starting points. ETM-Refine shows competitive performance while maintaining efficient computation. By utilizing the methods above, one can transform SemiUOT into classic optimal transport problem by adjusting initial marginal weights. In the following, we will introduce the deduction and optimization details for the proposed ETM-based method on SemiUOT.

**Proposition 1.** (Principles of Equivalent Transformation Mechanism for SemiUOT) *Given SemiUOT with KL-Divergence $J_{\text{SemiUOT}}$, one can obtain its Fenchel-Lagrange multipliers form as:*

$$\min_{\boldsymbol{f},\boldsymbol{g},\zeta} \left[ \tau \sum_{i=1}^{M} a_i \exp\left(-\frac{f_i + \zeta}{\tau}\right) - \sum_{j=1}^{N} b_j(g_j - \zeta) \right] \qquad s.t. \ f_i + g_j + s_{ij} = C_{ij}, \quad s_{ij} \geq 0. \tag{3}$$

*where $\boldsymbol{f}$, $\boldsymbol{g}$, $\boldsymbol{s}$ and $\zeta$ denote Lagrange multipliers. Moreover, SemiUOT problem can be further transformed into the form of optimal transport with marginal constraints as follows:*

$$\min_{\boldsymbol{\pi} \geq 0} \mathcal{J}_{\text{P}} = \langle \boldsymbol{C}, \boldsymbol{\pi} \rangle, \qquad s.t. \ \boldsymbol{\pi} \boldsymbol{1}_N = \boldsymbol{a} \odot \exp\left(-\frac{\boldsymbol{f}^* + \zeta^*}{\tau}\right) = \boldsymbol{\alpha}, \quad \boldsymbol{\pi}^\top \boldsymbol{1}_M = \boldsymbol{b}. \tag{4}$$

*When $\tau \to \infty$, the source marginal probability is given as $\boldsymbol{\pi} \boldsymbol{1}_N = \omega_{\text{L}} \boldsymbol{a}$ and $\omega_{\text{L}} = \langle \boldsymbol{b}, \boldsymbol{1}_N \rangle / \langle \boldsymbol{a}, \boldsymbol{1}_M \rangle$.*

The proof of Proposition 1 can be found in Appendix B. We can observe that transforming SemiUOT into classic OT is to elementally adjust the initial weights of data samples by $\exp(-(\boldsymbol{f}^* + \zeta^*/\tau))$. To further simplify the calculation by reducing variable $\boldsymbol{g}$, we set $g_j = \inf_{k \in [M]} (C_{kj} - f_k)$ according to the $c$-transform theorem [103]. Hence we only need to optimize $\boldsymbol{f}$ and $\zeta$ without additional constraints:

$$\min_{\boldsymbol{f},\zeta} L_{\text{P}} = \tau \sum_{i=1}^{M} a_i \exp\left(-\frac{f_i + \zeta}{\tau}\right) - \sum_{j=1}^{N} \left[ \inf_{k \in [M]} [C_{kj} - f_k] - \zeta \right] b_j, \tag{5}$$

We refer to $L_{\text{P}}$ as the newly proposed *Exact SemiUOT Equation*. To solve this exact SemiUOT in Eq.(5), we initialize $\zeta = 0$ for the optimization and introduce an iterative method derived from LBFGS [109]. Specifically, we first fix $\zeta$ then adopting LBFGS method to reach optimal results of $\boldsymbol{f}^\ell$ and $g_j^\ell = \inf_{k \in [M]} (C_{kj} - f_k^\ell)$ at the $\ell$-th iteration. Then we optimize $\zeta = \tau[\log(\sum_{i=1}^{M} a_i \exp(-f_i^\ell/\tau)) - \log(\sum_{j=1}^{N} b_j)]$ which is obtained by considering $\nabla_\zeta L_{\text{P}} = 0$ and it guarantees $\sum_{i=1}^{M} a_i e^{-(f_i^\ell + \zeta)/\tau} = \sum_{j=1}^{N} b_j$. We iteratively update $L_{\text{U}}$ to reach the optimal solutions on $\zeta^*$, $\boldsymbol{f}^*$ and $\boldsymbol{g}^*$. We refer to the entire optimization scheme as ETM-Exact approach for Eq.(5).

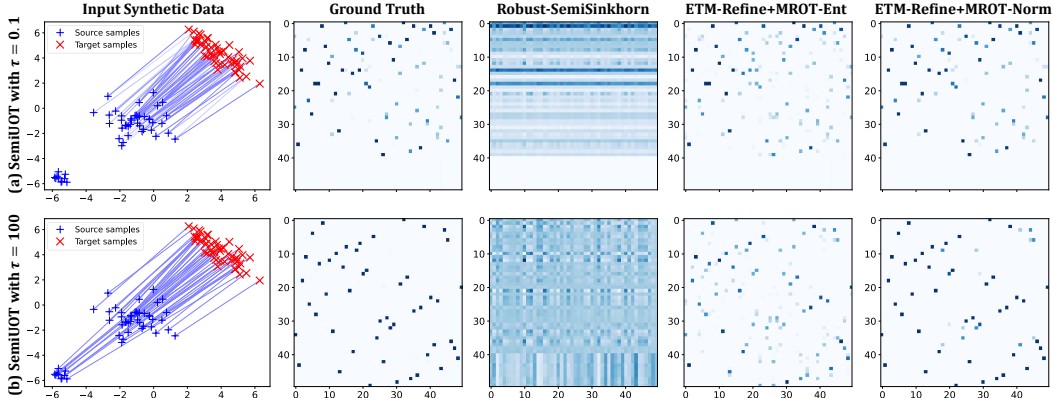

Figure 1: The SemiUOT matching solutions on $\boldsymbol{\pi}^*$ when $\tau = 0.1$ or $\tau = 100$ among the Robust-SemiSinkhorn [52] and our proposed ETM-Refine + MROT-Ent, ETM-Refine + MROT-Norm with $\eta_G = 10^2$ and $\epsilon = 10^{-2}$. We set $\eta_{\text{Reg}} = 0.1$ for entropy or $L_2$-norm regularization term.

Although $L_{\text{P}}$ is convex and has unique solutions, the presence of $\inf(\cdot)$ renders it a non-smooth function, preventing it from efficient optimization [1]. To further accelerate the optimization process, we consider making a smooth approximation on replacing $\inf(\cdot)$ with $\text{LogSumExp}(\cdot)$ as $\inf_{k \in [M]}[C_{kj} - f_k] \approx -\epsilon \log[\sum_{k=1}^M e^{\frac{f_k - C_{kj}}{\epsilon}}]$ since $|\epsilon \log[\sum_{k=1}^M e^{\frac{f_k - C_{kj}}{\epsilon}}] - \sup_{k \in [M]}[f_k - C_{kj}]| \leq \epsilon \log M$ following [75]. Here $\epsilon > 0$ denotes the balanced hyperparameter between accuracy and function smoothness. Smaller $\epsilon$ (e.g., $\epsilon \to 0$) could lead to more accurate but less smooth solutions. Then we can obtain the proposed *Approximate SemiUOT Equation* as $\widehat{L}_{\text{P}}$ by replacing $\inf(\cdot)$ with the smoothness term for $\widehat{f}$:

$$\min_{\widehat{\boldsymbol{f}}, \zeta} \widehat{L}_{\text{P}} = \tau \sum_{i=1}^M a_i \exp\left(-\frac{\widehat{f}_i + \zeta}{\tau}\right) + \sum_{j=1}^N b_j \left[\epsilon \log\left[\sum_{k=1}^M \exp\left(\frac{\widehat{f}_k - C_{kj}}{\epsilon}\right)\right] + \zeta\right]. \quad (6)$$

**Proposition 2.** (Calculation for Approximate SemiUOT Equation) *Given Approximate SemiUOT equation $\widehat{L}_{\text{P}}$, it can be optimized via Equivalent Transformation Mechanism with Approximation (ETM-Approx). That is, ETM-Approx aims to solve the following equation for each $\widehat{f}_s$:*

$$\frac{\partial \widehat{L}_{\text{P}}}{\partial \widehat{f}_s} = -a_s \exp\left(-\frac{\widehat{f}_s + \zeta}{\tau}\right) + \exp\left(\frac{\widehat{f}_s}{\epsilon}\right) \sum_{j=1}^N \left[\frac{b_j \exp\left(-\frac{C_{sj}}{\epsilon}\right)}{\sum_{k=1}^M \exp\left(\frac{\widehat{f}_k - C_{kj}}{\epsilon}\right)}\right] = 0. \quad (7)$$

*Specifically, we can adopt fixed-point iteration method for solving Eq.(7) at the $\ell$-th iteration:*

$$\widehat{f}_s^{\ell+1} = \nu \left[\log\left(a_s \exp\left(-\frac{\zeta}{\tau}\right)\right) - \log\left[\sum_{j=1}^N \left(\frac{b_j}{\mathscr{W}_{\epsilon,j}(\widehat{\boldsymbol{f}}^\ell)} \exp\left(-\frac{C_{sj}}{\epsilon}\right)\right)\right]\right], \quad \forall s \in [1, M], \quad (8)$$

*where $\nu = \tau\epsilon/(\tau + \epsilon)$ for simplification and $\mathscr{W}_{\epsilon,j}(\widehat{\boldsymbol{f}}^\ell)$ denotes the corresponding calculation as shown $\mathscr{W}_{\epsilon,j}(\widehat{\boldsymbol{f}}^\ell) = \sum_{k=1}^M \exp((\widehat{f}_k^\ell - C_{kj})/\epsilon)$. The proposed procedure can converge with a theoretical guarantee. Finally, updating variable $\zeta$ by further considering $\nabla_\zeta \widehat{L}_{\text{P}} = 0$ via $\zeta = \tau[\log(\sum_{i=1}^M a_i \exp(-\widehat{f}_i^*/\tau)) - \log(\sum_{j=1}^N b_j)]$, finally achieving optimal results $\widehat{\boldsymbol{f}}^*$ and $\zeta^*$.*

The proof of Proposition 2 can be found in Appendix C. Generally, Proposition 2 outlines the optimization procedure using the newly proposed ETM-Approx approach for addressing the Approximate Semi-UOT Equation. We can observe that the ETM-Approx approach is easy to compute and implement, while avoiding complex calculations (e.g., searching the descent direction and finding the step size) and not requiring a large amount of storage space against previous methods. Therefore, the ETM-Approx approach is an efficient method for determining the result of $\widehat{f}^*$ and $\widehat{g}_j^* = -\epsilon \log[\sum_{k=1}^M \exp((\widehat{f}_k^* - C_{kj})/\epsilon)]$, transforming SemiUOT into the optimal transport problem.

**Remark 1.** *ETM-Approx can reach the linear convergence rate via the fixed-point iteration shown as* $\mathcal{O}(NM\log(1/\varepsilon_{\mathrm{err}}))$ *where* $\varepsilon_{\mathrm{err}} = ||\widehat{\boldsymbol{f}} - \widehat{\boldsymbol{f}}^*||_\infty$ *and* $\widehat{\boldsymbol{f}}^*$ *denotes the optimal solution.*

Moreover, we can finally figure out the exact optimal solution $\boldsymbol{f}^*$ via the approximate optimal solution $\widehat{\boldsymbol{f}}^*$ on $\widehat{L}_{\mathrm{P}}$ using Proposition 2. That is, if we directly optimize $L_{\mathrm{P}}$ from a randomly initial point, we could spend more time on quasi-Newton gradient descent (e.g., LBFGS) for reaching $\boldsymbol{f}^*$. Since $\widehat{\boldsymbol{f}}^*$ is close to $\boldsymbol{f}^*$, it should be more efficient to use $\widehat{\boldsymbol{f}}^*$ as the initial guess for optimizing $\boldsymbol{f}^*$ via ETM-Exact which has super-linear convergence rate [46, 47, 86, 109, 39]. And we regard the whole procedure as ETM-Refine method with time complexity of $\mathcal{O}(NM\log(1/\varepsilon_{\mathrm{err}}) + NM(\log M)d_T)$ where $d_T$ denotes the number of iterations. ETM-Refine utilizes the strength of ETM-Approx in efficient computation for the exact results. In summary, we can utilize ETM-based methods to transform SemiUOT into classic OT problem. We illustrate the optimization details in Alg.1 and Appendix D.

---

**Algorithm 1** The algorithm of ETM-Based method on SemiUOT

---

    **Input:** $\boldsymbol{C}$: cost matrix; $\boldsymbol{a}, \boldsymbol{b}$: initial marginal probability; $\tau, \epsilon$: Hyper parameters.
      Randomly initialize the value of $\boldsymbol{f}^{\mathrm{init}}$.
      Choose ETM-Exact, ETM-Approx or ETM-Refine on SemiUOT for optimization.
    **(1) Function:** ETM-Exact on SemiUOT($\boldsymbol{C}, \boldsymbol{a}, \boldsymbol{b}, \tau, \boldsymbol{f}^{t=0} = \boldsymbol{f}^{\mathrm{init}}$)
      Optimize $\boldsymbol{f}$ via L-BFGS algorithm on $L_{\mathrm{P}}$.
      Optimize $\boldsymbol{g}$ via $g_j = \inf_{k \in [M]}(C_{kj} - f_k^t)$.
      Optimize $\zeta$ via $\zeta = \tau[\log(\sum_{i=1}^M a_i \exp(-f_i/\tau)) - \log(\sum_{j=1}^N b_j)]$.
    **Return**: The optimal solutions of $\boldsymbol{f}^*, \boldsymbol{g}^*$ and $\zeta^*$.
    **(2) Function:** ETM-Approx on SemiUOT($\boldsymbol{C}, \boldsymbol{a}, \boldsymbol{b}, \tau, \widehat{\boldsymbol{f}}^{t=0} = \boldsymbol{f}^{\mathrm{init}}$)
      Optimize $\widehat{\boldsymbol{f}}$ via Proposition 2 on $\widehat{L}_{\mathrm{P}}$.
      Optimize $\widehat{\boldsymbol{g}}$ via $\widehat{g}_j = -\epsilon\log[\sum_{k=1}^M \exp((\widehat{f}_k - C_{kj})/\epsilon)]$.
      Optimize $\zeta$ via $\zeta = \tau[\log(\sum_{i=1}^M a_i \exp(-\widehat{f}_i/\tau)) - \log(\sum_{j=1}^N b_j)]$.
    **Return**: The optimal solutions of $\widehat{\boldsymbol{f}}^*, \widehat{\boldsymbol{g}}^*$ and $\zeta^*$.
    **(3) Function:** ETM-Refine on SemiUOT($\boldsymbol{C}, \boldsymbol{a}, \boldsymbol{b}, \tau, \widehat{\boldsymbol{f}}^{t=0} = \boldsymbol{f}^{\mathrm{init}}$)
      Obtain $\widehat{\boldsymbol{f}}^* = $ ETM-Approx on SemiUOT($\boldsymbol{C}, \boldsymbol{a}, \boldsymbol{b}, \tau, \widehat{\boldsymbol{f}}^{t=0} = \boldsymbol{f}^{\mathrm{init}}$).
      Obtain $\boldsymbol{f}^* = $ ETM-Exact on SemiUOT($\boldsymbol{C}, \boldsymbol{a}, \boldsymbol{b}, \tau, \boldsymbol{f}^{t=0} = \widehat{\boldsymbol{f}}^*$).
    **Return**: The optimal solutions of $\boldsymbol{f}^*, \boldsymbol{g}^*$ and $\zeta^*$.

---

**ETM for UOT.** We have obtained the marginal probability of SemiUOT via tackling Proposition 1 with proposed ETM-based method. In this section, we will further extend ETM for solving the marginal probability on UOT, which is also a commonly existing optimization problem.

**Proposition 3.** (Principles of Equivalent Transformation Mechanism for UOT) *Given UOT with KL-Divergence* $J_{\mathrm{UOT}}$*, its Fenchel-Lagrange multipliers form is given:*

$$\min_{\boldsymbol{u},\boldsymbol{v},\zeta}\left[\tau_a\sum_{i=1}^M a_i\exp\left(-\frac{u_i+\zeta}{\tau_a}\right) + \tau_b\sum_{j=1}^N b_j\exp\left(-\frac{v_j-\zeta}{\tau_b}\right)\right], \quad s.t. \begin{cases} u_i + v_j + s_{ij} = C_{ij}, \\ s_{ij} \geq 0, \end{cases} \tag{9}$$

*where* $\boldsymbol{u}, \boldsymbol{v}, \boldsymbol{s}$ *and* $\zeta$ *denote Lagrange multipliers. Moreover, UOT problem can also be transformed into classic optimal transport as follows:*

$$\min_{\boldsymbol{\pi}\geq 0}\mathcal{J}_{\mathrm{U}} = \langle\boldsymbol{C},\boldsymbol{\pi}\rangle, \quad s.t. \ \boldsymbol{\pi}\mathbf{1}_N = \boldsymbol{a}\odot\exp\left(-\frac{\boldsymbol{u}^*+\zeta^*}{\tau_a}\right) = \boldsymbol{\alpha}, \ \boldsymbol{\pi}^\top\mathbf{1}_M = \boldsymbol{b}\odot\exp\left(-\frac{\boldsymbol{v}^*-\zeta^*}{\tau_b}\right) = \boldsymbol{\beta}. \tag{10}$$

*Note that when* $\tau_a, \tau_b \to \infty$*, the source and target marginal probabilities can be determined as* $\boldsymbol{\pi}\mathbf{1}_N = \sqrt{\omega_{\mathrm{L}}}\boldsymbol{a}$ *and* $\boldsymbol{\pi}^\top\mathbf{1}_M = \boldsymbol{b}/\sqrt{\omega_{\mathrm{L}}}$ *where* $\omega_{\mathrm{L}} = \langle\boldsymbol{b},\mathbf{1}_N\rangle/\langle\boldsymbol{a},\mathbf{1}_M\rangle$ *respectively.*

The proof of Proposition 3 can be found in Appendix E. Likewise, we set $v_j = \inf_{k \in [M]}(C_{kj} - u_k)$ by the $c$-transform theorem [103] to simplify the calculation. Hence we obtain *Exact UOT Equation*:

$$\min_{\boldsymbol{u},\zeta} L_{\mathrm{U}} = \tau_a\sum_{i=1}^M a_i\exp\left(-\frac{u_i+\zeta}{\tau_a}\right) + \tau_b\exp\left(\frac{\zeta}{\tau_b}\right)\sum_{j=1}^N b_j\exp\left(\frac{\sup_{k \in [M]}(u_k - C_{kj})}{\tau_b}\right). \tag{11}$$

We first fix $\zeta$ then adopting LBFGS approach to optimize $L_{\mathrm{U}}$ on $\boldsymbol{u}$. Then we futher optimize $\zeta = \kappa[\log(\sum_{i=1}^M a_i\exp(-u_i^\ell/\tau_a)) - \log(\sum_{j=1}^N b_j\exp(-v_j^\ell/\tau_b))]$ at the $\ell$-th iteration where $v_j^\ell = $

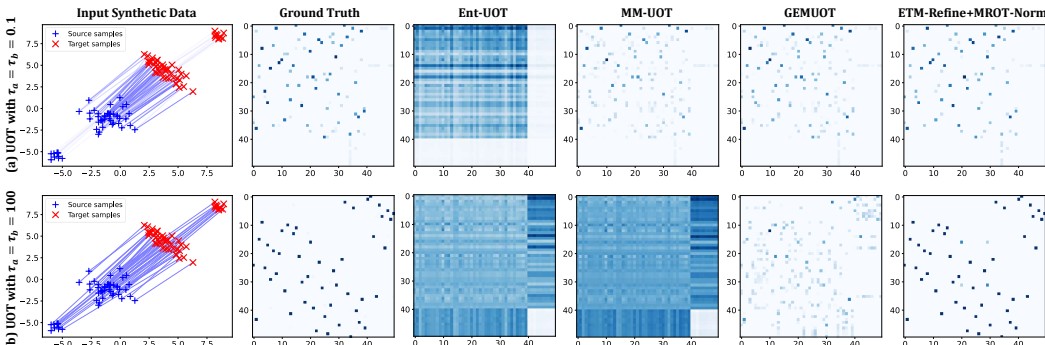

Figure 2: Results of $\boldsymbol{\pi}^*$ on UOT when $\tau_a = \tau_b = 0.1$ or $\tau_a = \tau_b = 100$ among Ent-UOT [83], MM-UOT [17], GEMUOT [78] and ETM-Refine+MROT-Norm with $\eta_G = 10^2$ and $\eta_{\text{Reg}} = 0.1$.

$\inf_{k \in [M]}(C_{kj} - u_k^\ell)$ and $\kappa = \tau_a \tau_b / (\tau_a + \tau_b)$ by considering $\nabla_\zeta L_U = 0$. Here we regard the above process as the ETM-Exact approach for solving UOT problem. Note that the non-smooth function $\sup(\cdot)$ will result in inefficient optimization. However, if we directly apply a similar function approximation to replace $\sup(\cdot)$ following Eq.(6), the optimization problem becomes quite complex, making it relatively difficult to determine the iterative solutions. Meanwhile, Proposition 2 enlightens us with a completely new ETM-Approx approach for optimizing UOT.

**Optimization 1.** (Calculation of ETM-Approx approach for UOT) Since the optimization problem in Eq.(9) is convex, we can also utilize block gradient descent to optimize the problem. Specifically, we first fix $\widehat{v}^l$ and optimize variable $\widehat{u}^l$ at the $l$-th iteration by replacing the original marginal probability $\boldsymbol{b}$ in Eq.(6) with $\boldsymbol{b} \odot \exp(-(\widehat{\boldsymbol{v}} - \zeta)/\tau_b) = \boldsymbol{\beta}$ accordingly to transform UOT into SemiUOT problem:

$$\min_{\widehat{\boldsymbol{u}}} \widehat{L}_U^u = \tau_a \sum_{i=1}^M a_i \exp\left(-\frac{\widehat{u}_i + \zeta}{\tau_a}\right) + \sum_{j=1}^N \beta_j \left[\epsilon \log\left[\sum_{k=1}^M \exp\left(\frac{\widehat{u}_k - C_{kj}}{\epsilon}\right)\right] + \zeta\right]. \qquad (12)$$

It is equivalent to solve the equation by taking the differentiation w.r.t. on $\widehat{u}_s$ over $\widehat{L}_U^u$ and set it 0:

$$\frac{\partial \widehat{L}_U^u}{\partial \widehat{u}_s} = -a_s \exp\left(-\frac{\widehat{u}_s + \zeta}{\tau_a}\right) + \exp\left(\frac{\widehat{u}_s}{\epsilon}\right) \sum_{j=1}^N \left[\frac{\beta_j \exp\left(-\frac{C_{sj}}{\epsilon}\right)}{\sum_{k=1}^M \exp\left(\frac{\widehat{u}_k - C_{kj}}{\epsilon}\right)}\right] = 0. \qquad (13)$$

Obviously, it is equivalent to replace $\boldsymbol{b}$ with $\boldsymbol{\beta}$ in Eq.(7) for solving Eq.(13). Then we can utilize the iteration step shown in Eq.(8) to obtain $\widehat{\boldsymbol{u}}^{l+1}$. After that we fix $\widehat{\boldsymbol{u}}^{l+1}$ and optimize variable $\widehat{\boldsymbol{v}}^{l+1}$ via $\widehat{v}_j^{l+1} = -\epsilon \log[\sum_{k=1}^M \exp((\widehat{u}_k^{l+1} - C_{kj})/\epsilon)]$. We can achieve the optimal solution on $\widehat{\boldsymbol{u}}^*$ and $\widehat{\boldsymbol{v}}^*$ via iteratively computing via the above procedure accordingly. Finally, we update variable $\zeta$ via considering $\zeta = (\tau_a \tau_b / (\tau_a + \tau_b))[\log(\sum_{i=1}^M a_i \exp(-\widehat{u}_i^*/\tau_a)) - \log(\sum_{j=1}^N b_j \exp(-\widehat{v}_j^*/\tau_b))]$. Due to the space limits, the deduction details are provided in Appendix F. Moreover, it has the time complexity of $\mathcal{O}(NM \log(1/\varepsilon_{\text{err}}))$ where $\varepsilon_{\text{err}} = ||\widehat{\boldsymbol{u}} - \widehat{\boldsymbol{u}}^*||_\infty$ with the linear convergence rate.

In summary, Optimization 1 for solving the UOT can be seen as an extension of Proposition 2 applied to SemiUOT, demonstrating the robust generalization capability of the proposed ETM method. Likewise, one can utilize $\widehat{\boldsymbol{u}}^*$ and $\widehat{\boldsymbol{v}}^*$ as the initial guess for solving Exact UOT Equation on Eq.(11) via ETM-Refine. Hence, ETM-based methods (i.e., ETM-Exact, ETM-Approx and ETM-Refine) transform UOT into classic optimal transport via computing the exact marginal distributions.

### 3.2 Multiplier Regularized Optimal Transport Induced by KKT-Multiplier Regularization

According to the Proposition 1-3 that discussed in Section 3.1, we have figured out the marginal probability distributions on both SemiUOT and UOT with commonly used KL Divergence via proposed ETM-based methods. Motivated by this, we can observe that the core mechanism of UOT/SemiUOT is carefully reweighting the weights of different samples accordingly. If the samples are noise or outliers, the corresponding weights will be much smaller than the corresponding weights among similar data samples. Therefore, UOT/SemiUOT has better adaptability than traditional OT that commonly treats all data samples equally. In this section, we will further exploit the matching results of $\boldsymbol{\pi}$ for SemiUOT and UOT using the following corollary:

**Corollary 1.** *Given any UOT/SemiUOT with KL divergence, we can transfer the original problem into classic optimal transport via adopting proposed ETM approach flexibly. We can further utilize existing OT solver for solving $\boldsymbol{\pi}^*$ as* $(\mathrm{UOT}, \mathrm{SemiUOT}) \xrightarrow{\text{ETM Method}} \mathrm{OT} \xrightarrow{\text{OT Solver}} \boldsymbol{\pi}^*$.

This observation provides us with entirely new unified insight into solving the matching results of $\boldsymbol{\pi}^*$ for SemiUOT and UOT. It is essential to utilize the proposed ETM-based method, as it offers a variety of OT solvers that yield more efficient and accurate results than directly optimizing UOT or SemiUOT. In general, one can further adopt Sinkhorn [24, 14], $\ell_2$-norm term [8] or some other sparsification OT solver [58, 41] with different regularization terms to achieve the transportation $\boldsymbol{\pi}$.

Although some OT solvers (e.g., Sinkhorn [24]) could figure out $\boldsymbol{\pi}$ efficiently comparing to the linear programming with cubic time complexity [48], they often provide ambiguous results that may deviate significantly from the correct solutions [72, 58]. Hence it remains a challenge to efficiently find an accurate result for $\boldsymbol{\pi}^*$. Recalling the whole process of ETM method, we not only obtain the marginal probabilities, but also derive multipliers $\boldsymbol{s}$ which can be further utilized as guidance.

**Corollary 2.** *Given the optimal $\boldsymbol{u}^*$ and $\boldsymbol{v}^*$ in UOT via ETM-based method, one can obtain $\boldsymbol{s}$ on UOT by $s_{ij} = \max(0, C_{ij} - u_i^* - v_j^*)$. Likewise, the multipliers $\boldsymbol{s}$ on SemiUOT can be obtained via ETM-based method as $s_{ij} = \max(0, C_{ij} - f_i^* - g_j^*)$. Multipliers $\boldsymbol{s}$ indicate the value of $\boldsymbol{\pi}$, i.e., (case 1) $s_{ij} > 0$ when $\pi_{ij} = 0$ and (case 2) $s_{ij} = 0$ when $\pi_{ij} > 0$ according to the KKT conditions.*

The Corollary 2 demonstrates that the value of $\pi_{ij}$ can be reflected via $s_{ij}$. This observation inspires us to further utilize such useful information in accurately calculating matching results $\boldsymbol{\pi}^*$.

**Proposition 4.** (The Definition and Usage of KKT-Multiplier Regularization) *Given any OT with multiplier $\boldsymbol{s}$, one can obtain accurate solution $\boldsymbol{\pi}^*$ via proposed KKT-multiplier regularization term $\mathcal{G}(\boldsymbol{\pi}, \boldsymbol{s}) = \langle \boldsymbol{\pi}, \boldsymbol{s} \rangle$, which formulates Multiplier Regularized Optimal Transport (MROT):*

$$\min_{\boldsymbol{\pi} \geq 0} \mathcal{J}_{\mathrm{G}} = \langle \boldsymbol{C}, \boldsymbol{\pi} \rangle + \eta_G \langle \boldsymbol{\pi}, \boldsymbol{s} \rangle + \eta_{\mathrm{Reg}} \mathcal{L}_{\mathrm{Reg}}(\boldsymbol{\pi}), \quad s.t. \ \boldsymbol{\pi} \mathbf{1}_N = \boldsymbol{\alpha}, \quad \boldsymbol{\pi}^\top \mathbf{1}_M = \boldsymbol{\beta}, \quad (14)$$

*where $\mathcal{L}_{\mathrm{Reg}}(\boldsymbol{\pi})$ denotes the regularization term on $\boldsymbol{\pi}$. $\boldsymbol{\alpha}$, $\boldsymbol{\beta}$ denote the final marginal probabilities obtained by ETM-based method, while $\eta_{\mathrm{Reg}}$ and $\eta_G$ denote the hyperparameters. Ideally, $\eta_G$ should be set as a relatively large number. Meanwhile the dual form of MROT is given as:*

$$\max_{\boldsymbol{\psi}, \boldsymbol{\phi}} L_{\mathrm{G}} = \langle \boldsymbol{\alpha}, \boldsymbol{\psi} \rangle + \langle \boldsymbol{\beta}, \boldsymbol{\phi} \rangle - \eta_{\mathrm{Reg}} \mathcal{L}_{\mathrm{Reg}}^*((\psi_i + \phi_j - \widetilde{C}_{ij})/\eta_{\mathrm{Reg}}), \quad (15)$$

*where $\widetilde{C}_{ij} = C_{ij} + \eta_G s_{ij}$, $\boldsymbol{\phi}$ and $\boldsymbol{\psi}$ denote the Lagrange multipliers for MROT. $\mathcal{L}_{\mathrm{Reg}}^*(\cdot)$ denotes the conjugate function of $\mathcal{L}_{\mathrm{Reg}}(\cdot)$ and one can figure out the matching results of $\boldsymbol{\pi}$ via solving the following equation $\nabla_{\pi_{ij}} \mathcal{L}_{\mathrm{Reg}}(\pi_{ij}) = (\psi_i + \phi_j - \widetilde{C}_{ij})/\eta_{\mathrm{Reg}}$.*

We provide the deduction of MROT in Appendix H. That is, minimizing $s_{ij}\pi_{ij}$ to 0 could result in $s_{ij}\pi_{ij} = 0$, which not only aligns with the KKT complementary condition, but also reweights the matching for more accurate results. Generally, MROT is orthogonal to adopting different kinds of regularization term $\mathcal{L}_{\mathrm{Reg}}(\cdot)$ for efficient optimization. For instance, one can use the widely adopted entropy regularization term $\mathcal{L}_{\mathrm{Reg}}(\boldsymbol{\pi}) = -\langle \boldsymbol{\pi}, \log(\boldsymbol{\pi}) - 1 \rangle$ to formulate Entropic Multiplier Regularized Optimal Transport (MROT-Ent), whose matching results satisfy $\pi_{ij} = \exp(-\eta_G s_{ij}/\eta_{\mathrm{Reg}}) \exp((\psi_i + \phi_j - C_{ij})/\eta_{\mathrm{Reg}})$. Obviously, involving the multipliers information $\boldsymbol{s}$ has achieved more accurate solutions. Specifically, the non-matching samples pairs will get lower value on $\pi_{ij}$ since $\mathcal{G}(\boldsymbol{\pi}, \boldsymbol{s}) = \langle \boldsymbol{\pi}, \boldsymbol{s} \rangle$ avoids rigorous results. Otherwise, the matching results on $\pi_{ij}$ will mainly be determined by the transportation cost. Similarly, one can also adopt $L_2$-norm regularization term $\mathcal{L}_{\mathrm{Reg}}(\boldsymbol{\pi}) = \frac{1}{2}\langle \boldsymbol{\pi}, \boldsymbol{\pi} \rangle$ to formulate Sparse Multiplier Regularized Optimal Transport (MROT-Norm) with similar characteristics. The time complexity of MROT depends on the regularization term $\mathcal{L}_{\mathrm{Reg}}(\boldsymbol{\pi})$, that is, MROT-Ent and MROT-Norm have a complexity of $\mathcal{O}(NMd_\pi)$ where $d_\pi$ is the number of iterations. In conclusion, we can integrate the ETM-based methods with MROT method to solve the SemiUOT and UOT problems, achieving accurate results for both marginal probabilities and the matching solution $\pi_{ij}$.

## 4 Experiments

### 4.1 Experimental setup

**Datasets.** We conduct experiments on both synthetic and real-world datasets to evaluate the methods. **(1) Synthetic Datasets.** We first conduct the experiments on the synthetic datasets. That is, we set

Table 1: Classification accuracy (%) on *Office-Home* for UDA and Partial UDA

| Method for UDA | Ar→Cl | Ar→Pr | Ar→Rw | Cl→Ar | Cl→Pr | Cl→Rw | Pr→Ar | Pr→Cl | Pr→Rw | Rw→Ar | Rw→Cl | Rw→Pr | Avg |
|---|---|---|---|---|---|---|---|---|---|---|---|---|---|
| ResNet [43] | 34.9 | 50.0 | 58.0 | 37.4 | 41.9 | 46.2 | 38.5 | 31.2 | 60.4 | 53.9 | 41.2 | 59.9 | 46.1 |
| DeepJDOT [25] | 50.7 | 68.6 | 74.4 | 59.9 | 65.8 | 68.1 | 55.2 | 46.3 | 73.8 | 66.0 | 54.9 | 78.3 | 63.5 |
| ROT [5] | 47.2 | 71.8 | 76.4 | 58.6 | 68.1 | 70.2 | 56.5 | 45.0 | 75.8 | 69.4 | 52.1 | 80.6 | 64.3 |
| JUMBOT [34] | 55.2 | 75.5 | 80.8 | 65.5 | 74.4 | 74.9 | 65.2 | 52.7 | 79.2 | 73.0 | 59.9 | 83.4 | 70.0 |
| JUMBOT + UOT(MM-UOT) | 56.3 | 76.2 | 81.6 | 66.0 | 75.3 | 75.1 | 66.4 | 52.9 | 79.2 | 73.8 | 60.7 | 84.1 | 70.6 |
| JUMBOT + UOT(GEMUOT) | 57.5 | 77.4 | 82.7 | 67.2 | 76.0 | 75.6 | 66.1 | 54.5 | 80.5 | 74.9 | 61.8 | 85.2 | 71.6 |
| JUMBOT + UOT($\ell_2$-Norm Solver) | 57.0 | 76.7 | 81.8 | 66.1 | 74.5 | 75.5 | 65.9 | 53.4 | 79.6 | 74.2 | 60.6 | 83.3 | 70.7 |
| JUMBOT + UOT(Sparse Solver) | 57.8 | 77.1 | 82.3 | 66.7 | 76.2 | 75.6 | 67.0 | 54.1 | 80.7 | 75.4 | 61.3 | 84.6 | 71.5 |
| JUMBOT + UOT(ETM-Refine + MROT-Ent) | 59.0 | 78.5 | 83.4 | **68.7** | 77.1 | 77.6 | 68.3 | 57.2 | 82.4 | 76.2 | **62.5** | 86.4 | 73.1 |
| JUMBOT + UOT(ETM-Refine + MROT-Norm) | **59.4** | **78.7** | **84.1** | 68.5 | **77.3** | **78.5** | **68.6** | **57.9** | **82.8** | 76.3 | **62.5** | **86.5** | **73.4** |

| Method for Partial UDA | Ar→Cl | Ar→Pr | Ar→Rw | Cl→Ar | Cl→Pr | Cl→Rw | Pr→Ar | Pr→Cl | Pr→Rw | Rw→Ar | Rw→Cl | Rw→Pr | Avg |
|---|---|---|---|---|---|---|---|---|---|---|---|---|---|
| ResNet [43] | 46.3 | 67.5 | 75.9 | 59.1 | 59.9 | 62.7 | 58.2 | 41.8 | 74.9 | 67.4 | 48.2 | 74.2 | 61.4 |
| ETN [12] | 59.2 | 77.0 | 79.5 | 62.9 | 65.7 | 75.0 | 68.3 | 55.4 | 84.4 | 75.7 | 57.7 | 84.5 | 70.5 |
| JUMBOT [34] | 62.7 | 77.5 | 84.4 | 76.0 | 73.3 | 80.5 | 74.7 | 60.8 | 85.1 | 80.2 | 66.5 | 83.9 | 75.5 |
| AR [42] | 67.4 | 85.3 | 90.0 | 77.3 | 70.6 | 85.2 | 79.0 | 64.8 | 89.5 | 80.4 | 66.2 | 86.4 | 78.3 |
| m-POT [77] | 64.6 | 80.6 | 87.2 | 76.4 | 77.6 | 83.6 | 77.1 | 63.7 | 87.6 | 81.4 | 68.5 | 87.4 | 78.0 |
| MOT [65] | 63.1 | 86.1 | 92.3 | 78.7 | 85.4 | 89.6 | 79.8 | 62.3 | 89.7 | 83.8 | 67.0 | 89.6 | 80.6 |
| MOT + UOT(ETM + MROT-Ent) | 65.2 | 87.3 | 92.8 | 79.5 | 86.4 | 91.0 | 80.8 | 64.5 | 90.7 | 84.5 | 67.9 | 90.4 | 81.8 |
| MOT + UOT(ETM + MROT-Norm) | 65.8 | 88.0 | 93.1 | 79.9 | 86.2 | 91.3 | 81.4 | 64.9 | 91.2 | 84.9 | 68.3 | 90.7 | 82.1 |
| MOT + SemiUOT(Robust-SemiSinkhorn) | 66.0 | 88.2 | 93.0 | 80.5 | 86.8 | 91.5 | 81.3 | 65.2 | 91.6 | 85.2 | 68.5 | 90.9 | 82.4 |
| MOT + SemiUOT(ETM-Refine + MROT-Ent) | 68.6 | 90.4 | 94.2 | 83.7 | 89.5 | 93.9 | 83.5 | 67.4 | 93.9 | 88.4 | **71.8** | 92.1 | 84.8 |
| MOT + SemiUOT(ETM-Refine + MROT-Norm) | **69.1** | **90.7** | **94.6** | **84.0** | **90.3** | **94.0** | **83.8** | **67.9** | **94.4** | **88.5** | 71.3 | **93.6** | **85.2** |

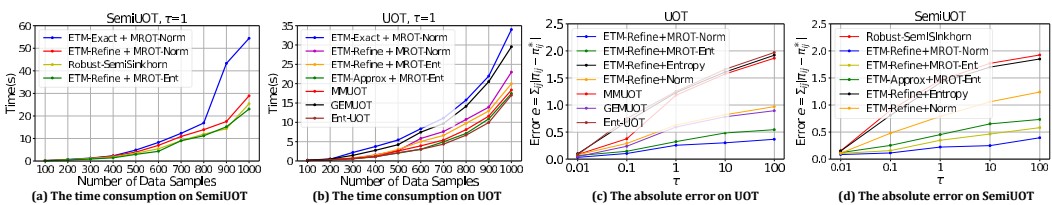

Figure 3: The time consumption and computation error analysis on UOT and SemiUOT.

the source and target domain distributions as $\mathbb{P}_X = \mathcal{N}\left(\begin{bmatrix} -1 \\ -1 \end{bmatrix}, \begin{bmatrix} 1 & 0 \\ 0 & 1 \end{bmatrix}\right)$ and $\mathbb{P}_Z = \mathcal{N}\left(\begin{bmatrix} 4 \\ 4 \end{bmatrix}, \begin{bmatrix} 1 & -0.8 \\ -0.8 & 1 \end{bmatrix}\right)$ following previous works [36, 17]. For the SemiUOT scenario, we set target distribution as $\mathbb{P}_Z$ then we sample 80% data from $\mathbb{P}_X$ with 20% outlier data to generate the source distribution. For the UOT scenario, we sample 80% data from $\mathbb{P}_X$ and $\mathbb{P}_Z$ accordingly while randomly sampling 20% outlier data for both $\mathbb{P}_X$ and $\mathbb{P}_Z$. **(2) Real-world Datasets.** We conduct the domain adaptation tasks on *Office-31* [88], *Office-Home* [102], and *ImageCLEF* [13]. More details are provided in Appendix I,J,K.

**Baselines.** We first compare the proposed ETM-Refine with MROT method with the following state-of-the-art UOT/SemiUOT solvers on the synthetic datasets. (1) **Ent-UOT** [83] utilizes the entropy regularization term on tackling UOT problem. (2) **MM-UOT** [17] adopts majority maximization algorithm for solving UOT. (3) **GEMUOT** [78] adopts $\ell_2$-norm term for reaching transport solutions on UOT which is the state-of-the-art approach. (4) **Robust-SemiSinkhorn** [52] adopts the entropy regularization term for solving SemiUOT problem. We also involve **DeepJDOT** [25], **ROT** [5], **JUMBOT** [34], **ETN** [12], **AR** [42], **m-POT** [77], **MOT** [65] as the model baselines for the real-world domain adaptation task. These model details are provided in Appendix.J.

**Implemented details.** For both synthetic and real-world datasets, we set $\epsilon = 0.01$ on both $\widehat{L}_{\mathrm{U}}$ and $\widehat{L}_{\mathrm{P}}$. We set $\eta_G = 10^2$ and $\eta_{\mathrm{Reg}} = 0.1$ for MROT in the calculation. The initial value of $\widehat{u}^{(0)}$ and $\widehat{f}^{(0)}$ as set as zero vectors. The initial sample weights are set to be equal, i.e., $a_i = 1/M$ and $b_j = 1/N$. And we adopt square Euclidean distance for the cost $C_{ij}$. Besides, we adopt the *same* framework and experimental settings of the UDA model JUMBOT [34] for unsupervised domain adaptation and the partial UDA model MOT [65] for partial unsupervised domain adaptation with the fair comparison. For all the experiments, we perform five random experiments and report the average results.

### 4.2 Performance and Extensive Analysis on Synthetic and Real-World Datasets

**Performance on Synthetic Datasets.** We sample 50 data samples on both source and target distributions for finding $\pi^*$ on UOT/SemiUOT. We first set $\tau = \{0.1, 100\}$ on SemiUOT and the matching solutions are shown in Fig.1(a)-(b). Note that we randomly sample 20% of noise in the source datasets. We can observe that previous method **Robust-SemiSinkhorn** could lead to ambiguous matching results. Our proposed ETM-Refine with MROT+Ent can reach relatively accurate results even if $\tau$ is large (e.g., $\tau = 100$). More importantly, ETM-Refine with MROT-Norm can achieve more precise results comparing with ETM-Refine with MROT-Ent shown in Fig.1. Then we also set $\tau_a = \tau_b = \{0.1, 100\}$ on UOT and the matching solutions are shown in Fig.2(a)-(b). From that we can observe: (1) **Ent-UOT** could merely provide dense matching solutions which are inaccurate. (2) **MM-UOT** obtains relatively accurate solutions when $\tau$ is small. However, **MM-UOT**

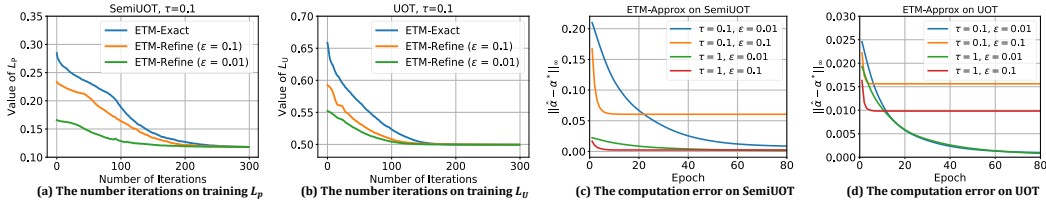

Figure 4: The effects on tuning different $\epsilon = \{0.01, 0.1\}$ with the loss descent curve and computation error $e_\alpha = ||\widehat{\alpha} - \alpha^*||_\infty$ for ETM-Approx method on solving SemiUOT and UOT problems.

Table 2: Classification accuracy (%) on *Office-31* and *ImageCLEF* for partial UDA

| Method | A→W | D→W | W→D | A→D | D→A | W→A | Avg | I→P | P→I | I→C | C→I | C→P | P→C | Avg |
|---|---|---|---|---|---|---|---|---|---|---|---|---|---|---|
| ResNet [43] | 75.6 | 96.3 | 98.1 | 83.4 | 83.9 | 85.0 | 87.1 | 78.3 | 86.9 | 91.0 | 84.3 | 72.5 | 91.5 | 84.1 |
| ETN [12] | 84.7 | 97.4 | 99.2 | 91.3 | 90.2 | 92.8 | 92.6 | 79.6 | 88.5 | 92.9 | 87.2 | 74.1 | 93.4 | 86.0 |
| JUMBOT [34] | 90.2 | 98.9 | 99.3 | 94.5 | 93.8 | 93.4 | 95.0 | 80.1 | 91.3 | 93.6 | 90.9 | 75.7 | 94.2 | 87.6 |
| AR [42] | 93.5 | **100.0** | 99.7 | 96.8 | 95.5 | 96.0 | 96.9 | 83.1 | 92.8 | 94.5 | 92.4 | 76.3 | 95.0 | 89.0 |
| m-POT [77] | 96.2 | 99.5 | **100.0** | 97.6 | 94.4 | 95.3 | 97.2 | 82.6 | 94.1 | 96.3 | 94.7 | 78.5 | 96.2 | 90.4 |
| MOT [65] | 99.3 | **100.0** | **100.0** | 98.7 | 96.1 | 96.4 | 98.4 | 87.7 | 95.0 | 98.0 | 95.0 | 87.0 | 98.7 | 93.6 |
| MOT + UOT(ETM-Refine + MROT-Ent) | 99.4 | **100.0** | **100.0** | 98.9 | 96.8 | 97.3 | 98.7 | 88.3 | 95.6 | 98.4 | 95.3 | 87.6 | 99.0 | 94.0 |
| MOT + UOT(ETM-Refine + MROT-Norm) | 99.6 | **100.0** | **100.0** | 99.2 | 97.3 | 97.7 | 99.0 | 88.7 | 95.9 | 98.7 | 95.8 | 88.0 | 99.1 | 94.4 |
| MOT + SemiUOT(ETM-Refine + MROT-Ent) | 99.7 | **100.0** | **100.0** | 99.4 | 97.8 | 98.4 | 99.2 | 89.1 | 96.2 | 99.2 | 96.1 | 88.5 | 99.4 | 94.8 |
| MOT + SemiUOT(ETM-Refine + MROT-Norm) | **99.8** | **100.0** | **100.0** | **99.7** | **98.4** | **98.8** | **99.5** | **89.6** | **96.7** | **99.4** | **96.5** | **89.1** | **99.6** | **95.2** |

cannot better handle the case when $\tau$ is large (e.g., $\tau = 100$) due to the deterioration of majority maximization algorithm. (3) **GEMUOT** can even reach more sparse matching solution against **Ent-UOT** and **MM-UOT** with the aid of $\ell_2$-norm term. However, the matching results obtained from **GEMUOT** remain coarse and ambiguous, especially when $\tau$ is large. (4) Benefiting from multipliers $s$, ETM-Refine with MROT-Norm achieves the most accurate results among existing UOT methods.

**Performance on Real-World Datasets.** We further conduct the experiments on the real-world datasets to validate our proposed method. The experimental UDA task results on *Office-Home* are shown in Table.1. We also directly adopt $\ell_2$-norm [8] and sparse solver [58] on solving $\mathcal{J}_U$. Meanwhile, our proposed ETM-Refine with MROT-Norm obtains the best performance, which indicates its efficacy for finding more accurate matching results on UDA. Then we adopt the same experimental setting as **MOT** to evaluate the partial UDA task where target label space is a subset of source label space. This makes it more challenging than classic UDA task [11, 66]. The partial UDA results on *Office-Home*, *Office-31* and *ImageCLEF* are also shown in Table.1 and Table.2. We can easily observe that all methods using EMT-Refine with MROT-Norm or MROT-Ent significantly improve the performance on partial UDA task. Especially, MOT + SemiUOT (ETM-Refine + MROT-Norm) achieves the best performance, benefiting from its powerful matching capability. UOT relaxes the dual transportation constraints, causing some target samples cannot be transported to the source domain. However, SemiUOT overcomes the mentioned issue while avoiding negative transfer in partial UDA, which boosts the model performance. We also conduct more domain adaptation experiments to verify the effects of ETM-Refine with MROT-Norm in Appendix.J, K, L.

**Solver Comparison Analysis.** To further analyze the proposed ETM-based methods with MROT, we conduct the solver comparison in terms of *computation time* and *computation error*. We first sample the same number of source/target data samples from $\mathbb{P}_X$ and $\mathbb{P}_Z$, respectively. As shown in Fig.3(a)-(b), we conduct the experiments on both UOT and SemiUOT with $\tau_a = \tau_b = \tau = 1$. We can conclude that ETM-Exact with MROT-Norm is most time-consuming, due to directly seeking the optimum from a random initial point. Meanwhile, ETM-Refine reaches a similar computation time with ETM-Approx, validating that it accelerates the process of finding the optimal $u^*$ or $f^*$ by utilizing $\widehat{u}^*$ or $\widehat{f}^*$. Moreover, we calculate the absolute computation error between matching solution $\pi$ learned by ETM with MROT and the standard UOT/SemiUOT solution with CVXPY as $\pi^*$, i.e., $e = \sum_{i,j} ||\pi_{ij} - \pi_{ij}^*||_1$. We sample 500 number of data samples ranging from $\tau_a = \tau_b = \tau = \{0.01, 0.1, 1, 10, 100\}$ for calculation and the results are shown in Fig.3(c)-(d). We can observe that although ETM-Approx with MROT-Ent has the fastest computation speed, the provided results $\pi$ still have the highest error compared to the ground truth $\pi^*$. Meanwhile ETM-Refine with MROT-Norm can further reach more accurate solutions against MROT-Ent. Though regularization terms, e.g., entropy and norm, can be directly used in SemiUOT and UOT, they bring larger computation errors without the guidance of KKT-multiplier $s$. **Robust-SemiSinkhorn** causes the largest computation error in solving SemiUOT. We also find that all existing UOT methods, i.e., **Ent-UOT**, **MM-UOT**, and **GEMUOT**, not only underperform in matching, but also cost more

computation time. We can observe that ETM-Refine method achieves much better results, especially when $\tau$ is relatively large, which is consistent with our discovery in Fig.1-Fig.2.

**Parameter sensitivity Analysis.** We finally study the effects of hyper-parameters on model performance. We tune $\epsilon$ in range of $\epsilon \in \{0.01, 0.1\}$ and show the results in Fig.4(a)-(d). We can observe that smaller $\epsilon$ could provide good approximation on UOT/SemiUOT, reducing the iteration steps for optimizing $L_\text{U}$ and $L_\text{P}$. Although $\epsilon$ could hardly affect the performance on ETM-Refine, larger value on $\epsilon$ could consume more iteration steps for solving $L_\text{U}$ and $L_\text{P}$ since the initial values are more random. Additionally, we collect the computation error $e_\alpha = ||\widehat{\alpha} - \alpha^*||_\infty$, which measures the discrepancy between the marginal probability learned via ETM-Approx $\widehat{\alpha}$ and the ground truth $\alpha^*$. Larger values of $\epsilon$ may fail to reduce the computation error $e_\alpha$ when compared to smaller values of $\epsilon$. Hence we set $\epsilon = 0.01$ empirically and more experimental results can be found in Appendix.M, N.

## 5 Related Works

**Unbalanced and Semi-Unbalanced Optimal Transport.** (1) *Related works on UOT*: UOT with KL divergence has been widely investigated for dealing with diverse applications [82, 30, 94, 51, 40, 32, 106, 64]. Different types of UOT solutions can be distinguished in terms of using entropy regularization term or not. Involving entropy in UOT can enhance the model scalability, yet resulting in dense matching results [98, 5]. Latest, [17] further considers UOT without entropy terms by Majorization-Minimization (MM) [20, 99] or regularization path methods [68, 69, 57]. However, the nature of MM algorithm inherits inexact proximal point of KL term [105], which still causes dense mapping when $\tau$ becomes larger. Meanwhile regularization path methods could be quite slow in computation, especially when $\tau \to +\infty$. Furthermore, as the number of samples increases, it can lead to high storage space consumption, which can be problematic. Recently, [21] discovers a similar transformation between continuous UOT and classic OT problem. However, this discovery cannot directly extend to SemiUOT and UOT in discrete scenarios, and provides no hint to compute the exact marginal distributions and corresponding matching $\pi$. (2) *Related works on SemiUOT*: SemiUOT with KL divergence only relaxes one of the marginal constraints comparing with UOT. [52] first fully investigated the corresponding problem and proposed Robust-SemiSinkhorn algorithm. Nevertheless, it still suffers from inaccurate matching solutions with entropy regularization term. Currently, there are only extremely few works for solving SemiUOT [71]. Therefore, how to efficiently provide accurate matching solutions on both discrete SemiUOT and UOT is still a challenging problem.

## 6 Conclusion

In this paper, we propose Equivalent Transformation Mechanism (ETM) approach with ETM-Exact, ETM-Approx, and ETM-Refine to solve the marginal probabilities of SemiUOT and UOT. We illustrate that the essence of SemiUOT/UOT is reweighting data samples accordingly and thus SemiUOT/UOT problem can be transformed into standard optimal transport. Moreover, we propose KKT-Multiplier Regularization with Multiplier Regularized Optimal Transport (MROT) to obtain more accurate solutions. We conduct extensive experiments to show the superior performance of ETM with MROT, on both synthetic and real-world datasets of different tasks and applications.

## Acknowledgments and Disclosure of Funding

This work was supported by the National Natural Science Foundation of China (No. 62572486), Natural Science Foundation of Shandong Province (No. ZR2023MF007). This work is also partly supported by the National Research Foundation (NRF), Singapore, through the AI Singapore Programme under the project titled "AI-based Urban Cooling Technology Development"(Award No. AISG3-TC-2024-014-SGKR), partly supported by the National Research Foundation (NRF), Singapore, through the AI Singapore Programme under the project titled "Learning Assisted Human-AI Collaboration for Large-scale Practical Combinatorial Optimization" (AISG Award No: AISG3-RP-2022-031).

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

# Appendix

## A    Notation Table

We provide the important notations and their descriptions for clarification on Table.3.

| Symbol | Description |
|---|---|
| $\boldsymbol{X} \in \mathbb{R}^{M \times D}$ | Source domain data matrix |
| $\boldsymbol{Z} \in \mathbb{R}^{N \times D}$ | Target domain data matrix |
| $M$ | Number of source samples |
| $N$ | Number of target samples |
| $D$ | Data dimension |
| $\boldsymbol{a} \in \mathbb{R}^M$ | Source marginal probability vector |
| $\boldsymbol{b} \in \mathbb{R}^N$ | Target marginal probability vector |
| $\boldsymbol{\pi} \in \mathbb{R}^{M \times N}$ | Coupling matching matrix (transport plan) |
| $\boldsymbol{C} \in \mathbb{R}^{M \times N}$ | Cost (distance) matrix |
| $\tau$ | KL divergence coefficient for SemiUOT |
| $\tau_a$ | KL divergence coefficient for source (UOT) |
| $\tau_b$ | KL divergence coefficient for target (UOT) |
| $\boldsymbol{\alpha}$ | Adjusted source marginal (via ETM) |
| $\boldsymbol{\beta}$ | Adjusted target marginal (via ETM) |
| $\boldsymbol{f}$ | Dual variable (SemiUOT, source) |
| $\boldsymbol{g}$ | Dual variable (SemiUOT, target) |
| $\boldsymbol{u}$ | Dual variable (UOT, source) |
| $\boldsymbol{v}$ | Dual variable (UOT, target) |
| $\zeta$ | Scalar dual variable (shared) |
| $\boldsymbol{s}$ | KKT multiplier variable |
| $\omega_{\mathrm{L}}$ | Scaling factor: $\omega_{\mathrm{L}} = \frac{\langle \boldsymbol{b}, \mathbf{1} \rangle}{\langle \boldsymbol{a}, \mathbf{1} \rangle}$ |
| $\epsilon$ | LogSumExp smoothing parameter |
| $\nu$ | Update step: $\nu = \frac{\tau \epsilon}{\tau + \epsilon}$ |
| $J_{\mathrm{SemiUOT}}$ | Objective function of SemiUOT |
| $J_{\mathrm{UOT}}$ | Objective function of UOT |
| $L_P$ | Exact SemiUOT Equation |
| $\hat{L}_P$ | Approximate SemiUOT Equation |
| $L_U$ | Exact UOT Equation |
| $\hat{L}_U$ | Approximate UOT Equation |
| $G(\boldsymbol{\pi}, \boldsymbol{s})$ | KKT-multiplier regularization term: $\langle \boldsymbol{\pi}, \boldsymbol{s} \rangle$ |
| $L_{\mathrm{Reg}}(\boldsymbol{\pi})$ | Regularization term for OT (e.g. entropy or $\ell_2$) |
| $\eta_G$ | Regularization weight for multiplier term |
| $\eta_{\mathrm{Reg}}$ | Regularization weight (entropy or $\ell_2$) |
| $\tilde{C}_{ij}$ | Adjusted cost: $\tilde{C}_{ij} = C_{ij} + \eta_G s_{ij}$ |
| $\boldsymbol{\psi}$ | Dual variable in MROT (source side) |
| $\boldsymbol{\phi}$ | Dual variable in MROT (target side) |

Table 3: Important notations

## B    Proof of Proposition 1

**Proposition 1.** (Principles of Equivalent Transformation Mechanism for SemiUOT) *Given SemiUOT with KL-Divergence* $J_{\mathrm{SemiUOT}}$*, one can obtain its Fenchel-Lagrange multipliers form as:*

$$\min_{\boldsymbol{f}, \boldsymbol{g}, \zeta} \left[ \tau \sum_{i=1}^{M} a_i \exp\left( -\frac{f_i + \zeta}{\tau} \right) - \sum_{j=1}^{N} b_j (g_j - \zeta) \right] \tag{16}$$

$$s.t.\ f_i + g_j + s_{ij} = C_{ij}, \quad s_{ij} \geq 0.$$

*where $\boldsymbol{f}$, $\boldsymbol{g}$, $\boldsymbol{s}$ and $\zeta$ denote Lagrange multipliers. Moreover, SemiUOT problem can be further transformed into the form of optimal transport with marginal constraints as follows:*

$$\min_{\boldsymbol{\pi} \geq 0} \mathcal{J}_{\mathrm{P}} = \langle \boldsymbol{C}, \boldsymbol{\pi} \rangle,$$

$$s.t. \; \boldsymbol{\pi} \mathbf{1}_N = \boldsymbol{a} \odot \exp\left(-\frac{\boldsymbol{f}^* + \zeta^*}{\tau}\right) = \boldsymbol{\alpha}, \quad \boldsymbol{\pi}^\top \mathbf{1}_M = \boldsymbol{b}. \tag{17}$$

*When $\tau \to \infty$, the source marginal probability is given as $\boldsymbol{\pi} \mathbf{1}_N = \omega_{\mathrm{L}} \boldsymbol{a}$ and $\omega_{\mathrm{L}} = \langle \boldsymbol{b}, \mathbf{1}_N \rangle / \langle \boldsymbol{a}, \mathbf{1}_M \rangle$.*

*Proof.* To start with, we first review the definition of SemiUOT as shown below:

$$\min_{\pi_{ij} \geq 0} J_{\mathrm{SemiUOT}} = \langle \boldsymbol{C}, \boldsymbol{\pi} \rangle + \tau \mathrm{KL}\left(\boldsymbol{\pi} \mathbf{1}_N \| \boldsymbol{a}\right)$$

$$s.t. \; \boldsymbol{\pi}^\top \mathbf{1}_M = \boldsymbol{b}. \tag{18}$$

Then we can rewrite the optimization problem:

$$\min_{\boldsymbol{\pi} \geq 0} J = \langle \boldsymbol{C}, \boldsymbol{\pi} \rangle + \tau \mathrm{KL}\left(\boldsymbol{\pi} \mathbf{1}_N \| \boldsymbol{a}\right)$$

$$s.t. \begin{cases} (\text{Constraint}): \; \boldsymbol{\pi}^\top \mathbf{1}_M = \boldsymbol{b} \\ (\text{Optional}): \; \boldsymbol{\pi} \mathbf{1}_N = \boldsymbol{\alpha} \end{cases}. \tag{19}$$

Note that we do not need to know the exact value of $\boldsymbol{\alpha}$ beforehand. We adopt this optional constraint only for simplifying the following deduction. The Lagrange multipliers of SemiUOT with KL-Divergence is given as:

$$\max_{\boldsymbol{s} \geq 0, \boldsymbol{f}, \boldsymbol{g}, \zeta} \min_{\boldsymbol{\pi} \geq 0} \mathcal{J}_{\mathrm{SemiUOT}} = \tau \mathrm{KL}\left(\boldsymbol{\pi} \mathbf{1}_N \| \boldsymbol{a}\right) + \langle \boldsymbol{f} + \zeta, \boldsymbol{\pi} \mathbf{1}_N \rangle + \langle \boldsymbol{g} - \zeta, \boldsymbol{b} \rangle + $$

$$\langle \boldsymbol{C} - \boldsymbol{f} \otimes \mathbf{1}_N^\top - \mathbf{1}_M \otimes \boldsymbol{g}^\top - \boldsymbol{s}, \boldsymbol{\pi} \rangle, \tag{20}$$

where $\boldsymbol{f}$, $\boldsymbol{g}$, $\boldsymbol{s}$ and $\zeta$ are dual variables. By taking the differentiation on $\pi_{ij}$ we have:

$$\frac{\partial \mathcal{J}_{\mathrm{SemiUOT}}}{\partial \pi_{ij}} = \left[\tau \log \frac{\sum_{j=1}^N \pi_{ij}}{a_i} + f_i + \zeta\right] + (C_{ij} - f_i - g_j - s_{ij})$$

$$= C_{ij} + \tau \log \frac{\sum_{j=1}^N \pi_{ij}}{a_i} + \zeta - g_j - s_{ij} \tag{21}$$

$$= 0.$$

Therefore, we can obtain the results as:

$$\begin{cases} \sum_{j=1}^N \pi_{ij} = a_i \exp\left(-\frac{f_i + \zeta}{\tau}\right) \\ \sum_{i=1}^M \pi_{ij} = b_j \\ C_{ij} - f_i - g_j - s_{ij} = 0 \end{cases}. \tag{22}$$

After that, we can take these back into KL-Divergence to simplify the calculation:

$$\tau \mathrm{KL}\left(\boldsymbol{\pi} \mathbf{1}_N \| \boldsymbol{a}\right) + \langle \boldsymbol{f} + \zeta, \boldsymbol{\pi} \mathbf{1}_N \rangle$$

$$= \tau \mathrm{KL}\left(\boldsymbol{a} \exp\left(-\frac{\boldsymbol{f} + \zeta}{\tau}\right) \| \boldsymbol{a}\right) + \left\langle \boldsymbol{f} + \zeta, \boldsymbol{a} \exp\left(-\frac{\boldsymbol{f} + \zeta}{\tau}\right) \right\rangle$$

$$= \tau \sum_{i=1}^M \left[a_i \exp\left(-\frac{f_i + \zeta}{\tau}\right) \log \frac{a_i \exp\left(-\frac{f_i + \zeta}{\tau}\right)}{a_i} - a_i \exp\left(-\frac{f_i + \zeta}{\tau}\right) + a_i\right] + \sum_{i=1}^M (f_i + \zeta) a_i \exp\left(-\frac{f_i + \zeta}{\tau}\right)$$

$$= \sum_{i=1}^M \left[-\tau a_i \exp\left(-\frac{f_i + \zeta}{\tau}\right) + \tau a_i\right]. \tag{23}$$

Therefore we can obtain its Fenchel-Lagrange multipliers form of SemiUOT as:

$$\min_{\boldsymbol{f},\boldsymbol{g},\zeta} \mathcal{J}_{\text{SemiUOT}} = -\tau \text{KL}\left(\boldsymbol{\pi}\boldsymbol{1}_N \| \boldsymbol{a}\right) - \langle \boldsymbol{f} + \zeta, \boldsymbol{\pi}\boldsymbol{1}_N \rangle - \langle \boldsymbol{g} - \zeta, \boldsymbol{\pi}^\top \boldsymbol{1}_M \rangle$$

$$= \tau \exp\left(-\frac{\zeta}{\tau}\right)\left\langle \boldsymbol{a}, \exp\left(-\frac{\boldsymbol{f}}{\tau}\right)\right\rangle - \langle \boldsymbol{g} - \zeta, \boldsymbol{b}\rangle + \mathcal{O}_{\text{Const}} \tag{24}$$

$$s.t. \ f_i + g_j \le C_{ij},$$

where $\mathcal{O}_{\text{Const}} = -\sum_{i=1}^{M} \tau a_i$ and we can neglect it during the following calculation. Once we obtain the optimal solution on $\boldsymbol{f}^*$, $\boldsymbol{g}^*$ and $\zeta^*$, we will discover that:

$$\tau \text{KL}\left(\boldsymbol{\pi}\boldsymbol{1}_N \| \boldsymbol{a}\right) = \tau \text{KL}\left(\boldsymbol{a}, \exp\left(-\frac{\boldsymbol{f}^* + \zeta^*}{\tau}\right) \| \boldsymbol{a}\right) = \text{Const.} \tag{25}$$

Hence SemiUOT problem can be transformed into classic optimal transport problem accordingly. Finally we can obtain the optimal solution on $\zeta$ by considering $\frac{\partial \mathcal{J}_{\text{SemiUOT}}}{\partial \zeta} = 0$ as below:

$$\zeta = \tau \left[\log\left(\sum_{i=1}^{M} a_i \exp\left(-\frac{f_i}{\tau}\right)\right) - \log\left(\sum_{j=1}^{N} b_j\right)\right]. \tag{26}$$

Once we set $\tau \to \infty$, the results of the limitation will be shown as:

$$\lim_{\tau \to +\infty} a_i \exp\left(-\frac{f_i + \zeta}{\tau}\right) = \lim_{\tau \to +\infty} a_i \exp\left(-\frac{\zeta}{\tau}\right) = a_i \frac{\langle \boldsymbol{b}, \boldsymbol{1}_N \rangle}{\langle \boldsymbol{a}, \boldsymbol{1}_M \rangle} = \omega_{\text{L}} a_i. \tag{27}$$

Therefore we conclude the proof of the Proposition 1. $\qquad\square$

## C   Proof of Proposition 2

**Proposition 2.** (Calculation for Approximate SemiUOT Equation) *Given Approximate SemiUOT equation $\widehat{L}_{\text{P}}$, it can be optimized via Equivalent Transformation Mechanism with Approximation (ETM-Approx). That is, ETM-Approx aims to solve the following equation for each $\widehat{f}_s$:*

$$\frac{\partial \widehat{L}_{\text{P}}}{\partial \widehat{f}_s} = -a_s \exp\left(-\frac{\widehat{f}_s + \zeta}{\tau}\right) + \exp\left(\frac{\widehat{f}_s}{\epsilon}\right) \sum_{j=1}^{N}\left[\frac{b_j \exp\left(-\frac{C_{sj}}{\epsilon}\right)}{\sum_{k=1}^{M} \exp\left(\frac{\widehat{f}_k - C_{kj}}{\epsilon}\right)}\right] = 0. \tag{28}$$

*Specifically, we can adopt fixed-point iteration method for solving Eq.(28) at the $\ell$-th iteration as follows:*

$$\begin{cases} \widehat{f}_1^{\ell+1} = \nu\left[\log\left(a_1 \exp\left(-\frac{\zeta}{\tau}\right)\right) - \log\left[\sum_{j=1}^{N}\left(\frac{b_j}{\mathscr{W}_{\epsilon,j}(\widehat{\boldsymbol{f}}^\ell)} \exp\left(-\frac{C_{1j}}{\epsilon}\right)\right)\right]\right] \\ \vdots \\ \widehat{f}_M^{\ell+1} = \nu\left[\log\left(a_M \exp\left(-\frac{\zeta}{\tau}\right)\right) - \log\left[\sum_{j=1}^{N}\left(\frac{b_j}{\mathscr{W}_{\epsilon,j}(\widehat{\boldsymbol{f}}^\ell)} \exp\left(-\frac{C_{Mj}}{\epsilon}\right)\right)\right]\right] \end{cases}, \tag{29}$$

*where $\nu = \tau\epsilon/(\tau + \epsilon)$ for simplification and $\mathscr{W}_{\epsilon,j}(\widehat{\boldsymbol{f}}^\ell)$ denotes the corresponding calculation:*

$$\mathscr{W}_{\epsilon,j}(\widehat{\boldsymbol{f}}^\ell) = \sum_{k=1}^{M} \exp\left(\frac{\widehat{f}_k^\ell - C_{kj}}{\epsilon}\right). \tag{30}$$

*The proposed procedure can be convergent with a theoretical guarantee. Finally, updating the Lagrange multiplier $\zeta$ by further considering $\nabla_\zeta \widehat{L}_{\text{P}} = 0$ via $\zeta = \tau[\log(\sum_{i=1}^{M} a_i \exp(-\widehat{f}_i/\tau)) - \log(\sum_{j=1}^{N} b_j)]$. One can achieve the optimal results on $\widehat{\boldsymbol{f}}^*$ and $\zeta^*$ via iterative computing accordingly.*

*Proof.* We first review the proposed Approximate SemiUOT Equation $\widehat{L}_P$ as below:

$$\min_{\widehat{\boldsymbol{f}}, \zeta} \widehat{L}_P = \tau \sum_{i=1}^{M} a_i e^{-\frac{\widehat{f}_i + \zeta}{\tau}} + \sum_{j=1}^{N} b_j \left[ \epsilon \log \left[ \sum_{k=1}^{M} e^{\frac{\widehat{f}_k - C_{kj}}{\epsilon}} \right] + \zeta \right]. \tag{31}$$

Then we consider optimizing $\widehat{f}_s$ as follows:

$$\frac{\partial \widehat{L}_P}{\partial \widehat{f}_s} = 0 \quad \Rightarrow \quad \exp\left( \frac{\tau + \epsilon}{\tau \epsilon} \widehat{f}_s \right) = \frac{a_s e^{-\frac{\zeta}{\tau}}}{\sum\limits_{j=1}^{N} \left( \frac{b_j \exp(-C_{sj}/\epsilon)}{\mathscr{W}_{\epsilon,j}(\widehat{\boldsymbol{f}})} \right)}, \tag{32}$$

where $\mathscr{W}_{\epsilon,j}(\widehat{\boldsymbol{f}})$ is defined as Eq.(30). At that time we adopt fixed-point iteration method to optimize $\widehat{\boldsymbol{f}}$ accordingly:

$$\begin{cases} \widehat{f}_1^{\ell+1} = \nu \left[ \log\left( a_1 e^{-\frac{\zeta}{\tau}} \right) - \log\left[ \sum\limits_{j=1}^{N} \left( \frac{b_j e^{-C_{1j}/\epsilon}}{\mathscr{W}_{\epsilon,j}(\widehat{\boldsymbol{f}}^\ell)} \right) \right] \right] = \mathcal{F}_1\left( \widehat{f}_1^\ell, \cdots, \widehat{f}_M^\ell \right) \\ \vdots \\ \widehat{f}_s^{\ell+1} = \nu \left[ \log\left( a_s e^{-\frac{\zeta}{\tau}} \right) - \log\left[ \sum\limits_{j=1}^{N} \left( \frac{b_j e^{-C_{sj}/\epsilon}}{\mathscr{W}_{\epsilon,j}(\widehat{\boldsymbol{f}}^\ell)} \right) \right] \right] = \mathcal{F}_s\left( \widehat{f}_1^\ell, \cdots, \widehat{f}_M^\ell \right) \\ \vdots \\ \widehat{f}_M^{\ell+1} = \nu \left[ \log\left( a_M e^{-\frac{\zeta}{\tau}} \right) - \log\left[ \sum\limits_{j=1}^{N} \left( \frac{b_j e^{-C_{Mj}/\epsilon}}{\mathscr{W}_{\epsilon,j}(\widehat{\boldsymbol{f}}^\ell)} \right) \right] \right] = \mathcal{F}_M\left( \widehat{f}_1^\ell, \cdots, \widehat{f}_M^\ell \right) \end{cases}, \tag{33}$$

where $\nu = \frac{\tau \epsilon}{\tau + \epsilon}$. By taking the gradient on $\mathcal{F}_s(\widehat{f}_s^\ell)$ w.r.t $\widehat{f}_s^\ell$, we can observe that:

$$\frac{\partial \mathcal{F}_s(\widehat{f}_s^\ell)}{\partial \widehat{f}_s^\ell} = -\frac{\tau \epsilon}{\tau + \epsilon} \frac{1}{\sum\limits_{j=1}^{N} \left[ \frac{\exp\left( -\frac{C_{sj}}{\epsilon} \right)}{\mathscr{W}_{\epsilon,j}(\widehat{\boldsymbol{f}}^\ell)} \right] b_j} \frac{\partial}{\partial \widehat{f}_s^\ell} \left( \sum_{j=1}^{N} \left[ \frac{\exp\left( -\frac{C_{sj}}{\epsilon} \right)}{\mathscr{W}_{\epsilon,j}(\widehat{\boldsymbol{f}}^\ell)} \right] b_j \right)$$

$$= \frac{\tau}{\tau + \epsilon} \frac{1}{\sum\limits_{j=1}^{N} \left[ \frac{\exp\left( -\frac{C_{sj}}{\epsilon} \right)}{\mathscr{W}_{\epsilon,j}(\widehat{\boldsymbol{f}}^\ell)} \right] b_j} \underbrace{\sum_{j=1}^{N} \left[ \frac{b_j \exp\left( -\frac{C_{sj}}{\epsilon} \right)}{\mathscr{W}_{\epsilon,j}(\widehat{\boldsymbol{f}}^\ell)} \cdot \frac{\exp\left( \frac{\widehat{f}_s^\ell - C_{sj}}{\epsilon} \right)}{\mathscr{W}_{\epsilon,j}(\widehat{\boldsymbol{f}}^\ell)} \right]}_{<1} \tag{34}$$

$$< 1.$$

Likewise we can obtain the result:

$$\mathscr{F}_s\left( \widehat{f}_1^\ell, \cdots, \widehat{f}_M^\ell \right) = \left| \frac{\partial \mathcal{F}_s(\widehat{f}_1^\ell)}{\partial \widehat{f}_1^\ell} \right| + \cdots + \left| \frac{\partial \mathcal{F}_s(\widehat{f}_s^\ell)}{\partial \widehat{f}_s^\ell} \right| + \cdots + \left| \frac{\partial \mathcal{F}_s(\widehat{f}_M^\ell)}{\partial \widehat{f}_M^\ell} \right|$$

$$= \frac{\tau}{\tau + \epsilon} \frac{1}{\sum\limits_{j=1}^{N} \left[ \frac{\exp\left( -\frac{C_{sj}}{\epsilon} \right)}{\mathscr{W}_{\epsilon,j}(\widehat{\boldsymbol{f}}^\ell)} \right] b_j} \sum_{j=1}^{N} \left[ \frac{b_j \exp\left( -\frac{C_{sj}}{\epsilon} \right)}{\mathscr{W}_{\epsilon,j}(\widehat{\boldsymbol{f}}^\ell)} \cdot \sum_{u=1}^{M} \left( \frac{\exp\left( \frac{\widehat{f}_u^\ell - C_{uj}}{\epsilon} \right)}{\mathscr{W}_{\epsilon,j}(\widehat{\boldsymbol{f}}^\ell)} \right) \right]$$

$$= \frac{\tau}{\tau + \epsilon} < 1. \tag{35}$$

We can easily conclude that:

$$
\begin{cases}
\mathscr{F}_1\left(\widehat{f}_1^{\ell}, \cdots, \widehat{f}_M^{\ell}\right) < 1 \\
\vdots \\
\mathscr{F}_s\left(\widehat{f}_1^{\ell}, \cdots, \widehat{f}_M^{\ell}\right) < 1 \\
\vdots \\
\mathscr{F}_M\left(\widehat{f}_1^{\ell}, \cdots, \widehat{f}_M^{\ell}\right) < 1
\end{cases} . \tag{36}
$$

Therefore, we can conclude that the proposed ETM-Approx method guarantees convergence according to Theorem 2.9 in [70]. □

**Remark 1.** *ETM-Approx can reach the linear convergence rate via the fixed-point iteration shown as* $\mathcal{O}(NM\log(1/\varepsilon_{\mathrm{err}}))$ *where* $\varepsilon_{\mathrm{err}} = ||\widehat{\boldsymbol{f}} - \widehat{\boldsymbol{f}}^*||_\infty$ *and* $\widehat{\boldsymbol{f}}^*$ *denotes the optimal solution.*

*Proof.* We can formulate the whole optimization process for Proposition 2 as below :

$$
\begin{cases}
\widehat{f}_1^{\ell+1} = \mathcal{F}_1\left(\widehat{f}_1^{\ell}, \cdots, \widehat{f}_M^{\ell}\right) \\
\vdots \\
\widehat{f}_s^{\ell+1} = \mathcal{F}_s\left(\widehat{f}_1^{\ell}, \cdots, \widehat{f}_M^{\ell}\right) \\
\vdots \\
\widehat{f}_M^{\ell+1} = \mathcal{F}_M\left(\widehat{f}_1^{\ell}, \cdots, \widehat{f}_M^{\ell}\right)
\end{cases}
\Rightarrow
\left(\widehat{f}_1^{\ell+1}, \cdots, \widehat{f}_M^{\ell+1}\right) = \mathcal{F}_{\mathrm{update}}\left(\widehat{f}_1^{\ell}, \cdots, \widehat{f}_M^{\ell}\right). \tag{37}
$$

According to the above discussion, we have the following results:

$$
\begin{aligned}
\left\|\widehat{\boldsymbol{f}}^{\ell+1} - \widehat{\boldsymbol{f}}^*\right\|_\infty &= \left\|\mathcal{F}_{\mathrm{update}}\left(\widehat{\boldsymbol{f}}^{\ell}\right) - \mathcal{F}_{\mathrm{update}}\left(\widehat{\boldsymbol{f}}^*\right)\right\|_\infty \\
&\le \frac{\tau}{\tau+\epsilon}\left\|\widehat{\boldsymbol{f}}^{\ell} - \widehat{\boldsymbol{f}}^*\right\|_\infty \\
&\le \frac{\tau}{\tau+\epsilon}\left\|\widehat{\boldsymbol{f}}^{\ell+1} - \widehat{\boldsymbol{f}}^*\right\|_\infty + \frac{\tau}{\tau+\epsilon}\left\|\widehat{\boldsymbol{f}}^{\ell+1} - \widehat{\boldsymbol{f}}^{\ell}\right\|_\infty .
\end{aligned} \tag{38}
$$

Therefore the error between the solution $\widehat{\boldsymbol{f}}^{\ell+1}$ at the $(\ell+1)$ iteration and the optimal results $\widehat{\boldsymbol{f}}^*$ is given as:

$$
\left\|\widehat{\boldsymbol{f}}^{\ell+1} - \widehat{\boldsymbol{f}}^*\right\|_\infty \le \frac{\tau+\epsilon}{\epsilon}\left(\frac{\tau}{\tau+\epsilon}\right)^{\ell}\left\|\widehat{\boldsymbol{f}}^{(1)} - \widehat{\boldsymbol{f}}^{(0)}\right\|_\infty . \tag{39}
$$

Hence ETM-Approx can be linear convergence via the fixed-point iteration shown as $\mathcal{O}(NM\log(1/\varepsilon_{\mathrm{err}}))$ where $\varepsilon_{\mathrm{err}} = ||\widehat{\boldsymbol{f}} - \widehat{\boldsymbol{f}}^*||_\infty$ and $\widehat{\boldsymbol{f}}^*$ denotes the optimal solution. □

## D   Algorithm for ETM-Based Method on SemiUOT

We also provide the pseudo algorithm of the proposed ETM-Based approachs (e.g., ETM-Exact, ETM-Approx, and ETM-Refine) for solving SemiUOT in Alg.1 to make a clearer illustration.

## E   Proof of Proposition 3

**Proposition 3.** (Principles of Equivalent Transformation Mechanism for UOT) *Given UOT with KL-Divergence* $J_{\mathrm{UOT}}$*, its Fenchel-Lagrange multipliers form is given:*

$$
\min_{\boldsymbol{u},\boldsymbol{v},\zeta}\left[\tau_a\sum_{i=1}^{M}a_i e^{-\frac{u_i+\zeta}{\tau_a}} + \tau_b\sum_{j=1}^{N}b_j e^{-\frac{v_j-\zeta}{\tau_b}}\right] \tag{40}
$$

$$
s.t.\ u_i + v_j + s_{ij} = C_{ij}, \quad s_{ij} \ge 0,
$$

*where $\boldsymbol{u}$, $\boldsymbol{v}$, $\boldsymbol{s}$ and $\zeta$ denote Lagrange multipliers. Moreover, UOT problem can also be transformed into classic optimal transport as follows:*

$$\min_{\boldsymbol{\pi} \geq 0} \mathcal{J}_{\mathrm{U}} = \langle \boldsymbol{C}, \boldsymbol{\pi} \rangle$$

$$s.t. \begin{cases} \boldsymbol{\pi}\mathbf{1}_N = \boldsymbol{a} \odot \exp\left(-\dfrac{\boldsymbol{u}^* + \zeta^*}{\tau_a}\right) = \boldsymbol{\alpha} \\ \boldsymbol{\pi}^\top \mathbf{1}_M = \boldsymbol{b} \odot \exp\left(-\dfrac{\boldsymbol{v}^* - \zeta^*}{\tau_b}\right) = \boldsymbol{\beta} \end{cases}. \tag{41}$$

*Note that when $\tau_a, \tau_b \to \infty$, the source and target marginal probabilities can be determined as $\boldsymbol{\pi}\mathbf{1}_N = \sqrt{\omega_{\mathrm{L}}}\boldsymbol{a}$ and $\boldsymbol{\pi}^\top\mathbf{1}_M = \boldsymbol{b}/\sqrt{\omega_{\mathrm{L}}}$ where $\omega_{\mathrm{L}} = \langle \boldsymbol{b}, \mathbf{1}_N\rangle / \langle \boldsymbol{a}, \mathbf{1}_M\rangle$ respectively.*

*Proof.* To start with, we first rewrite the optimization problem as below:

$$\min_{\boldsymbol{\pi} \geq 0} J = \langle \boldsymbol{C}, \boldsymbol{\pi} \rangle + \tau_a \mathrm{KL}\left(\boldsymbol{\pi}\mathbf{1}_N \| \boldsymbol{a}\right) + \tau_b \mathrm{KL}(\boldsymbol{\pi}^\top \mathbf{1}_M \| \boldsymbol{b})$$

$$s.t. \text{ (Optional)}: \ \boldsymbol{\pi}\mathbf{1}_N = \boldsymbol{\alpha}, \quad \boldsymbol{\pi}^\top \mathbf{1}_M = \boldsymbol{\beta}. \tag{42}$$

where $\boldsymbol{\alpha}$ and $\boldsymbol{\beta}$ denote the marginal probabilities for source and target domains respectively. Note that we do not need the true value fo $\boldsymbol{\alpha}$ and $\boldsymbol{\beta}$ beforehand. That is, the constraints here are optional for the following UOT deduction. The Lagrange multipliers of UOT with KL-Divergence is given as:

$$\max_{\boldsymbol{s} \geq 0, \boldsymbol{u}, \boldsymbol{v}, \zeta} \min_{\boldsymbol{\pi} \geq 0} \mathcal{J}_{\mathrm{UOT}} = \tau_a \mathrm{KL}\left(\boldsymbol{\pi}\mathbf{1}_N \| \boldsymbol{a}\right) + \langle \boldsymbol{u} + \zeta, \boldsymbol{\pi}\mathbf{1}_N \rangle + \tau_b \mathrm{KL}(\boldsymbol{\pi}^\top \mathbf{1}_M \| \boldsymbol{b}) + \langle \boldsymbol{v} - \zeta, \boldsymbol{\pi}^\top \mathbf{1}_M \rangle + \mathscr{C}_{\mathrm{UOT}}, \tag{43}$$

where $\mathscr{C}_{\mathrm{UOT}} = \sum_{i,j}(C_{ij} - u_i - v_j - s_{ij})\pi_{ij} = \langle \boldsymbol{C} - \boldsymbol{u} \otimes \mathbf{1}_N^\top - \mathbf{1}_M \otimes \boldsymbol{v}^\top - \boldsymbol{s}, \boldsymbol{\pi}\rangle$ and $\boldsymbol{u}$, $\boldsymbol{v}$ and $\zeta$ are dual variables. By taking the differentiation on $\pi_{ij}$ we have:

$$\frac{\partial \mathcal{J}_{\mathrm{UOT}}}{\partial \pi_{ij}} = \left[\tau_a \log \frac{\sum\limits_{j=1}^{N} \pi_{ij}}{a_i} + u_i + \zeta\right] + \left[\tau_b \log \frac{\sum\limits_{i=1}^{M} \pi_{ij}}{b_j} + v_j - \zeta\right] + (C_{ij} - u_i - v_j - s_{ij})$$

$$= C_{ij} + \tau_a \log \frac{\sum\limits_{j=1}^{N} \pi_{ij}}{a_i} + \tau_b \log \frac{\sum\limits_{i=1}^{M} \pi_{ij}}{b_j} - s_{ij} = 0. \tag{44}$$

Then we can obtain the results:

$$\begin{cases} \sum\limits_{j=1}^{N} \pi_{ij} = a_i \exp\left(-\dfrac{u_i + \zeta}{\tau_a}\right) \\ \sum\limits_{i=1}^{M} \pi_{ij} = b_j \exp\left(-\dfrac{v_j - \zeta}{\tau_b}\right) \\ C_{ij} - u_i - v_j - s_{ij} = 0 \end{cases}. \tag{45}$$

By taking the above results into KL-Divergence, we can further simplify the results:

$$\begin{cases} \tau_a \mathrm{KL}\left(\boldsymbol{\pi}\mathbf{1}_N \| \boldsymbol{a}\right) + \langle \boldsymbol{u} + \zeta, \boldsymbol{\pi}\mathbf{1}_N \rangle = \sum\limits_{i=1}^{M}\left[-\tau_a a_i \exp\left(-\dfrac{f_i + \zeta}{\tau_a}\right) + \tau_a a_i\right] \\ \tau_b \mathrm{KL}\left(\boldsymbol{\pi}^\top\mathbf{1}_M \| \boldsymbol{b}\right) + \langle \boldsymbol{v} - \zeta, \boldsymbol{\pi}^\top\mathbf{1}_M \rangle = \sum\limits_{j=1}^{N}\left[-\tau_b b_j \exp\left(-\dfrac{g_j - \zeta}{\tau_b}\right) + \tau_b b_j\right] \end{cases}. \tag{46}$$

Therefore we can obtain its Fenchel-Lagrange multipliers form of UOT as:

$$\min_{\boldsymbol{u},\boldsymbol{v},\zeta} \mathcal{J}_{\mathrm{UOT}} = -\tau_a \mathrm{KL}\left(\boldsymbol{\pi}\mathbf{1}_N \| \boldsymbol{a}\right) - \langle \boldsymbol{u} + \zeta, \boldsymbol{\pi}\mathbf{1}_N \rangle - \tau_b \mathrm{KL}(\boldsymbol{\pi}^\top\mathbf{1}_M \| \boldsymbol{b}) - \langle \boldsymbol{v} - \zeta, \boldsymbol{\pi}^\top\mathbf{1}_M \rangle$$

$$= \tau_a \exp\left(-\frac{\zeta}{\tau_a}\right)\left\langle \boldsymbol{a}, \exp\left(-\frac{\boldsymbol{u}}{\tau_a}\right)\right\rangle + \tau_b \exp\left(\frac{\zeta}{\tau_b}\right)\left\langle \boldsymbol{b}, \exp\left(-\frac{\boldsymbol{v}}{\tau_b}\right)\right\rangle + \mathcal{O}_{\mathrm{Const}}$$

$$s.t. \ u_i + v_j \leq C_{ij}, \tag{47}$$

where $\mathcal{O}_{\text{Const}} = -\sum_{i=1}^{M} \tau_a a_i - \sum_{j=1}^{N} \tau_b b_j$, and we can neglect it during the following calculation. Once we obtain the optimal solution on $\boldsymbol{u}^*$, $\boldsymbol{v}^*$ and $\zeta^*$, the KL-Divergence will turn out to be constants and therefore the original optimization problem can be transformed into classic optimal transport. Finally we can obtain the optimal solution on $\zeta$ by considering $\frac{\partial \mathcal{J}_{\text{UOT}}}{\partial \zeta} = 0$ as below:

$$\zeta = \frac{\tau_a \tau_b}{\tau_a + \tau_b} \left[ \log \left\langle \boldsymbol{a}, \exp\left(-\frac{\boldsymbol{u}}{\tau_a}\right) \right\rangle - \log \left\langle \boldsymbol{b}, \exp\left(-\frac{\boldsymbol{v}}{\tau_b}\right) \right\rangle \right]. \tag{48}$$

Once we set $\tau_a \to \infty$ and $\tau_b \to \infty$, the results of the limitation will be shown as:

$$\lim_{\tau_a \to +\infty, \tau_b \to +\infty} a_i \exp\left(-\frac{u_i + \zeta}{\tau_a}\right) = \lim_{\tau_a \to +\infty, \tau_b \to +\infty} a_i \exp\left(-\frac{\zeta}{\tau_a}\right) = a_i \sqrt{\frac{\langle \boldsymbol{b}, \boldsymbol{1}_N \rangle}{\langle \boldsymbol{a}, \boldsymbol{1}_M \rangle}} = \sqrt{\omega_{\text{L}}} a_i,$$

$$\lim_{\tau_a \to +\infty, \tau_b \to +\infty} b_j \exp\left(-\frac{v_j - \zeta}{\tau_b}\right) = \lim_{\tau_a \to +\infty, \tau_b \to +\infty} b_j \exp\left(-\frac{\zeta}{\tau_b}\right) = b_j \sqrt{\frac{\langle \boldsymbol{a}, \boldsymbol{1}_M \rangle}{\langle \boldsymbol{b}, \boldsymbol{1}_N \rangle}} = \frac{1}{\sqrt{\omega_{\text{L}}}} b_j.$$
$$\tag{49}$$

Therefore we conclude the proof of the Proposition 3. $\qquad\square$

## F   Illustrations of Optimization 1

**Optimization 1.** (Calculation of ETM-Approx approach for UOT) To start with, we first review the Exact UOT Equation is defined as:

$$\min_{\boldsymbol{u},\zeta} L_{\text{U}} = \tau_a \sum_{i=1}^{M} a_i \exp\left(-\frac{u_i + \zeta}{\tau_a}\right) + \tau_b \exp\left(\frac{\zeta}{\tau_b}\right) \sum_{j=1}^{N} b_j \exp\left(-\frac{v_j}{\tau_b}\right)$$

$$= \tau_a \sum_{i=1}^{M} a_i \exp\left(-\frac{u_i + \zeta}{\tau_a}\right) + \tau_b \exp\left(\frac{\zeta}{\tau_b}\right) \sum_{j=1}^{N} b_j \exp\left(\frac{\sup_{k \in [M]} (u_k - C_{kj})}{\tau_b}\right), \tag{50}$$

where $v_j = -\sup_{k \in [M]} (u_k - C_{kj})$ meanwhile the marginal probabilities are set as $\boldsymbol{\pi} \boldsymbol{1}_N = \boldsymbol{a} \odot \exp\left(-(\boldsymbol{u} + \zeta)/\tau_a\right) = \boldsymbol{\alpha}$ and $\boldsymbol{\pi}^\top \boldsymbol{1}_M = \boldsymbol{b} \odot \exp\left(-(\boldsymbol{v} - \zeta)/\tau_b\right) = \boldsymbol{\beta}$. Since the optimization problem in Eq.(9) is convex, we can also utilize block gradient descent to optimize the problem. Specifically, we first fix $v^l$ and optimize variable $u^l$ at the $l$-th iteration by replacing the original marginal probability $\boldsymbol{b}$ in Eq.(6) with $\boldsymbol{\beta}$ accordingly to transform UOT into SemiUOT problem:

$$\min_{\boldsymbol{\pi} \geq 0} J_{\text{U}}^u = \langle \boldsymbol{C}, \boldsymbol{\pi} \rangle + \tau_a \text{KL} \left( \boldsymbol{\pi} \boldsymbol{1}_N \| \boldsymbol{a} \right),$$

$$s.t. \begin{cases} (\text{Constraint}): \ \boldsymbol{\pi}^\top \boldsymbol{1}_M = \boldsymbol{b} \odot \exp\left(-\frac{\boldsymbol{v} - \zeta}{\tau_b}\right) = \boldsymbol{\beta} \\ (\text{Optional}): \ \boldsymbol{\pi} \boldsymbol{1}_N = \boldsymbol{a} \odot \exp\left(-\frac{\boldsymbol{u} + \zeta}{\tau_a}\right) = \boldsymbol{\alpha} \end{cases}. \tag{51}$$

At that time, the Fenchel-Lagrange multipliers form of Eq.(51) is given via the Proposition 1:

$$\min_{\boldsymbol{u}} L_{\text{U}}^u = \tau_a \sum_{i=1}^{M} a_i \exp\left(-\frac{\widetilde{u}_i + \zeta}{\tau_a}\right) - \sum_{j=1}^{N} \beta_j (\widetilde{v}_j - \zeta)$$

$$= \tau_a \sum_{i=1}^{M} a_i \exp\left(-\frac{u_i + \zeta}{\tau_a}\right) - \sum_{j=1}^{N} \left( \inf_{k \in [M]} [C_{kj} - u_k] - \zeta \right) \beta_j. \tag{52}$$

Note that $\widetilde{\boldsymbol{u}}$ and $\widetilde{\boldsymbol{v}}$ denote the Lagrange multiplier for Eq.(51) while we have $\widetilde{v}_j = \inf_{k \in [M]} [C_{kj} - u_k] = v_j$ and $\widetilde{\boldsymbol{u}} = \boldsymbol{u}$. To further accelerate the optimization process, we consider to make a smooth approximation on replacing $\inf(\cdot)$ as $\inf_{k \in [M]} [C_{kj} - u_k] \approx -\epsilon \log[\sum_{k=1}^{M} e^{\frac{u_k - C_{kj}}{\epsilon}}] = \widehat{v}_j$. Therefore, we first fix $\widehat{v}^l$ and optimize variable $\widehat{u}^l$ at the $l$-th iteration to solve the following

equation on $\widehat{L}_U^u$ accordingly:

$$
\begin{aligned}
\min_{\widehat{\boldsymbol{u}}} \widehat{L}_U^u &= \tau_a \sum_{i=1}^{M} a_i \exp\left(-\frac{\widehat{u}_i + \zeta}{\tau_a}\right) + \sum_{j=1}^{N} \beta_j \left[\epsilon \log\left[\sum_{k=1}^{M} e^{\frac{\widehat{u}_k - C_{kj}}{\epsilon}}\right] + \zeta\right] \\
&= \tau_a \sum_{i=1}^{M} a_i \exp\left(-\frac{\widehat{u}_i + \zeta}{\tau_a}\right) + \sum_{j=1}^{N} b_j \exp\left(-\frac{\widehat{v}_j - \zeta}{\tau_b}\right) \left[\epsilon \log\left[\sum_{k=1}^{M} e^{\frac{\widehat{u}_k - C_{kj}}{\epsilon}}\right] + \zeta\right].
\end{aligned}
\tag{53}
$$

The optimization objective shares a similar formulation as Eq.31. At that time we adopt fixed-point iteration method to optimize $\widehat{u}$ accordingly based on the Proposition 2:

$$
\begin{cases}
\widehat{u}_1^{\ell+1} = \dfrac{\tau_a \epsilon}{\tau_a + \epsilon}\left[\log\left(a_1 e^{-\frac{\zeta}{\tau_a}}\right) - \log\left[\sum_{j=1}^{N}\left(\dfrac{\beta_j e^{-C_{1j}/\epsilon}}{\mathscr{W}_{\epsilon,j}(\widehat{\boldsymbol{u}}^{\ell})}\right)\right]\right] = \mathcal{U}_1\left(\widehat{u}_1^{\ell}, \cdots, \widehat{u}_M^{\ell}\right) \\
\vdots \\
\widehat{u}_s^{\ell+1} = \dfrac{\tau_a \epsilon}{\tau_a + \epsilon}\left[\log\left(a_s e^{-\frac{\zeta}{\tau_a}}\right) - \log\left[\sum_{j=1}^{N}\left(\dfrac{\beta_j e^{-C_{sj}/\epsilon}}{\mathscr{W}_{\epsilon,j}(\widehat{\boldsymbol{u}}^{\ell})}\right)\right]\right] = \mathcal{U}_s\left(\widehat{u}_1^{\ell}, \cdots, \widehat{u}_M^{\ell}\right) \\
\vdots \\
\widehat{u}_M^{\ell+1} = \dfrac{\tau_a \epsilon}{\tau_a + \epsilon}\left[\log\left(a_M e^{-\frac{\zeta}{\tau_a}}\right) - \log\left[\sum_{j=1}^{N}\left(\dfrac{\beta_j e^{-C_{Mj}/\epsilon}}{\mathscr{W}_{\epsilon,j}(\widehat{\boldsymbol{u}}^{\ell})}\right)\right]\right] = \mathcal{U}_M\left(\widehat{u}_1^{\ell}, \cdots, \widehat{u}_M^{\ell}\right)
\end{cases}
\tag{54}
$$

The iteration process can be shown to converge efficiently based on Proposition 2. After that we fix $\widehat{u}$ and optimize variable $\widehat{v}$ via $\widehat{v}_j = -\epsilon \log[\sum_{k=1}^{M} \exp((\widehat{u}_k - C_{kj})/\epsilon)]$. We can achieve the optimal solution on $\widehat{u}^*$ and $\widehat{v}^*$ via iteratively computing via the above procedure accordingly. Finally, we update $\zeta$ via $\zeta = (\tau_a \tau_b/(\tau_a + \tau_b))[\log(\sum_{i=1}^{M} a_i \exp(-\widehat{u}_i^*/\tau_a)) - \log(\sum_{j=1}^{N} b_j \exp(-\widehat{v}_j^*/\tau_b))]$.

## G  Algorithm for ETM-Based Method on UOT

We also provide the pseudo algorithm of the proposed ETM-Based approachs (e.g., ETM-Exact, ETM-Approx and ETM-Refine) for solving UOT in Alg.2 to make a more clear illustration.

## H  Proof of Proposition 4

**Proposition 4.** (The Definition and Usage of KKT-Multiplier Regularization) *Given any OT with multiplier $s$, one can obtain accurate solution $\boldsymbol{\pi}^*$ via proposed KKT-multiplier regularization term $\mathcal{G}(\boldsymbol{\pi}, \boldsymbol{s}) = \langle \boldsymbol{\pi}, \boldsymbol{s} \rangle$, which formulates Multiplier Regularized Optimal Transport (MROT):*

$$
\begin{aligned}
&\min_{\boldsymbol{\pi} \geq 0} \mathcal{J}_G = \langle \boldsymbol{C}, \boldsymbol{\pi} \rangle + \eta_G \langle \boldsymbol{\pi}, \boldsymbol{s} \rangle + \eta_{\text{Reg}} \mathcal{L}_{\text{Reg}}(\boldsymbol{\pi}) \\
&s.t. \ \boldsymbol{\pi} \mathbf{1}_N = \boldsymbol{\alpha}, \quad \boldsymbol{\pi}^\top \mathbf{1}_M = \boldsymbol{\beta},
\end{aligned}
\tag{55}
$$

*where $\mathcal{L}_{\text{Reg}}(\boldsymbol{\pi})$ denotes the regularization term on $\boldsymbol{\pi}$. $\boldsymbol{\alpha}$, $\boldsymbol{\beta}$ denote the final marginal probabilities obtained by ETM-based method, while $\eta_{\text{Reg}}$, $\eta_G$ denotes the hyper parameter. Ideally, $\eta_G$ should be set as a relatively large number. Meanwhile the dual form of MROT is given as:*

$$
\max_{\boldsymbol{\psi}, \boldsymbol{\phi}} L_G = \langle \boldsymbol{\alpha}, \boldsymbol{\psi} \rangle + \langle \boldsymbol{\beta}, \boldsymbol{\phi} \rangle - \eta_{\text{Reg}} \mathcal{L}_{\text{Reg}}^*\left(\frac{\psi_i + \phi_j - \widetilde{C}_{ij}}{\eta_{\text{Reg}}}\right),
\tag{56}
$$

*where $\widetilde{C}_{ij} = C_{ij} + \eta_G s_{ij}$, $\boldsymbol{\phi}$ and $\boldsymbol{\psi}$ denote the Lagrange multipliers for MROT. $\mathcal{L}_{\text{Reg}}^*(\cdot)$ denotes the conjugate function of $\mathcal{L}_{\text{Reg}}(\cdot)$ and one can figure out the matching results of $\boldsymbol{\pi}$ via solving the following equation $\nabla_{\pi_{ij}} \mathcal{L}_{\text{Reg}}(\pi_{ij}) = (\psi_i + \phi_j - \widetilde{C}_{ij})/\eta_{\text{Reg}}$.*

---

**Algorithm 2** The algorithm of ETM-Based method on UOT

---

**Input:** $C$: cost matrix; $\boldsymbol{a}, \boldsymbol{b}$: initial marginal probability; $\tau_a, \tau_b, \epsilon$: Hyper parameters.
Randomly initialize the value of $\boldsymbol{u}^{\text{init}}$.
Choose ETM-Exact, ETM-Approx or ETM-Refine on UOT for optimization.
**(1) Function:** ETM-Exact on UOT($C, \boldsymbol{a}, \boldsymbol{b}, \tau_a, \tau_b, \boldsymbol{u}^{t=0} = \boldsymbol{u}^{\text{init}}$)
Optimize $\boldsymbol{u}$ L-BFGS algorithm to optimize $L_{\text{U}}$ as:

$$\min_{\boldsymbol{u}} L_{\text{U}} = \tau_a \sum_{i=1}^{M} a_i \exp\left(-\frac{u_i + \zeta}{\tau_a}\right) + \tau_b \exp\left(\frac{\zeta}{\tau_b}\right) \sum_{j=1}^{N} b_j \exp\left(\frac{\sup_{k \in [M]} (u_k - C_{kj})}{\tau_b}\right)$$

Optimize $\boldsymbol{v}$ via $v_j = \inf_{k \in [M]} (C_{kj} - u_k)$.

Optimize $\zeta$ via $\zeta = \frac{\tau_a \tau_b}{\tau_a + \tau_b} \left[\log \left\langle \boldsymbol{a}, \exp\left(-\frac{\boldsymbol{u}}{\tau_a}\right)\right\rangle - \log \left\langle \boldsymbol{b}, \exp\left(-\frac{\boldsymbol{v}}{\tau_b}\right)\right\rangle\right]$ as shown in Eq.(48).
**Return:** The optimal solutions of $\boldsymbol{u}^*$, $\boldsymbol{v}^*$ and $\zeta^*$.
**(2) Function:** ETM-Approx on UOT($C, \boldsymbol{a}, \boldsymbol{b}, \tau_a, \tau_b, \widehat{\boldsymbol{u}}^{t=0} = \boldsymbol{u}^{\text{init}}$)
Randomly initialize the value of $\widehat{\boldsymbol{v}}^{t'=1}$.
**for** $t' = 1$ to $T'$ **do**
Optimize $\widehat{\boldsymbol{u}}^{t'}$ via Proposition 2 to optimize $\widehat{L}_{\text{U}}^u$ as:

$$\min_{\widehat{\boldsymbol{u}}} \widehat{L}_{\text{U}}^u = \tau_a \sum_{i=1}^{M} a_i \exp\left(-\frac{\widehat{u}_i + \zeta}{\tau_a}\right) + \sum_{j=1}^{N} b_j \exp\left(-\frac{\widehat{v}_j - \zeta}{\tau_b}\right) \left[\epsilon \log \left[\sum_{k=1}^{M} e^{\frac{\widehat{u}_k - C_{kj}}{\epsilon}}\right] + \zeta\right]$$

Optimize $\widehat{\boldsymbol{v}}^{t'}$ via $\widehat{v}_j^{t'} = -\epsilon \log[\sum_{k=1}^{M} \exp((\widehat{u}_k^{t'} - C_{kj})/\epsilon)]$.
**end for**
Optimize $\zeta$ via $\zeta = \frac{\tau_a \tau_b}{\tau_a + \tau_b} \left[\log \left\langle \boldsymbol{a}, \exp\left(-\frac{\widehat{\boldsymbol{u}}}{\tau_a}\right)\right\rangle - \log \left\langle \boldsymbol{b}, \exp\left(-\frac{\widehat{\boldsymbol{v}}}{\tau_b}\right)\right\rangle\right]$ as shown in Eq.(48).
**Return:** The optimal solutions of $\widehat{\boldsymbol{u}}^*$, $\widehat{\boldsymbol{v}}^*$ and $\zeta^*$.
**(3) Function:** ETM-Refine on UOT($C, \boldsymbol{a}, \boldsymbol{b}, \tau_a, \tau_b, \widehat{\boldsymbol{u}}^{t=0} = \boldsymbol{u}^{\text{init}}$)
Obtain $\widehat{\boldsymbol{u}}^* = $ ETM-Approx on UOT($C, \boldsymbol{a}, \boldsymbol{b}, \tau_a, \tau_b, \widehat{\boldsymbol{u}}^{t=0} = \boldsymbol{u}^{\text{init}}$).
Obtain $\boldsymbol{u}^* = $ ETM-Exact on UOT($C, \boldsymbol{a}, \boldsymbol{b}, \tau_a, \tau_b, \boldsymbol{u}^{t=0} = \widehat{\boldsymbol{u}}^*$).
**Return:** The optimal solutions of $\boldsymbol{u}^*$, $\boldsymbol{v}^*$ and $\zeta^*$.

---

*Proof.* We first provide the Lagrange multiplier of MROT as:

$$\max_{\boldsymbol{\psi}, \boldsymbol{\phi}} \min_{\boldsymbol{\pi} \geq 0} \mathcal{J}_{\text{MROT}} = \langle C, \boldsymbol{\pi} \rangle + \eta_G \langle \boldsymbol{\pi}, \boldsymbol{s} \rangle + \eta_{\text{Reg}} \mathcal{L}_{\text{Reg}}(\boldsymbol{\pi}) - \langle \boldsymbol{\psi}, \boldsymbol{\pi} \mathbf{1}_N - \boldsymbol{\alpha} \rangle - \langle \boldsymbol{\phi}, \boldsymbol{\pi}^\top \mathbf{1}_M - \boldsymbol{\beta} \rangle$$

$$= \langle \boldsymbol{\alpha}, \boldsymbol{\psi} \rangle + \langle \boldsymbol{\beta}, \boldsymbol{\phi} \rangle + \eta_{\text{Reg}} \inf_{\boldsymbol{\pi}} \left[\sum_{i,j} \left[\frac{C_{ij} + \eta_G s_{ij} - \psi_i - \phi_j}{\eta_{\text{Reg}}} \pi_{ij} + \mathcal{L}_{\text{Reg}}(\pi_{ij})\right]\right]$$

$$= \langle \boldsymbol{\alpha}, \boldsymbol{\psi} \rangle + \langle \boldsymbol{\beta}, \boldsymbol{\phi} \rangle - \eta_{\text{Reg}} \sup_{\boldsymbol{\pi}} \left[\sum_{i,j} \left[\frac{\psi_i + \phi_j - \widetilde{C}_{ij}}{\eta_{\text{Reg}}} \pi_{ij} - \mathcal{L}_{\text{Reg}}(\pi_{ij})\right]\right]$$

$$= \langle \boldsymbol{\alpha}, \boldsymbol{\psi} \rangle + \langle \boldsymbol{\beta}, \boldsymbol{\phi} \rangle - \eta_{\text{Reg}} \mathcal{L}_{\text{Reg}}^* \left(\frac{\psi_i + \phi_j - \widetilde{C}_{ij}}{\eta_{\text{Reg}}}\right).$$

$$(57)$$

At that time we have the following results:

$$\begin{cases} \frac{\partial \mathcal{J}_{\text{MROT}}}{\partial \psi_i} = 0 \\ \frac{\partial \mathcal{J}_{\text{MROT}}}{\partial \phi_j} = 0 \end{cases} \Rightarrow \begin{cases} \nabla_{\psi_i} \mathcal{L}_{\text{Reg}}^* \left(\frac{\psi_i + \phi_j - \widetilde{C}_{ij}}{\eta_{\text{Reg}}}\right) = \alpha_i \\ \nabla_{\phi_j} \mathcal{L}_{\text{Reg}}^* \left(\frac{\psi_i + \phi_j - \widetilde{C}_{ij}}{\eta_{\text{Reg}}}\right) = \beta_j \end{cases}. \quad (58)$$

By taking the differentiation on $\pi_{ij}$ we have:

$$\frac{\partial \mathcal{J}_{\text{MROT}}}{\partial \pi_{ij}} = \widetilde{C}_{ij} + \eta_{\text{Reg}} \nabla_{\pi_{ij}} \mathcal{L}_{\text{Reg}}(\pi_{ij}) - \psi_i - \phi_j = 0. \quad (59)$$

For instance, when $\mathcal{L}_{\text{Reg}}(\boldsymbol{\pi}) = -\langle \boldsymbol{\pi}, \log(\boldsymbol{\pi}) - 1 \rangle$ denotes as the entropy regularization term, the dual form of MROT-Ent is shown as:

$$
\begin{cases}
\max_{\boldsymbol{\psi}, \boldsymbol{\phi}} \mathcal{J}_{\text{MROT−Ent}} = \langle \boldsymbol{\alpha}, \boldsymbol{\psi} \rangle + \langle \boldsymbol{\beta}, \boldsymbol{\phi} \rangle - \eta_{\text{Reg}} \sum_{i,j} \exp\left( \dfrac{\psi_i + \phi_j - \widetilde{C}_{ij}}{\eta_{\text{Reg}}} \right) \\
\pi_{ij} = \exp\left( \dfrac{\psi_i + \phi_j - \widetilde{C}_{ij}}{\eta_{\text{Reg}}} \right)
\end{cases}.
\tag{60}
$$

When $\mathcal{L}_{\text{Reg}}(\boldsymbol{\pi}) = \langle \boldsymbol{\pi}, \boldsymbol{\pi} \rangle / 2$ denotes the square-norm regularization term, the dual form of MROT-Norm is shown as:

$$
\begin{cases}
\max_{\boldsymbol{\psi}, \boldsymbol{\phi}} \mathcal{J}_{\text{MROT−Norm}} = \langle \boldsymbol{\alpha}, \boldsymbol{\psi} \rangle + \langle \boldsymbol{\beta}, \boldsymbol{\phi} \rangle - \dfrac{\eta_{\text{Reg}}}{2} \sum_{i,j} \left[ \dfrac{\psi_i + \phi_j - \widetilde{C}_{ij}}{\eta_{\text{Reg}}} \right]_+^2 \\
\pi_{ij} = \left[ \dfrac{\psi_i + \phi_j - \widetilde{C}_{ij}}{\eta_{\text{Reg}}} \right]_+
\end{cases}.
\tag{61}
$$

Therefore we conclude the proof of Proposition 4. $\qquad\square$

**Extensions.** MROT can be even extended to solve classic optimal transport problem. That is, the classic optimal transport problem and its dual form can be represented as below:

$$
\begin{aligned}
&J = \arg\min_{\boldsymbol{\pi} \geq 0} \langle \boldsymbol{\pi}, \boldsymbol{C} \rangle &\quad& \max_{\boldsymbol{f}^\triangle, \boldsymbol{g}^\triangle, \boldsymbol{s}} \mathcal{J}_{\text{OT}} = \left\langle \boldsymbol{f}^\triangle, \boldsymbol{\alpha} \right\rangle + \left\langle \boldsymbol{g}^\triangle, \boldsymbol{\beta} \right\rangle \\
&s.t.\ \boldsymbol{\pi} \mathbf{1}_N = \boldsymbol{\alpha}, \quad \boldsymbol{\pi}^\top \mathbf{1}_M = \boldsymbol{\beta} &\Leftrightarrow& \quad s.t.\ \begin{cases} f_i^\triangle + g_j^\triangle + s_{ij} = C_{ij}, \quad s_{ij} \geq 0 \\ g_j^\triangle = \inf_{k \in [M]} \left( C_{kj} - f_k^\triangle \right) \end{cases}
\end{aligned}
\tag{62}
$$

where $\boldsymbol{f}^\triangle$, $\boldsymbol{g}^\triangle$, and $\boldsymbol{s}$ represent the dual variables. To solve the dual form of the classic OT problem, unconstrained optimization techniques, such as L-BFGS or the Sinkhorn algorithm, can be employed to optimize for $\boldsymbol{s}$. Then one can further adopts MROT to solve $\boldsymbol{\pi}$ for classic optimal transport.

In summary, the time complexity of the proposed ETM-Approx+MROT-Ent or ETM-Approx+MROT-Norm method is provided as $\mathcal{O}\left( NM \log\left( 1/\varepsilon_{\text{err}} \right) + NM d_\pi \right)$ where $d_\pi$ denotes the number of iterations on MROT. Meanwhile, the time complexity of the proposed ETM-Refine+MROT-Ent or ETM-Refine+MROT-Norm method is provided as $\mathcal{O}\left( NM \log\left( 1/\varepsilon_{\text{err}} \right) + NM(\log M) d_T + NM d_\pi \right)$ where $d_T$ denotes the number of iterations on ETM-Refine.

# I   Experiments on Domain Adaptations

**Datasets.** We conduct the unsupervised domain adaptation tasks on *Digits*, *Office-Home*, and *VisDA*. *Digits* is the classic dataset for digit classification which contains three standard digit classification datasets: **MNIST** [53], **USPS**[45] and **SVHN** [76]. Each dataset consists of 10 classes of digits, ranging from 0 to 9. *Office-Home* [102] is a standard benchmark dataset which includes 15,500 images in 65 object classes in office and home settings, forming four dissimilar domains: Artistic images (**Ar**), Clip Art (**Cl**), Product images (**Pr**), and Real-World (**Rw**). *VisDA* [81] is a large-scale computer vision dataset on two domains, i.e., **Synthetic** and **Real** with 280K images in 12 classes.

**Performance.** We also conduct the UDA domain adaptation tasks on Digits and VisDA and the results are shown in Table.4. We can observe that the proposed ETM-Refine with MROT-Norm on SemiUOT achieves state-of-the-art performance on Digits and VisDA.

# J   Experiments on Partial Domain Adaptations

**Datasets.** We further conduct the domain adaptation tasks on new datasets, i.e., *Office-31* [88] and *ImageCLEF* [13]. **Office-31** is the commonly-used computer vision dataset for domain adaptation

Table 4: Classification accuracy (%) on Digits (Source: LeNet) and VisDA dataset (Source:ResNet50) for UDA (unsupervised domain adaptation) task

| Method | S→M | M→U | U→M | Avg | VisDA |
|---|---|---|---|---|---|
| Source | 68.3±0.3 | 65.3±0.5 | 66.2±0.2 | 66.6 | 52.4 |
| DeepJDOT [25] | 95.4±0.1 | 95.6±0.4 | 96.4±0.3 | 95.8 | 68.0 |
| JUMBOT [34] | 98.9±0.1 | 96.7±0.5 | 98.2±0.1 | 97.9 | 72.5 |
| JUMBOT + UOT(ETM-Refine + MROT-Ent) | 99.4±0.1 | 98.7±0.3 | 99.2±0.1 | 99.1 | 73.6 |
| JUMBOT + UOT(ETM-Refine + MROT-Norm) | 99.7±0.1 | 99.3±0.2 | 99.6±0.1 | **99.5** | **74.2** |

Table 5: H-score (%) on *Office-Home* for universal unsupervised domain adaptation

| Method | Ar→Cl | Ar→Pr | Ar→Rw | Cl→Ar | Cl→Pr | Cl→Rw | Pr→Ar | Pr→Cl | Pr→Rw | Rw→Ar | Rw→Cl | Rw→Pr | Avg |
|---|---|---|---|---|---|---|---|---|---|---|---|---|---|
| ResNet [43] | 44.65 | 48.04 | 50.13 | 46.64 | 46.91 | 48.96 | 47.47 | 43.17 | 50.23 | 48.45 | 44.76 | 48.43 | 47.32 |
| OSBP [89] | 39.59 | 45.09 | 46.17 | 45.70 | 45.24 | 46.75 | 45.26 | 40.54 | 45.75 | 45.08 | 41.64 | 46.90 | 44.48 |
| UAN [108] | 51.64 | 51.70 | 54.30 | 61.74 | 57.63 | 61.86 | 50.38 | 47.62 | 61.46 | 62.87 | 52.61 | 65.19 | 56.58 |
| CMU [38] | 56.02 | 56.93 | 59.15 | 66.95 | 64.27 | 67.82 | 54.72 | 51.09 | 66.39 | 68.24 | 57.89 | 69.73 | 61.60 |
| DCC [54] | 57.97 | 54.05 | 58.01 | 74.64 | 70.62 | 77.52 | 64.34 | 73.60 | 74.94 | 80.96 | 75.12 | 80.38 | 70.18 |
| TNT [18] | 61.90 | 74.60 | 80.20 | 73.50 | 71.40 | 79.60 | 74.20 | 69.50 | 82.70 | 77.30 | 70.10 | 81.20 | 74.70 |
| UniOT [16] | 67.27 | 80.54 | 86.03 | 73.51 | 77.33 | 84.28 | 75.54 | 63.33 | 85.99 | 77.77 | 65.37 | 81.92 | 76.57 |
| UniOT + UOT(ETM-Refine + MROT-Ent) | 68.63 | 81.72 | 87.94 | 75.88 | 79.03 | 86.21 | 77.29 | 68.77 | 87.14 | 78.59 | 73.62 | 82.83 | 78.97 |
| UniOT + UOT(ETM-Refine + MROT-Norm) | **69.02** | **81.95** | **88.36** | **76.12** | **79.36** | **86.49** | **77.03** | 69.25 | **87.30** | **78.93** | 74.18 | **82.96** | **79.25** |

with 4,652 images from three different domains: *Amazon* (**A**), *Webcam* (**W**) and *DSLR* (**D**). Target domain has the first 10 classes (alphabetical order) following [11]. **ImageCLEF** contains 3 domains with 12 classes, i.e., *Caltech* (**C**), *ImageNet* (**I**) and *Pascal* (**P**). Target domain has the first 6 classes (alphabetical order) following [66].

**Baselines.** We involve **DeepJDOT** [25], **ROT** [5], **JUMBOT** [34], **ETN** [12], **AR** [42], **m-POT** [77], **MOT** [65], as the model baselines for the domain adaptation task. (1) **DeepJDOT** [25] first adopts optimal transport into solving domain adaptation problem with deep learning framework. (2) **ROT** [5] adopts robust optimal transport into adversarial training for domain adaptation. (3) **JUMBOT** [34] adopts mini-batch unbalanced optimal transport method for domain adaptation. (4) **ETN** [12] utilizes example transfer network to jointly learn domain-invariant representations and the progressive weighting scheme. (5) **AR** [42] adopts adversarial reweighting strategy on source domain data for alignment. (6) **m-POT** [77] adopts partial optimal transport method in the mini-batch settings for domain adaptation. (7) **MOT** [65] adopts masked unbalanced optimal transport technique on considering label information for PDA tasks.

# K   Experiments on Universal Domain Adaptations

We further conduct the experiments on universal domain adaptations. That is, there are shared labels between the source and target domains. Additionally, there are private labels specific to each domain [33, 111]. We conduct the universal domain adaptations on both *Office-31* and *Office-Home*. Specifically, we set the first 10 classes in alphabetical order as the common label set, the next 10 classes as source private label and the rest 11 classes as target private label for Office-31. Likewise, we set the first 10 classes in alphabetical order as the common label set, the next 5 classes as source private label and the rest 55 classes as target private label for Office-Home. We involve the following models as baselines: (1) **OSBP** [89] adopts domain adversarial learning for open-set domain adaptation, (2) **UAN** [108] utilizes transferability criterion for universal domain adaptation, (3) **CMU** [38] learns to detect open classes with uncertainty estimation, (4) **DCC** [54] adopts domain consensus clustering for adaptation, (5) **TNT** [18] adopts evidential neighborhood contrastive learning for adaptation, (6) **UniOT** [16] adopts unbalanced optimal transport with adaptive filtering for transferring.

We adopt the same experimental settings as UniOT [16]. We utilize the commonly-used H-score [38] to validate the final results as shown in Table 5-6. Note that UniOT + UOT(ETM + MROT) only replaces the entropic UOT in UniOT with our proposed ETM-Refine method with MROT. From that, we can observe that UniOT + UOT(ETM-Refine + MROT-Norm) reaches the best performance, indicating that UOT with ETM + MROT can provide more accurate matching results.

# L   Experiments on Treatment Effect Estimation

**Datasets for Treatment Effect Estimation.** We further conduct ETM-Refine on treatment effect estimation with two semi-synthetic datasets IHDP [97] and ACIC [107]. IHDP is set to estimate the effect of specialist home visits on infants' potential cognitive scores and it contains 747 observations and 25 covariates. ACIC includes 4802 observations and 58 covariates, which comes from the collaborative perinatal project.

Table 6: H-Score (%) on *Office-31* for universal unsupervised domain adaptation

| Method | A→D | A→W | D→A | D→W | W→A | W→D | Avg |
|---|---|---|---|---|---|---|---|
| ResNet [43] | 49.78 | 47.92 | 48.48 | 54.94 | 48.96 | 55.60 | 50.94 |
| OSBP [89] | 51.14 | 50.23 | 49.75 | 55.53 | 50.16 | 57.20 | 52.34 |
| UAN [108] | 59.68 | 58.61 | 60.11 | 70.62 | 60.34 | 71.42 | 63.46 |
| CMU [38] | 68.11 | 67.33 | 71.42 | 79.32 | 72.23 | 80.42 | 73.14 |
| DCC [54] | 88.50 | 78.54 | 70.18 | 79.29 | 75.87 | 88.58 | 80.16 |
| TNT [18] | 85.70 | 80.40 | 83.80 | 92.00 | 79.10 | 91.20 | 85.37 |
| UniOT [16] | 86.97 | 88.48 | 88.35 | 98.83 | 87.60 | 96.57 | 91.13 |
| UniOT + UOT(ETM-Refine + MROT-Ent) | 88.25 | 89.62 | 89.47 | 99.48 | 89.10 | 97.94 | 92.31 |
| UniOT + UOT(ETM-Refine + MROT-Norm) | **88.67** | **90.14** | **90.03** | **99.58** | **89.42** | **98.46** | **92.72** |

Table 7: Experimental results on Treatment Effect Estimation tasks.

| | ACIC (PEHE) | | ACIC (AUUC) | | IHDP (PEHE) | | IHDP (AUUC) | |
|---|---|---|---|---|---|---|---|---|
| | In-Sample | Out-Sample | In-Sample | Out-Sample | In-Sample | Out-Sample | In-Sample | Out-Sample |
| OLS [2] | 3.749 | 4.340 | 0.843 | 0.496 | 3.856 | 5.674 | 0.652 | 0.492 |
| TARNet [97] | 3.236 | 3.254 | **0.886** | 0.662 | 0.749 | 1.788 | 0.654 | 0.711 |
| PSM [87] | 5.228 | 5.094 | 0.884 | 0.745 | 3.219 | 4.634 | 0.740 | 0.681 |
| CFR-WASS [97] | 3.128 | 3.207 | 0.873 | 0.669 | 0.657 | 1.704 | 0.656 | 0.715 |
| ESCFR [104] | 2.252 | 2.316 | 0.796 | 0.754 | 0.502 | 1.282 | 0.665 | 0.719 |
| ESCFR + UOT(ETM-Refine + MROT-Ent) | 2.327 | 2.261 | 0.839 | 0.814 | 0.497 | 1.275 | 0.769 | 0.763 |
| ESCFR + UOT(ETM-Refine + MROT-Norm) | **2.104** | **2.216** | 0.883 | **0.839** | **0.475** | **1.146** | **0.798** | **0.802** |

**Results.** We involve the following models as baselines: (1) **OLS** [2] utilizes least square regression with treatment as covariates, (2) **TARNet** [97] adopts integral orobability metrics for adaptation, (3) **PSM** [87] adopts propensity score for causal effects, (4) **CFR-WASS** [87] utilizes standard optimal transport for adaptation, (5) **ESCFR** [104] further utilizes unbalanced optimal transport for adaptation. We adopt the same experimental settings as ESCFR [104]. We utilize Precision in Estimation of Heterogeneous Effect (PEHE) [97] and Area Under the Uplift Curve (AUUC) [7] for the evaluation. Note that ESCFR + UOT(ETM-Refine + MROT-Ent) only replaces the entropic UOT in ESCFR with our proposed approximate-to-exact ETM-Refine + MROT-Norm. The experimental results are shown in Table 7. From that, we can observe that ESCFR + UOT(ETM-Refine + MROT-Norm) achieves the best performance, indicating the efficacy of our proposed ETM-Refine method.

## M   More Experimental Results

**Parameter sensitivity.** We tune $\eta_G$ on SemiUOT via ETM-Refine with MROT-Norm in range of $\eta_G \in \{0, 1, 100\}$ using the same data samples shown in Fig.1 and show the results in Fig.5. We can observe that when $\eta_G$ is smaller (e.g., $\eta_G = 0$ or $\eta_G = 1$), the proposed KKT-multiplier regularization term $\mathcal{G}(\pi, s) = \langle \pi, s \rangle$ may struggle to play a significant role during the optimization process. Meanwhile when $\eta_G = 100$, ETM-Refine with MROT-Norm can achieve more accurate matching results. We can conclude that choosing a larger value of $\eta_G$ can fully utilize the knowledge provided by KKT multiplier and enhance the final results. Moreover, we conduct the experiments for the absolute error when $\tau = 1$ with $N = 500$ synthetic data samples on both SemiUOT and UOT and report the results in Fig.6(a)-(b). Larger value on $\eta_G$ can provide more useful KKT-multiplier information and boost the model performance and therefore we set $\eta_G = 100$ empirically. Furthermore, we conduct the hyper parameter experiments by varying $\epsilon = \{0.01, 0.05, 0.1, 0.5, 1\}$ on UDA task in Office-Home and report the results in Fig.6(c). We can observe that smaller value of $\epsilon$ can provide a more accurate approximation with higher accuracy and thus we set $\epsilon = 0.01$.

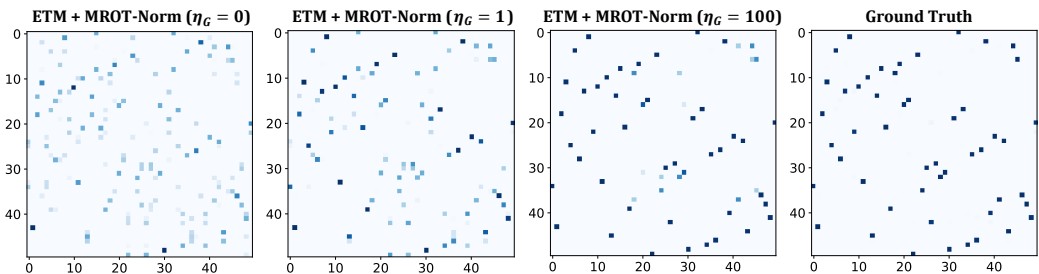

Figure 5: The matching results on ETM + MROT-Norm on SemiUOT with different values of $\eta_G = \{0, 1, 100\}$.

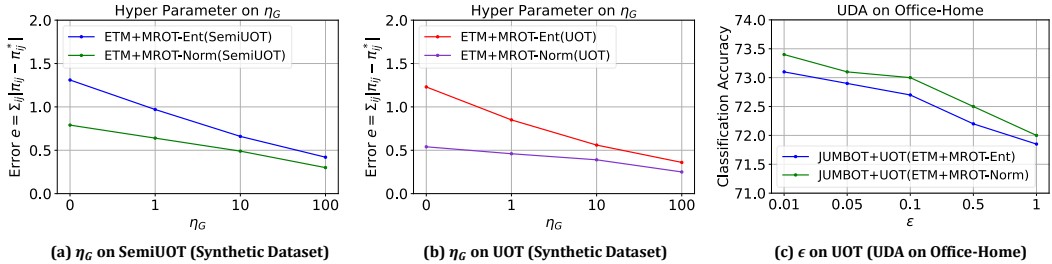

(a) $\eta_G$ on SemiUOT (Synthetic Dataset)  (b) $\eta_G$ on UOT (Synthetic Dataset)  (c) $\epsilon$ on UOT (UDA on Office-Home)

Figure 6: The hyper parameters on $\eta_G$ and $\epsilon$.

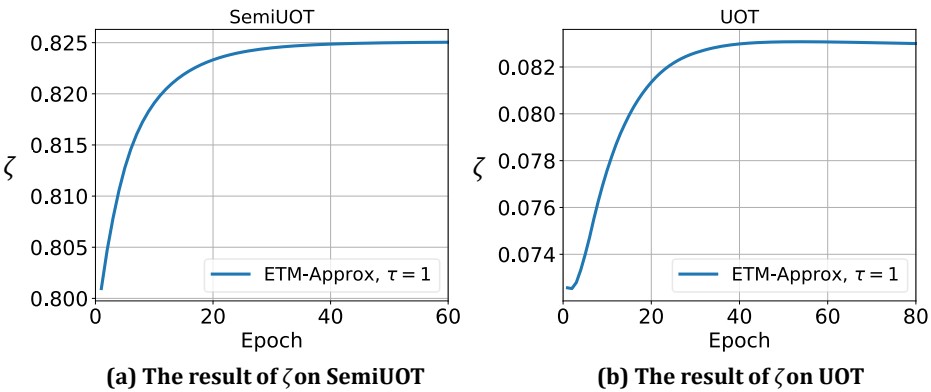

(a) The result of $\zeta$ on SemiUOT  (b) The result of $\zeta$ on UOT

Figure 7: The results of $\zeta$ and $L_{\mathrm{P}}$ on UOT and SemiUOT.

## N  Miscellaneous Discussions

### N.1  The role of $\zeta$ in ETM-based method

We first discuss why we should involve translation invariant $\zeta$ in both $L_{\mathrm{U}}$ and $L_{\mathrm{P}}$. Specifically, we first analyze the case of SemiUOT. The Fenchel-Lagrange conjugate form of SemiUOT without translation invariant mechanism is given as:

$$\min_{\boldsymbol{f},\boldsymbol{g},\zeta}\left[\tau\sum_{i=1}^{M}a_i\exp\left(-\frac{f_i}{\tau}\right)-\sum_{j=1}^{N}b_jg_j\right] \tag{63}$$
$$s.t.\ f_i+g_j\le C_{ij}.$$

We can adopt $c$-transform on Eq.(63) to obtain the unconstrained optimization problem as:

$$\min_{\boldsymbol{f}}\widetilde{L}_{\mathrm{P}}=\tau\sum_{i=1}^{M}a_i\exp\left(-\frac{f_i}{\tau}\right)-\sum_{j=1}^{N}\inf_{k\in[M]}[C_{kj}-f_k]b_j, \tag{64}$$

We adopt L-BFGS to optimize $\widetilde{L}_{\mathrm{P}}$ using the same data samples as shown in Fig.1 with $\tau=1$. Meanwhile, the translation invariant term $\zeta$ in SemiUOT should be calculated as follows:

$$\zeta=\tau\log\left(\sum_{i=1}^{M}a_i\exp\left(-\frac{f_i}{\tau}\right)\right)-\tau\log\left(\sum_{j=1}^{N}b_j\right). \tag{65}$$

Ideally, $\zeta$ should equals to 0 since $\sum_{i=1}^{M}a_i\exp\left(-\frac{f_i}{\tau}\right)=\sum_{j=1}^{N}b_j$. However, we can observe that $\zeta>0$ during the iteration epoch on optimizing $\widetilde{L}_{\mathrm{P}}$ as shown in Fig.7(a). Therefore we can conclude that $\zeta$ is indispensable during the calculation on SemiUOT. Likewise, the Fenchel-Lagrange conjugate form of UOT without translation invariant mechanism is given as:

$$\min_{\boldsymbol{v},\boldsymbol{u}}\left[\tau_a\Big\langle\boldsymbol{a},\exp\left(-\frac{\boldsymbol{u}}{\tau_a}\right)\Big\rangle+\tau_b\Big\langle\boldsymbol{b},\exp\left(-\frac{\boldsymbol{v}}{\tau_b}\right)\Big\rangle\right]\quad s.t.\ u_i+v_j\le C_{ij}. \tag{66}$$

Here we can adopt $c$-transform on Eq.(66) to obtain the unconstrained optimization problem as:

$$\min_{\boldsymbol{u}} \widetilde{L}_{\mathrm{U}} = \tau_a \sum_{i=1}^{M} a_i \exp\left(-\frac{u_i}{\tau_a}\right) + \tau_b \sum_{j=1}^{N} b_j \exp\left(\frac{\sup_{k=1}^{M}(u_k - C_{kj})}{\tau_b}\right). \tag{67}$$

We also adopt L-BFGS to optimize $\widetilde{L}_{\mathrm{U}}$ using the same data samples as shown in Fig.2 with $\tau_a = \tau_b = 1$. Meanwhile, the translation invariant term $\zeta$ in UOT should be calculated as follows:

$$\zeta = \frac{\tau_a \tau_b}{\tau_a + \tau_b} \left[\log\left\langle \boldsymbol{a}, \exp\left(-\frac{\boldsymbol{u}}{\tau_a}\right)\right\rangle - \log\left\langle \boldsymbol{b}, \exp\left(-\frac{\boldsymbol{v}}{\tau_b}\right)\right\rangle\right]. \tag{68}$$

Ideally, $\zeta$ should equals to 0 since $\left\langle \boldsymbol{a}, \exp\left(-\frac{\boldsymbol{u}}{\tau_a}\right)\right\rangle = \left\langle \boldsymbol{b}, \exp\left(-\frac{\boldsymbol{v}}{\tau_b}\right)\right\rangle$. However, we can observe that $\zeta > 0$ during the iteration epoch on optimizing $\widetilde{L}_{\mathrm{U}}$ as shown in Fig.7(b). Therefore we can conclude that $\zeta$ is indispensable during the calculation on UOT. In conclusion, the concept of translation invariant was first proposed in [96]. However, [96] only utilizes translation invariant for entropic UOT. **We highlight that, in this paper, we further extend translation invariant for standard UOT/SemiUOT scenario**. We illustrate that translation invariant is essential in solving UOT and SemiUOT problems.

