# OpenReview forum: "Solving Discrete (Semi) Unbalanced Optimal Transport with Equivalent Transformation Mechanism and KKT-Multiplier Regularization"
_NeurIPS.cc/2025/Conference — NeurIPS 2025 poster_

### Official Review · Reviewer_ETJd · 2025-06-02

**Clarity:** 3
**Significance:** 3
**Originality:** 3
**Rating:** 5
**Confidence:** 3

**Summary:**

This paper tackles the problem of efficiently obtaining exact solutions for the Unbalanced Optimal Transport and Semi-Unbalanced Optimal Transport problems. The proposed method, called the Equivalent Transformation Mechanism (ETM), enables the determination of marginal probabilities on which a classical Optimal Transport optimization can be performed.

**Questions:**

Questions:

1) In the well-known article "Free boundaries in optimal transport and Monge-Ampère obstacle problems" by L. Caffarelli and R. J. McCann, an unbalanced formulation of optimal transport is transformed into a classical optimal transport problem. Is there any connection between that approach and the one taken in the article under review?

2) The classical unbalanced problem typically includes constraints such as $\pi 1_N \leq a$ and $\pi^T 1_M \leq b$. These are not included in the formulation of the UOT problem in the paper. Is there a particular reason for omitting them?

3) Proposition 1 is the first key result. However, the notation is not fully clear at that point in the article. Could the authors clarify the use of the star symbol $\ast$ in expressions like $f^\ast$ and $\zeta^\ast$? (A similar comment applies to Proposition 3.)

4) Regarding the statement of Proposition 1: I suggest rephrasing it as, "Given the SemiUOT functional with KL divergence $J_{\text{SemiUOT}}$..." A similar revision could be applied to Proposition 3.

5) Do the new marginals $\alpha$ and $\beta$ represent vectors encoding the weights of discrete probability measures? That is, do their coefficients add to 1?

6) I did not understand the first sentence on line 180. Could the authors clarify it?

Suggestions:

1) Corollary 1 is not formally stated, so I would suggest presenting it as a Remark instead.

2) While not necessary, a brief discussion of the advantages and disadvantages of SemiUOT compared to UOT would enrich the paper.

3) In the Related Works section, I suggest including a citation of the paper [Y. Bai et al., "Linear Optimal Partial Transport Embedding," ICML, 2023], as it presents an alternative approach to addressing the computational complexity of unbalanced optimal transport problems, without relying on the addition of regularization terms.

4) I suggest referencing Algorithms 1 and 2 explicitly in the main text, even if only briefly, to guide the reader.

Typos and Minor Comments:

1) Line 44: After "i.e.," I suggest using a lowercase letter and not capitalizing the following word.

2) Lines 179, 180, and the paragraph between lines 193 and 199: Add commas after "Hence," "Then," and "After that," respectively.

**Ethical Concerns:**

["NO or VERY MINOR ethics concerns only"]

**Final Justification:**

I thank all the authors for your answers, clarifications and comments. I decide to keep my score (Accept).

**Limitations:**

yes

**Paper Formatting Concerns:**

I haven't noticed any major formatting concerns.

**Quality:**

3

**Strengths And Weaknesses:**

Strengths:
- The article addresses a fundamental problem in the optimal transport community. Rather than relying on popular regularization terms in the UOT and Semi-UOT problems—which allow for efficient computations but yield inexact and dense solutions—the authors propose a novel idea: transforming these problems into classical optimal transport problems by computing the exact marginal probability distributions that participate in the transport process. Moreover, for the Semi-UOT version, the authors propose a way to smooth out the $L_p$ functional, highlighting its usefulness to get a good initialization for the main method of interest. In addition, the proposed ETM approach presented in Section 4.1 allows for the computation of a so-called multiplier "s" (Corollary 2), which is used to formulate a new regularized version of the optimal transport problem (Section 4.2).
- Experiments were conducted with synthetic and real-world datsets.

Weaknesses:
The algorithms are provided in the appendix, but the methodology is somewhat convoluted to follow in the main text, even though the authors include all the necessary components. I understand that space limitations can make it difficult to fully detail the steps that a reader would ultimately need to follow.

---

> ### Author Rebuttal · Authors · 2025-07-28
>
> - W1: The algorithms are provided in the appendix which should have appeared in the main text.
>
> A1: We will add the algorithm 1-2 into the main text in the camera-ready version.
>
> - Q1: "Free boundaries in optimal transport and Monge-Ampère obstacle problems" (Paper A) by L. Caffarelli and R. J. McCann, an unbalanced formulation of optimal transport is transformed into a classical optimal transport problem. Is there any connection between that approach and the one taken in the article under review?
>
> QA1: We read through Paper A above and we notice that there are several differences between Paper A and ours: (1) Paper A focuses on the free-boundary optimal transport problem where only a fraction of the mass is transported. Indeed, the free-boundary optimal transport problem (i.e., does not involve KL-divergence) is different from SemiUOT and UOT as we investigate in our paper; (2) Paper A adopts the Monge-Ampère equation to solve the problem in the continuous space, meanwhile our paper focuses on solving SemiUOT and UOT in the discrete space with newly proposed ETM and MROT method.
>
> - Q2: The classical unbalanced problem typically includes marginal constraints. These are not included in the formulation of the UOT problem in the paper. Is there a particular reason for omitting them?
>
> QA2: Let us kindly make some clarification here. Classical unbalanced problem was first introduced in [1], and it does not include marginal constraints such as $\pi1_{N} \le a$ and $\pi^T1_{M}\le b$. Meanwhile optimal transport with $\pi1_{N} \le a$ and $\pi^T1_{M}\le b$ constraints is defined as partial optimal transport which was first investigated in [2] as shown:
>
> $$
> \min_{\pi} J = \langle {C}, \pi \rangle  \quad \quad
> s.t.   \pi1_{N} \le a, \pi^T1_{M}\le b, 1_{M}^{T}\pi1_{N} = S
> $$
>
> [2] has transformed partial optimal transport into a classical optimal transport problem. However, we want to highlight that SemiUOT/UOT and partial optimal transport are completely different problems since the latter does not involve KL-Divergence during the computation.
>
> Ref:
>
> [1] On Unbalanced Optimal Transport: An Analysis of Sinkhorn Algorithm
>
> [2] Partial Optimal Transport with Applications on Positive-Unlabeled Learning
>
> - Q3: Proposition 1 is the first key result. However, the notation is not fully clear at that point in the article. Could the authors clarify the use of the star symbol * in expressions like $f^{*}$? (A similar comment applies to Proposition 3.)
>
> QA3: Thank you for your comment. The star symbol (*) in expressions denotes the optimal solution for $f$ and $\zeta$. We apologize for any confusion and will clarify this in the revised manuscript to ensure that the notation is clearer.
>
> - Q4: Regarding the statement of Proposition 1: I suggest rephrasing it as, "Given the SemiUOT functional with KL divergence ${J}_{SemiUOT}$..." A similar revision could be applied to Proposition 3.
>
> QA4: Thank you for your helpful suggestion.
> We agree with your proposed rephrasing and will revise the statement of Proposition 1 to: "Given the SemiUOT functional with KL divergence ${J}_{SemiUOT}$...".
> A similar revision will also be applied to Proposition 3 for consistency.
> We appreciate your input and will incorporate this change in the revised manuscript.
>
> - Q5: Do the new marginals $\alpha$ and $\beta$ represent vectors encoding the weights of discrete probability measures? That is, do their coefficients add to 1?
>
> QA5: Thank you for your question. In fact, the new marginals $\alpha$ and $\beta$ represent marginal weights. The coefficients of $\alpha$ and $\beta$ do not necessarily sum to 1. However, it is important to note that the sum of $\alpha$ and $\beta$ is equal (i.e, $\sum_{i=1}^M \alpha_i = \sum_{j=1}^N \beta_j$), ensuring consistency between the two marginals. We hope this clarifies the notation, and we will make sure to explain this more clearly in the revised manuscript.
>
>
> - Q6: I did not understand the first sentence on line 180. Could the authors clarify it?
>
> QA6: Thank you for your comment. The first sentence on line 180 refers to the procedure in Equation (11). Specifically, we first fix $\zeta = 0$, and then follow the procedure outlined in line 129. We use the LBFSGS algorithm to optimize the variable $u$ in $L_U$ and iteratively solve for the optimal solution of $u$. We will clarify this in the revised manuscript to ensure better understanding.
>
>
> - S1: Corollary 1 is not formally stated, so I would suggest presenting it as a Remark instead.
>
> SA1: Thank you for your valuable suggestion. We agree that Corollary 1 is not formally stated, and we will revise it to be presented as a Remark in the revised manuscript, as you suggested.
>
>
> - S2: While not necessary, a brief discussion of the advantages and disadvantages of SemiUOT compared to UOT would enrich the paper.
>
> SA2: Thank you for your helpful suggestion. While providing a direct, intuitive comparison between SemiUOT and UOT is challenging due to their distinct problem formulations, we appreciate your point that discussing their differences could enrich the paper.
>
> SemiUOT and UOT differ in several important aspects. Specifically, SemiUOT and UOT are equipped with different marginal constraints. UOT relaxes both of the marginal constraints while SemiUOT still keeps one marginal constraint. Therefore, SemiUOT and UOT can be skilled for different scenarios. That is, SemiUOT obtains SOTA in partial UDA meanwhile UOT can be more suitable in UDA and universal UDA tasks, which is reflected in our empirical studies.
>
> We will add a brief discussion of these differences in the revised manuscript to clarify the distinct contexts and applications of SemiUOT and UOT.
>
>
> - S3: In the Related Works section, I suggest including a citation of the paper [Y. Bai et al., "Linear Optimal Partial Transport Embedding," ICML, 2023], as it presents an alternative approach to addressing the computational complexity of unbalanced optimal transport problems, without relying on the addition of regularization terms.
>
> SA3: Thank you for your suggestion. We appreciate your recommendation to include the paper by Y. Bai et al., "Linear Optimal Partial Transport Embedding," ICML, 2023. We agree that it presents an interesting alternative approach to addressing the computational complexity of unbalanced optimal transport problems. We will include this citation in the Related Works section in the revised manuscript.
>
>
> - S4: I suggest referencing Algorithms 1 and 2 explicitly in the main text, even if only briefly, to guide the reader.
>
> SA4: Thank you for your suggestion. We agree that referencing Algorithms 1 and 2 explicitly in the main text would help guide the reader. We will make sure to include brief references to these algorithms in the relevant sections of the revised manuscript.
>
>
> - Typos1: Line 44: After "i.e.," I suggest using a lowercase letter and not capitalizing the following word.
>
> TA1: Thank you for your valuable feedback. We will make the necessary correction and use a lowercase letter after "i.e." as suggested.
>
>
> - Typos2: Lines 179, 180, and the paragraph between lines 193 and 199: Add commas after "Hence," "Then," and "After that," respectively.
>
> TA2: Thank you for your helpful suggestion. We will add the commas after "Hence," "Then," and "After that," as recommended.
>
> We hope that we have sufficiently addressed your concerns. If there is anything else we can answer or explain or discuss further, kindly do let us know. We really appreciate your encouragement for our paper.
>
> Kind regards,
> Authors

---

> > ### Comment · Reviewer_ETJd · 2025-08-02
> >
> > The authors have addressed my comments in a detailed and thoughtful manner. In particular, thanks for the clarification made for Q2.

---

> > > ### Author Response · Authors · 2025-08-02
> > >
> > > We deeply appreciate your insightful and comprehensive reviews/comments and thanks again for your suggestions!

---

### Official Review · Reviewer_teoF · 2025-06-24

**Clarity:** 3
**Significance:** 3
**Originality:** 3
**Rating:** 5
**Confidence:** 4

**Summary:**

This paper proposes a new framework for solving discrete Semi-Unbalanced Optimal Transport (SemiUOT) and Unbalanced Optimal Transport (UOT) problems. Existing methods often rely on entropy regularization, which leads to dense and inaccurate transport plans. To address this, the authors introduce the Equivalent Transformation Mechanism (ETM), which reformulates SemiUOT and UOT problems into classic optimal transport problems. The authors develop three variants: ETM-Exact, ETM-Approx, and ETM-Refine.

The paper also proposes Multiplier Regularized Optimal Transport (MROT), which uses KKT multipliers to guide the transport plan and improve solution quality.

Experiments on synthetic data, domain adaptation benchmarks, and treatment effect estimation show that ETM combined with MROT improves accuracy and efficiency over existing methods.

**Questions:**

1. In line 133, 'We iteratively update $L_U$ to reach the optimal solutions'. It seems that $L_U$ doesn't appear before this sentence.

2. In line 162, 'Since $\hat{f}^\star$ is close to $f^\star$'. Is there a theoretical bound for this?

3. Figure 4 shows that the choice of $\epsilon$ significantly affects the convergence behavior of ETM-Approx and ETM-Refine. For larger values (e.g., $\epsilon=0.1$), the algorithm fails to recover accurate marginals. However, the paper does not provide guidance on how to choose $\epsilon$ in practice. Could the authors clarify whether there is a principled way to select or adapt $\epsilon$ especially for varying values of $\tau$ or under different data scales?

4. Since the ETM framework reformulates UOT and SemiUOT problems as classic OT with new marginal weights derived from KKT conditions, could the authors explain more clearly how and why these new weights lead to better performance under noisy or mismatched data? Specifically, in what sense does solving OT with reweighted marginals provide robustness or accuracy improvements over applying classic OT directly to the original noisy distributions?

**Ethical Concerns:**

["NO or VERY MINOR ethics concerns only"]

**Final Justification:**

The only concern I have is about the choice of the hyperparameter $\epsilon$. The numerical experiments suggest that the convergence relies largely on $\epsilon$ and the selection of this parameter in the paper (and the rebuttal) is mostly heuristic. While a smaller $\epsilon$ is advocated by the theoretical analysis, it may cause numerical issues.

Overall, the paper is solid. I champion the final acceptance of the paper.

**Limitations:**

Yes.

**Paper Formatting Concerns:**

NA.

**Quality:**

3

**Strengths And Weaknesses:**

Quality: The paper presents a solid and well-justified theoretical contribution. The proposed Equivalent Transformation Mechanism (ETM) is rigorously derived, and its variants (ETM-Exact, ETM-Approx, ETM-Refine) are grounded in convex analysis and KKT conditions. The integration with Multiplier Regularized Optimal Transport (MROT) provides a framework to enhance the sparsity and accuracy of transport plans. The empirical evaluation is extensive, covering both synthetic and real-world datasets across domain adaptation and treatment effect estimation tasks.

Originality: The paper introduces a new perspective by transforming discrete SemiUOT and UOT problems into classic OT through reweighted marginals. The use of KKT multipliers as a regularization term in MROT appears to be an interesting idea that improves transport plan quality.

Significance: This proposed method has potential impact on applications where interpretability and sparsity are important, such as domain adaptation, generative modeling, and causal inference.

Clarity: The paper is generally clear and well-organized. The connection between the ETM variants and standard OT is carefully explained. The derivations are detailed, and algorithmic procedures are summarized in pseudocode, which aids reproducibility.

---

> ### Author Rebuttal · Authors · 2025-07-28
>
> - Q1: In line 133, 'We iteratively update $L_{U}$  to reach the optimal solutions'. It seems that $L_{U}$ doesn't appear before this sentence.
>
> QA1: Thank you for your careful review. We apologize for the typographical error in line 133. The term "$L_{U}$" should indeed be "$L_{P}$". We will correct this mistake in the revised manuscript.
>
> - Q2: In line 162, 'Since $\hat{f}^{*}$ is close to $f^{*}$'. Is there a theoretical bound for this?
>
> QA2: Thank you for your insightful comment.  To start with, we consider the analysis between optimal results of $ f^o $ and $\hat{f}^o$ and thus we set $\zeta = 0$ in SemiUOT. Then we define $ E_{P}(f)= L_{P}({f}) $ and $ S_{P}(\hat{f}) = \hat{L}_{P}(\hat{f}) -  \epsilon \log M $.
>
> Hence we have the following relationships: (1) $S_P(f^o) \le E_P(f^o) \le E_P(\hat{f}^o) \le S_P(\hat{f}^o) + \epsilon \log M$, (2) $S_P(\hat{f}^o) \le S_P(f^o) \le E_P(f^o) \le S_P(f^o) + \epsilon \log M$ and thus it shows $| S_{P}(f^o)  -  S_{P}(\hat{f}^o)| \le \epsilon   \log M$ [1]. Moreover we have:
>
> $$
> |E_{P}(f^o) - S_{P}(\hat{f}^o)| \le |E_{P}(f^o) - S_{P}(f^o)| + |S_{P}(f^o) - S_{P}(\hat{f}^o)| \le 2\epsilon \log M
> $$
>
> Therefore we can observe that $ f^o $ and $\hat{f}^o$ will get closer with smaller $\epsilon$. Note that this coincides with the approximation written in line 138. Likewise we consider the optimal results of $ u^o $ and $\hat{u}^o$ with $\zeta = 0$ in UOT by $ E_{U}(f)= L^u_{U}({u}) $ and $ S_{U}(\hat{u}) = \hat{L}^u_{U}(\hat{u}) -  \epsilon \log M $. The definition of $L^u_{U}({u})$ and $\hat{L}^u_{U}(\hat{u})$ is given in Eq.(52) and Eq.(53). We can also have $| S_{U}(u^o)  -  S_{U}(\hat{u}^o)| \le \epsilon \log M$ and $|E_{U}(u^o) - S_{U}(\hat{u}^o)| \le  2\epsilon \log M$ accordingly. In conclusion, the solution on ETM-Approx will be relatively close to the exact solution provided by ETM-Exact with smaller $\epsilon$.
>
> In conclusion, the solution on ETM-Approx will be relatively close to the exact solution provided by ETM-Exact with a smaller $\epsilon$.
>
> Ref: [1] Smooth minimization of non-smooth functions.
>
>
> - Q3: Figure 4 shows that the choice of $\epsilon$ significantly affects the convergence behavior of ETM-Approx and ETM-Refine. For larger values (e.g., $\epsilon = 0.1$), the algorithm fails to recover accurate marginals. However, the paper does not provide guidance on how to choose $\epsilon$ in practice. Could the authors clarify whether there is a principled way to select or adapt $\epsilon$ especially for varying values of $\tau$ or under different data scales?
>
>
> QA3: Thank you for your valuable comment. As suggested, the choice of $\epsilon$ plays an important role in the convergence behavior of both ETM-Approx and ETM-Refine. Based on the error bound,
> i.e., $\left| \epsilon \log \left[ \sum_{k=1}^{M} e^{\frac{f_k - C_{kj}}{\epsilon}} \right] - \sup_{k \in [M]} \left[ f_k - C_{kj} \right] \right| \leq \epsilon \log M$, provided in line 138 of the paper, smaller values of $\epsilon$ generally lead to better approximation results.
> In practice, we typically select $\epsilon = 0.01$, as it strikes a good balance between performance and convergence stability.
> Therefore, we recommend using this value in most scenarios, but depending on the specific problem or data scale, slight adjustments may be necessary.
>
>
> - Q4: Since the ETM framework reformulates UOT and SemiUOT problems as classic OT with new marginal weights derived from KKT conditions, could the authors explain more clearly how and why these new weights lead to better performance under noisy or mismatched data? Specifically, in what sense does solving OT with reweighted marginals provide robustness or accuracy improvements over applying classic OT directly to the original noisy distributions?
>
>
> QA4: Thank you for your insightful comment. To address your question, we provide the following explanations: (1) We will assign small weights to the noise and outlier data via ETM-based method for obtaining new marginal weights. (2) In solving for the new marginal weights, our method introduces an additional term, $s_{ij}$, which adheres to the KKT conditions. This term provides useful information that classical OT with a regularization term (e.g., Sinkhorn) with reweighted marginals does not account for. The inclusion of $s_{ij}$ offers valuable guidance in solving the complete coupling matrix, helping to achieve a more precise solution. Traditional OT with regularization terms (e.g., Sinkhorn), without considering these guiding prior conditions, may lead to less accurate and dense matching results. Our approach incorporates a KKT multiplier regularization term into the optimization problem, enhancing the clarity of the matching process while minimizing the risk of incorrect pairings.
>
> We hope that we mainly addressed your concerns sufficiently. If there is anything else we can answer or explain or discuss further, kindly do let us know. We really appreciate your encouragement for our paper.
>
> Kind regards,
> Authors

---

> > ### Comment · Reviewer_teoF · 2025-08-03
> >
> > I would like to thank the authors for their comprehensive explanation. My concerns are mostly addressed.

---

> > > ### Author Response · Authors · 2025-08-03
> > >
> > > We sincerely thank the reviewer for their positive feedback and for acknowledging our efforts in the rebuttal. We are pleased to know that most of the concerns have been addressed. If there are any remaining issues or points requiring clarification, we would be more than happy to provide further explanation.

---

### Official Review · Reviewer_FSeC · 2025-06-28

**Clarity:** 2
**Significance:** 3
**Originality:** 2
**Rating:** 4
**Confidence:** 4

**Summary:**

This paper addresses the challenge of accurately solving discrete SemiUOT and UOT problems. They propose an Equivalent Transformation Mechanism (ETM) approach to determine the marginal probability distributions of SemiUOT with KL divergence. A KKT-Multiplier regularization term is proposed with Multi plier Regularized Optimal Transport (MROT) to achieve more accurate matching results. Extensive experiments demonstrate the effectiveness of our proposed methods in addressing SemiUOT and UOT problems. Overall, this paper contributes a practically effective framework for solving SemiUOT and UOT.

**Questions:**

1.	In (27), $f^*$ seems also relative to the $\tau$ and it is ignored when setting $\tau \to \infty$.
2.	There are many computations of the form $\exp(\frac{\cdot}{\epsilon})$ in ETM+MROT-ENT. Could numerical instability arise when $\epsilon\to 0$?
3.	In MROT, KKT multipliers are used as a regularization mechanism to induce sparsity. What is the theoretical validity for this design, and how does it affect optimization stability or convergence?
4.	What is the relationship between the optimal solution and optimal value of ETM-Exact and ETM-Approx? A theoretical explanation is  recommended.
5.	It would be positive if the authors demonstrate the adaptability of this method to outliers.
6.	If $u^*$ and $v^*$ are inexact due to the computational error, will the Corollary 2 and Proposition 4 still hold?

**Ethical Concerns:**

["NO or VERY MINOR ethics concerns only"]

**Final Justification:**

I suggest borderline accept. The authors have resolved most of my concerns.

**Limitations:**

Yes

**Paper Formatting Concerns:**

The  paper is well-formatted.

**Quality:**

3

**Strengths And Weaknesses:**

Strengths:

1. The SemiUOT and UOT problems are clarified explicitly as relaxations of classical OT. The ETM formulation allows the exact and approximate solutions and reduces the complexity of solving these problems. It is effective to use the solved marginal distributions for the transfer to standard OT problems.
2. This method balances the trade-off between computing exact solutions and improving efficiency in ETM, and further leverages the obtained exact multipliers for regularization, resulting in sparser and more accurate transport matching.
3. The comprehensive experiments demonstrate the effectiveness of ETM+MROT, which consistently yields state-of-the-art results across a diverse set of metrics in many applications.

Weaknesses:

1.	Theoretically, coherence and completeness of this paper should be improved.  On one hand, the authors should first state assumptions, such as the existence of a c-transform and other implicit conditions used in the proofs, and summarize the conclusion in a theorem. On the other hand, there is a lack of detailed proof and analysis of the convergence rate.
2.	The writing and organization could be improved to enhance clarity. The organization and layout seems disordered, and the inclusion of numerous detailed explanations in the main text hinders the understanding of the conclusions. Moreover, the paper lacks a diagram or concise pseudocode illustrating the overall algorithm overview. Typos, such as the incorrect notation in equation (20) and the misuse of inner product symbols in equations (23) and (25), further affect the readability of the paper.
3.	A better approach would be to design a customized optimization algorithm tailored for solving the OT problem, rather than directly applying L-BFGS. On one hand, leveraging the inherent structure of the problem may lead to faster computation. On the other hand, we are also concerned with the convergence behavior of the optimization algorithm for this specific problem.
4.	One potential theoretical weakness is the lack of discussion on robustness to outliers. In real-world datasets, noisy or out-of-distribution samples can degrade OT performance. It would be valuable for the authors to either empirically or theoretically analyze how their approach behaves under such perturbations.
5.	The novelty of the proposed approach may be somewhat limited. ETM resembles the discrete form of Equation (9) in Choi et al. (2023) under KL regularization, and MROT adapts standard regularized OT solvers with an additional multiplier. Clarifying the conceptual gap from prior work would strengthen the perceived contribution.

---

> ### Author Rebuttal · Authors · 2025-07-28
>
> - W1.1:  Theoretically, coherence and completeness of this paper should be improved.
>
> A1.1: We first highlight that our propositions are technically sound and correct. c-transform is the basic knowledge of optimal transport, and we have illustrated all implicit conditions and conclusions in our theoretical analysis. Please directly indicate the questions you are concerned about. We are willing to clarify them and further polish the writing in the camera-ready version.
>
> - W1.2: There is a lack of detailed proof and analysis of the convergence rate.
>
> A1.2: We **HAVE** provided the detailed proof and analysis of the convergence rate (shown in Remark 1, line 158-159) of our proposed ETM-Approx for SemiUOT as $\mathcal{O}(NM\log(1/\varepsilon_{\rm err}))$. The proof is shown on page 19 (line 668-673). The time complexity for ETM-Refine is $O(NM\log (1/ε_{err}) + MN (\log M) D_T)$ where $D_T = \log \log (1/ε_{err})$ denotes the number of iterations with super-linear convergence rate. In terms of MROT,  the convergence speed on MROT is determined by different kinds of regularization terms. For instance, MROT-Ent has the same convergence rate as Sinkhorn and ETM-Refine + MROT-Ent has the computation complexity as $O(NM\log (1/ε_{\rm err}) + NM (\log M) D_T +  NM/ε_p^2)$ where $ε_p$ denotes the error between $Tr(C^T\pi)$ and $Tr(C^T\pi^*)$.
>
> - W2.1: The organization and layout seems disordered, and the inclusion of numerous detailed explanations in the main text hinders the understanding.
>
> A2.1: We must highlight that we never change the paper layout and format. The detailed mathematical explanations will be shown as formulas in the camera-ready version.
>
> - W2.2: The paper lacks a diagram or concise algorithm pseudocode.
>
> A2.2: We respectfully but firmly disagree with the reviewer’s statement. We **HAVE** provided the concise pseudocode as Alg.1 and Alg.2 on page 20 and 24. Moreover, we **HAVE** provided the code in the supplementary material.
>
> - W2.3: Some typos exist.
>
> A2.3: (1) $u$ and $v$ should be $f$ and $g$ respectively in Eq.(20). (2) We will delete the inner product symbol inside the KL-Divergence.
>
> - W3: Design a customized optimization algorithm for solving the OT problem rather than directly applying L-BFGS is better. How about the convergence behavior?
>
> A3: Let us kindly point out the **factual misunderstanding** that we **HAVE** provided a customized optimization algorithm as ETM-Approx and ETM-Refine (our core contribution) for solving UOT/SemiUOT. The convergence analysis can be found in A1.2, that is, ETM-Exact is $O(NM \log M \log (1/ε_{err}))$ and  ETM-Approx is $O(NM\log (1/ε_{err}))$ and ETM-Refine is $O(NM\log (1/ε_{err}) + MN (\log M) \log \log (1/ε_{err}))$.
>
> - W4 and Q5: It would be positive if the authors demonstrate the adaptability of this method to outliers and out-of-distribution samples.
>
> A4 and QA5: We respectfully but firmly disagree with the reviewer’s statement. We **HAVE** conducted the outliers experiments in the main paper. (1) We add 20% outliers in the synthetic datasets and report the visualization results in Fig.1-4. (2) OOD samples are also considered on real-world datasets as partial UDA and universal UDA (Table 1-6). Our proposed ETM-based method can achieve much better results by tackling outliers or domain adaptation with OOD samples.
>
> - W5: The novelty may be somewhat limited.
>
> A5: **We strongly disagree with this comment.** Theorem in [Choi] and our proposed ETM differ significantly in several aspects:
>
> (1) [Choi] mainly considers the continuous case and does not involve the translation invariant term $\zeta$. As we showed in Appendix M, without $\zeta$, the transformed marginal probability will not be equal in practice, and therefore [Choi] cannot guarantee SemiUOT/UOT can be transformed into classic OT, making it impractical in the discrete scenario.
>
> (2) [Choi] primarily discusses UOT and does not explore SemiUOT. Our proposed ETM method specifically addresses the discrete case by directly calculating the exact value of $\boldsymbol{\pi}$, a topic not covered by [Choi], and considering the case when $\tau \rightarrow \infty$.
>
> (3) We provide time complexity and convergence analysis for the proposed method which [Choi] did not provide.
> In summary, our proposed ETM method ensures that the transformed marginal probability is equal, making the equivalent transformation practical for further obtaining $\boldsymbol{\pi}$ with KKT regularization as MROT. Although MROT shares the same calculation procedure as standard regularized OT solvers, it first proposes KKT regularization term, which leads to more precise results. Last but not least, we **HAVE** concluded this in our paper in line 70-73.
>
> - Q1: In (27), $f^∗$ seems also relative to the $\tau$?
>
> QA1: NO, $f^∗$ can be ignored and the final result is **correct**. Let us point out the factual misunderstanding here. Indeed, we can consider $\tau \rightarrow \infty$ in Eq.(5) by remembering $\lim_{\tau \rightarrow +\infty} \tau \exp \left( -\frac{f_i }{\tau }  \right) = -f_i$ and set $\zeta = 0$:
> $$
> \min_{f} L_{\rm P}= -  \sum_{i=1}^M    a_i f_i   -   \sum_{j=1}^N \left[  \inf\limits_{k \in [M]} \left[ C_{kj} - f_k \right] \right] b_j
> $$
> Therefore, $f^∗$ will not involve $\tau$ when $\tau \rightarrow \infty$. This relationship is also first proposed in this paper, while previous work in [Choi] did not point out.
>
> - Q2: Could numerical instability arise for ETM when $\epsilon \rightarrow 0$?
>
> QA2: We conduct the experiments by setting $\epsilon = \\{ 0.01, 0.1\\}$  shown in line 276 and Fig.4 and the proposed ETM method can provide stable output results. Note that $\epsilon = 0.01$ is relatively close to zero [1, 2] (i.e., see Fig.4.9 in [2]). Our proposed method can still reach stable output when $\epsilon=0.001$ and the results are almost the same as $\epsilon=0.01$. Moreover, we involve the log-sum-exp mechanism which can make the computation more stable according to the previous works (e.g., Chapter 3.3 in [1]).
>
> Ref:
>
> [1] Deriving the Gradients of Some Popular Optimal Transport Algorithms
>
> [2] Computational Optimal Transport: With Applications to Data Science
>
> - Q3: What is the theoretical validity for this design of MROT, and how does it affect optimization stability or convergence?
>
> QA3: The theoretical insight lies in the complementary KKT conditions as $s_{ij}\pi_{ij} = 0$ and we HAVE written it line 242-244 and line 248-251. **Adding multiplier regularization term will not affect optimization stability or convergence**. That is, adding the KKT-multiplier regularization term $\mathcal{G}({\pi}, {s}) = \langle {\pi}, {s} \rangle$ is equivalent to change the cost matrix from $C_{ij}$ to $C_{ij} + \eta s_{ij}$ shown in line 239. For instance, MROT-Ent has the same optimization stability and convergence against Sinkhorn.
>
> - Q4: What is the relationship between the optimal solution and optimal value of ETM-Exact and ETM-Approx?
>
> QA4: To start with, we consider the analysis between optimal results of $ f^o $ and $\hat{f}^o$ and thus we set $\zeta = 0$ in SemiUOT. Then we define $ E_{P}(f)= L_{P}({f}) $ and $ S_{P}(\hat{f}) = \hat{L}_{P}(\hat{f}) -  \epsilon \log M $.
>
> Hence we have the following relationships: (1) $S_P(f^o) \le E_P(f^o) \le E_P(\hat{f}^o) \le S_P(\hat{f}^o) + \epsilon \log M$, (2) $S_P(\hat{f}^o) \le S_P(f^o) \le E_P(f^o) \le S_P(f^o) + \epsilon \log M$ and thus it shows $| S_{P}(f^o)  -  S_{P}(\hat{f}^o)| \le \epsilon   \log M$ [1]. Moreover we have:
>
> $$
> |E_{P}(f^o) - S_{P}(\hat{f}^o)| \le |E_{P}(f^o) - S_{P}(f^o)| + |S_{P}(f^o) - S_{P}(\hat{f}^o)| \le 2\epsilon \log M
> $$
>
> Therefore we can observe that $ f^o $ and $\hat{f}^o$ will get closer with smaller $\epsilon$. Note that this coincides with the approximation written in line 138. Likewise we consider the optimal results of $ u^o $ and $\hat{u}^o$ with $\zeta = 0$ in UOT by $ E_{U}(f)= L^u_{U}({u}) $ and $ S_{U}(\hat{u}) = \hat{L}^u_{U}(\hat{u}) -  \epsilon \log M $. The definition of $L^u_{U}({u})$ and $\hat{L}^u_{U}(\hat{u})$ is given in Eq.(52) and Eq.(53). We can also have $| S_{U}(u^o)  -  S_{U}(\hat{u}^o)| \le \epsilon \log M$ and $|E_{U}(u^o) - S_{U}(\hat{u}^o)| \le  2\epsilon \log M$ accordingly. In conclusion, the solution on ETM-Approx will be relatively close to the exact solution provided by ETM-Exact with smaller $\epsilon$.
>
> Ref: [1] Smooth minimization of non-smooth functions.
>
> - Q6:If obtained $u$ and $v$ are inexact due to the computational error, will Corollary 2 and Proposition 4 still hold?
>
> QA6: We answer this question with two aspects.
>
> (1) Corollary 2 is strictly held when $u$ and $v$ (or $f$ and $g$) are the optimal solutions.
>
> (2) According to the response QA4, we show that the solution on ETM-Approx will be relatively close to the exact solution provided by ETM-Exact with smaller $\epsilon$. Therefore, ETM-Approx can also provide an approximate result on ${s}$ with an inexact but approximate solution on $u$ and $v$ (or $f$ and $g$). We also conduct the corresponding experiments as ETM-Approx + MROT-Ent and report the results in Fig.3(b) - (d). From that we can observe that ETM-Approx + MROT-Ent can also reduce the absolute error in SemiUOT. Moreover we conduct the ETM-Approx + MROT-Ent on UOT with $\tau=0.1$ and $N=500$ samples for synthetic datasets and report the absolute error below:
>
> | Method | ε=0.001 | ε=0.01 | ε=0.1 | ε=1 |
> |-|-|-|-|-|
> | ETM-Approx + MROT-Ent | 0.14 | 0.15 | 0.21 | 0.34 |
> | ETM-Refine + MROT-Ent | 0.10 | 0.11 | 0.14 | 0.22 |
>
> The above results indicate that one can still adopt MROT to find $\pi$ even if $u$ and $v$ (or $f$ and $g$) are just approximate/inexact solutions.
>
> We hope that we addressed mainly of your concerns sufficiently, and if you agree, we would kindly request for updating the review in light of this response. If there is anything else we can answer or explain or discuss further, kindly do let us know.
>
> Kind regards,
> Authors

---

> ### Comment · Reviewer_FSeC · 2025-08-02
> **Comments on Rebuttal**
>
> **Response to A1.1**
> Even though the c-transform is basic knowledge in optimal transport, it is still suggested to provide a rigorous definition in advance. For example, Choi et al. first specified that the dual multipliers are Lebesgue integrable with respect to the measure. The authors have claimed that the discrete case is different, which makes it even more important to include this part of the definition to help readers understand the distinction between the discrete and continuous settings. A rigorous definition does not conflict with the soundness of the theory.
>
> **Response to A1.2**
> It would be better to complete the paper with the rebuttal explanation on ETM-Refine convergence. There are also concerns on the convergence of $\hat{f}^\*$, which it is claimed to be a super-linear convergence rate [41,42,77,101,35].
>
> **Response to A2.1**
> To improve clarity, would it be helpful to summarize the derivations and conclusions into a few lemmas?
>
> **Response to A2.2**
> It is suggested that the algorithm pseudocode is shown or referenced in the main text instead of Appendix for better understanding the paper.
>
> **Response to A3**
> The "factual misunderstanding" mentioned by the authors might be incorrect. In Alg 1 and Alg 2, L-BFGS is indeed used on ETM-Exact, and ETM-Refine calls ETM-Exact as well. This is essentially a non-smooth optimization problem—wouldn't it be natural to expect the use of a customized algorithm in such a case? For ETM-Refine, the final convergence of $f^*$ is achieved by L-BFGS and the convergence of L-BFGS is analyzed for general functions, not specifically for this OT problem.
>
> **Response to A4**
> The comment highlights a theoretical limitation in robustness to outliers, whereas the authors’ rebuttal addresses only the experimental results. The authors may wish to consult the discussion in Lemma 1 of [1].
>
> **Response to A5**
> It would be helpful to add a clarification in the paper addressing the conceptual gap between this work and prior work, as suggested in the review.
>
> **Response to QA1**
> In (27), $f_i$ is the component of $f^\*$, the optimal solution of (16)
> $$
>     f^\*,g^\*,\zeta^\* =\arg \min_{f,g,\zeta} \left[\tau\sum_{i=1}^Ma_i\exp\left(-\frac{f_i+\zeta}{\tau}-\sum_{j=1}^Nb_j(g_j-\zeta)\right)\right].
>     $$
> Hence, $f_i(\tau)$ is relative to the parameter $\tau$. When $\tau\to \infty$, the authors should first analyze the relationship between $f_i(\tau)$ and $\tau$. Please show a rigorous proof for the readers.
>
> **Response to QA3**
> Since regularizers like Entropy and L2 do not encourage the sparsity, it is more solid to give a mathematical theorem to explain the sparsity from KKT multipliers instead of the insight that has been written.
>
> **Response to QA4**
> This is a promising result. Could we leverage the convergence properties of ETM-exact to determine an optimal trade-off point with minimal computational cost?
>
>
> [1] Fatras, Kilian, et al. "Unbalanced minibatch optimal transport; applications to domain adaptation." International conference on machine learning. PMLR, 2021.

---

> ### Author Response · Authors · 2025-08-03
>
> **Reply to A1.1 A 2.1, A2.2 and A5** We will improve the clarity as suggested.
>
> **Reply to A 1.2** We adopt LBFGS [101] to optimize ETM-Approx and ETM-Refine. LBFGS-based optimizers (e.g., sub-LBFGS [101]) can tackle the non-smooth convex optimization problem. ETM-Refine provides a close optimum point initialization, therefore it has indeed a super-linear convergence rate according to [101] (in page 16 [101]).
>
> **Reply to A3** **There are still factual misunderstandings, i.e., ETM-Refine calls ETM-Exact as well**. Indeed, **ETM-Refine is completely different from ETM-Exact**. **It can be easily verified in Algo.1 and Algo.2.** Indeed, ETM-Exact directly adopts LBFGS [101] for the optimization. Meanwhile LBFGS is indeed used on ETM-Refine by taking the approximate solution obtained by ETM-Approx as the initial point rather than using random initialization. Therefore, the convergence rate of **ETM-Refine is provided as $\mathcal{O}\left(NM \log\left(\frac{1}{\varepsilon_{\text{err}}}\right) + MN (\log M) \log\log\left(\frac{1}{\varepsilon_{\text{err}}}\right)\right)$, which is exactly customized for OT problem.** Meanwhile the time complexity of ETM-Exact is $O(NM\log M \log (1/ε_{err}))$. We can observe that ETM-Refine could be faster than ETM-Exact.
>
> **Reply to A4** We read through the Lemma1 provided in [1] and it shows that traditional OT cannot address outliers since the marginal weights are fixed. In other words, UOT relaxed the marginal constraints to overcome the outliers. **Indeed, in our paper, we definitely show that the UOT can adjust the data sample weights (see in Proposition 3) to enhance the model robustness against outliers which coincided with Lemma1 in [1].** That is, outliers will be assigned lower marginal weights to avoid getting matched as shown in Fig.1-2.
>
> **Reply to QA1** To start with, let us kindly point out the **errors** in your comment. That is, the optimization problem should be indeed written as below:
>
> $$
> \min_{{f}, \zeta}L_{\rm P}= \tau  \sum_{i=1}^M    a_i \exp \left( -\frac{f_i + \zeta}{\tau} \right)    -   \sum_{j=1}^N \left[  \inf\limits_{k \in [M]} \left[ C_{kj} - f_k \right] - \zeta \right] b_j
> $$
>
> According to the Algo.1, $\zeta$ will not affect the value of $f^*$. Therefore, it is reasonable to set $\zeta = 0$ to simplify the problem. (We also set $\zeta = 0$ shown in line 129 for initialization). Then the problem turns out to be:
> $$
> \min_{{f}}L_{\rm P}= \tau  \sum_{i=1}^M    a_i \exp \left( -\frac{f_i}{\tau} \right)    -   \sum_{j=1}^N \left[  \inf\limits_{k \in [M]} \left[ C_{kj} - f_k \right] \right] b_j
> $$
>
> **We can show that $f_i$ is not relative to $\tau$ when $\tau \rightarrow \infty$.**
>
> First we should review the Taylor expansion as $\lim_{\tau \rightarrow +\infty} \tau \exp \left( -\frac{f_i }{\tau }  \right) = \tau (1-f_i/\tau) = \tau - f_i$. Therefore the optimization problem can be viewed as:
>
> $$
> \min_{f} L_{\rm P}= \tau \sum_{i=1}^M    a_i -  \sum_{i=1}^M    a_i f_i   -   \sum_{j=1}^N \left[  \inf\limits_{k \in [M]} \left[ C_{kj} - f_k \right] \right] b_j
> $$
>
> Note that $\tau \sum_{i=1}^M  a_i$ can be neglected in the optimization problem. Thus it is equivalent to the following problem:
>
> $$
> \min_{f} L_{\rm P}= -  \sum_{i=1}^M    a_i f_i   -   \sum_{j=1}^N \left[  \inf\limits_{k \in [M]} \left[ C_{kj} - f_k \right] \right] b_j
> $$
>
> **It is quite apparently that $f^*$ does not involve $\tau$ when $\tau \to \infty$ and therefore our proposition is definitely correct**. In our opinion, it is not difficult to verify them by using **the basic knowledge of calculus and convex optimization.** **Please kindly point out any incorrect parts and we will be happy to discuss them further constructively.**
>
> **Reply to QA3** The proposed MROT based on KKT multipliers is used to solve SemiUOT and UOT efficiently and accurately. **We HAVE written in the line 242-257 with detailed illustrations**. When $\pi_{ij} = 0$ it leads to $s_{ij} > 0$. Using MROT-Ent as an example, $\lim_{\eta_G \to \infty} \exp(-\eta_G s_{ij}/\eta_{Reg}) = 0$ and thus our method can provide sparse results. So the main insight is rigidly consistent with our original paper, i.e., we use KKT multipliers to solve the matching problem with more precise solutions.
>
> **Reply to QA4** Firstly, we analyze the **error bound between the exact solution solved by ETM-Exact and the approximate solution solved by ETM-Approx** in QA4, rather than the convergence properties of ETM-Exact. The time complexity of ETM-Exact is $O(NM\log M \log (1/ε_{err}))$. In fact, we set $ε_{err} = 10^{-6}$ during the computation. Smaller $ε_{err}$ could lead to almost the same results reported in paper with more computation time.

---

> ### Author Response · Authors · 2025-08-03
>
> **In conclusion, we would like to clarify the relationships among three ETM methods, and emphasize that our primary contribution is to propose ETM-Refine, as detailed in our main paper.** Specifically, **ETM-Exact** is the method derived from L-BFGS [101], which can achieve the optimal solution but costs an overwhelming computation time. **ETM-Approx** is the efficient approximation method proposed by us, which can accelerate the computation by benefiting from the indicators ${s}$. However, our main contribution is to propose **ETM-Refine**, which takes the advantages of both ETM-Exact and ETM-Approx. ETM-Refine firstly finds the approximate results close to the optimal solution (ETM-Approx), and achieves the exact results with the superlinear convergence (ETM-Exact). **We kindly ask that you consider this context when reviewing our work, particularly in terms of evaluating its contribution and computational benefits.**
>
> We hope that we addressed mainly of your concerns sufficiently, and if you agree, we would kindly request for updating the review in light of this response. If there is anything else we can answer or explain or discuss further, kindly do let us know.
>
> Kind regards,
> Authors

---

### Official Review · Reviewer_PbZU · 2025-07-02

**Clarity:** 2
**Significance:** 3
**Originality:** 3
**Rating:** 4
**Confidence:** 2

**Summary:**

This paper provides a new technique for solving the unbalanced-OT (UOT) and semi unbalanced-OT (semi-UOT) problems. Both problems allow for partial optimal transport solutions (where not all mass is transported). In addition to the cost of the transport plan, an additional penalty function is added, which contributes to the objective function based on the amount of untransported mass. (In semi-OT, only one distribution is considered in this additional penalty function, while UOT considers both sides, with possibly different coefficients). Different variants of this “untransported mass” penalty function have been considered, but this paper specifically uses KL divergence.

The key contribution of this technique is a transformation, where any instance of the UOT or semi-UOT problem (under KL-divergence) can be transformed into an instance of the regular OT problem, in such a way that the solution to this regular OT problem instance provides a solution to the original UOT or semi-UOT problem. This transformation is named the “Equivalent Transformation Mechanism” (ETM). Furthermore, multiple variants of ETM are presented, offering tradeoffs in computation time and accuracy – these are named ETMExact, ETMApprox, and ETMRefine.

As both the semi-UOT problem and UOT problem have been investigated before, this paper seeks to differentiate itself in terms of quality, with a specific emphasis on not using entropy regularization like most previous methods, resulting in sparser solutions. The paper also provides extensive experimental evaluation on real and synthetic data. These include multiple variations of ETM, multiple prior work methods / baselines, and some different penalty functions for the untransported mass.

**Questions:**

Questions:
-	Just a question : Is it not true that Semi-OT is a special case of UOT, since you can always set the hyperparameter coefficient of one distribution to 0? I ask this because, one of your selling points is that your approach works for both. Why can’t existing UOT approaches apply to Semi-OT directly?

-	Under table 1, you take the best approach without UOT, called JUMBOT, and you add to it various methods of computing UOT, such as MM-UOT, GEM-UOT, $\ell_2$ norm solver, and sparse solver. All of these UOT approaches offer a modest improvement over JUMBOT. Then, you show that your ETM + MROT approach improves upon the previous UOT approaches for this task by an additional modest amount. However, I find it odd that, for all of your other similar tables, including Tables 2, 4, 5, 6, 7, you omit all variants of “state of the art + UOT(some other solver that’s not yours)”, and instead only consider your approach for the UOT component. Based on the trend in Table 1, it seems as if all UOT variants, yours or not, seem to improve upon the baseline “state of the art” without UOT. That’s not so important for your paper. Instead, your burden of proof is to illustrate that your variant of UOT specifically outperforms other existing variants of UOT in practice. Otherwise, what’s the role of the experiments here?

-	Please detail how your approach compares to the existing UOT approaches with regards to time complexity and practical performance. The only clear discussion I saw of this is Figure 3, which doesn’t quite clear things up for me. The thorough treatment of running time seems lacking in both the theory as well, since the iteration count is not bounded. Furthermore, when I see sizes like N = 50 for the experiments, I start to wonder why you can’t bring N higher. Often, OT is fine with N = 1000 or more without an issue. How fast is your approach?

-	Line 282 : You performed 5 experiments, but that is not a very large sample. How sure can you be that, if the experiment were repeated, the average would be comparable to what you have here?

-	Figure 3 : You have a couple of charts / labels swapped here. Please address this. I would also recommend adding a consistent color scheme for the plots across UOT and semi-UOT charts respectively. Furthermore, the list of variants of your algorithm between (b) and (c) are different. What I would really like to see is an accuracy versus runtime comparison of your approach(es) versus existing approaches. Finally, the only running times given here are for $\tau = 1$, which you stated in your introduction other approaches work well for. So, why not give your efficiency for large $\tau$?

Corrections :

Line 34 – 35 : Grammar : “While … solutions” is an incomplete sentence. Reword.

Line 58 : Did you mean “MROT”, not “MGOT”?
Also, see comments about Figure 3 above.

**Ethical Concerns:**

["NO or VERY MINOR ethics concerns only"]

**Final Justification:**

Thank you authors for your response. After reviewing your rebuttal, as well as the discussion from the other reviewers, I have increased my rating accordingly. I would recommend, in the final version, that the discussion of convergence analysis be highlighted more clearly in the main body of the paper.

**Limitations:**

Nothing to note.

**Quality:**

2

**Strengths And Weaknesses:**

Strengths :
-	Very extensive experimental evaluation, which shows an improvement over existing in terms of accuracy when applied to real-world applications.

-	Thorough treatment of technical details.

-	The technique is flexible in that any OT solver can be used along with ETM, and it can be easily applied to UOT and Semi-OT. In some sense, ETM seems to act as a “booster” of sorts when added to some existing approaches, based on the experimental results.

Weaknesses :
-	The experimental evaluations, in terms of accuracies, offer relatively modest improvements over the existing approaches, usually a couple percentage points (although this improvement is quite consistent). I have some concern about the experimental design.

-	Some parts of the complexity analysis of the ETM approach seem unclear in relation to other work. I would like to clearly see a comparison of Big-O time complexities of the approaches and steps of this work as well as descriptions of Big-O time complexities of other work. I want to be sure that the solution-time tradeoff of your approach is comparable to existing approaches.

-	I have a few questions about the experiments, which might be easily addressable in the author response.

---

> ### Author Rebuttal · Authors · 2025-07-28
>
> - W1.1: The experimental evaluations, in terms of accuracy, offer relatively modest improvements over the existing approaches.
>
> A1.1: The experiment design aims to validate the efficacy of our proposed method among different tasks. Our improvement is consistent but significant on multiple tasks, including UDA, partial UDA, and universal UDA. For instance, our proposed method boosts model performance by 5.71% on Office-Home for partial UDA and 3.50% on Office-Home for universal UDA.
>
> - W1.2: I would like to clearly see a comparison of Big-O time complexities of the approaches.
>
>
> A1.2: The time complexity of the proposed method can be analyzed via the following procedure:
>
> (1) To start with, we would like the highlight that we provide the details of ETM-Approx whose time complexity is $\mathcal{O}(NM\log(1/ε_{err}))$ with bounded iteration shown in page 19 (line 668-673) and convergence rate (shown in Remark 1, line 158-159).
>
> (2) The time complexity of ETM-Exact is $O(NM\log M \log (1/ε_{err}))$. The time complexity for ETM-Refine is $O(NM\log (1/ε_{err}) + MN (\log M) D_T)$ where $D_T = \log \log (1/ε_{err})$ denotes the number of iterations with super-linear convergence rate.
>
> (3) In terms of MROT,  the convergence speed on MROT is determined by different kinds of regularization terms. For instance, MROT-Ent has the same convergence rate as Sinkhorn and ETM-Refine + MROT-Ent has the computation complexity as $O(NM\log (1/ε_{\rm err}) + NM (\log M) \log \log (1/ε_{err}) +  NM/ε_p^2)$ where $ε_p$ denotes the error between $Tr(C^T\pi)$ and $Tr(C^T\pi^*)$.  Likewise, ETM-Approx + MROT-Ent has the computation complexity as $O(NM\log (1/ε_{\rm err}) +  NM/ε_p^2)$.
>
> (4) Ent-UOT has the time complexity of $\mathcal{O}(NM/ε_u)$ where  $ε_u$ denotes the error between $J_{UOT}(\pi)$ and $J_{UOT}(\pi^*)$ which is slightly faster than ETM-Approx and ETM-Approx+MROT-Ent. Meanwhile, GEMUOT has the time complexity of $O(NM\kappa\log(1/ε_u))$ where $\kappa = \max(N,M)$ with cubic order which could be time-consuming. Our proposed method is competitive among existing methods, i.e., Ent-UOT and GEMUOT.
>
> - Q1: Is it not true that Semi-UOT is a special case of UOT, since you can always set the hyperparameter coefficient of one distribution to 0? Why can’t existing UOT approaches apply to Semi-OT directly?
>
> QA1: Setting either $\tau_a$ or $\tau_b$ to 0 does not transform UOT into SemiUOT, as SemiUOT retains the marginal constraint $\pi^T \mathbf{1}_M = b$. Fundamentally, SemiUOT is a constrained optimization problem, whereas UOT is an unconstrained one. Consequently, most existing UOT methods, such as MMUOT with mirror gradient descent or GEMUOT, may be challenging to adapt for solving the SemiUOT problem.
>
> - Q2: All variants of “state of the art + UOT(some other solver that’s not yours)” are omitted.
>
> QA2: We omit the variants of “state-of-the-art+UOT” since their performance can be slightly better than state-of-the-art without UOT but significantly less than the proposed method, i.e., ETM-refine + MROT.
>
> We are willing to add more experimental results below. The experiment results on ImageCLEF for partial UDA (Table 2) are given below:
>
> | Method | I->P | P->I | I->C | C->I | C->P | P->C | Avg |
> |-|-|-|-|-|-|-|-|
> | MOT | 87.7 | 95.0 | 98.0 | 95.0 | 87.0 | 98.7 | 93.6 |
> | MOT + UOT (MMUOT) | 87.8 | 95.2 | 98.1 | 95.0 | 87.2 | 98.7 | 93.7 |
> | MOT + UOT (GEMUOT) | 87.9 | 95.4 | 98.1 | 95.2 | 87.3 | 98.8 | 93.7 |
> | MOT + UOT (Sparse Solver) | 88.0 | 95.4 | 98.2 | 95.1 | 87.3 | 98.8 | 93.8 |
> | MOT + UOT (ETM-Refine + MROT-Ent) | 88.3 | 95.6 | 98.4 | 95.3 | 87.6 | 99.0 | 94.0 |
> | MOT + UOT (ETM-Refine + MROT-Norm) | 88.7 | 95.9 | 98.7 | 95.8 | 88.0 | 99.1 | 94.4 |
> | MOT + SemiUOT(Robust-SemiSinkhorn) | 88.4 | 95.3 | 98.5 | 95.5 | 87.3 | 98.7 | 93.9 |
> | MOT + SemiUOT(ETM-Refine + MROT-Ent)| 89.1 | 96.2 | 99.2 | 96.1 | 88.5 | 99.4 | 94.8 |
> | MOT + SemiUOT(ETM-Refine + MROT-Norm)| 89.6 | 96.7 | 99.4 | 96.5 | 89.1 | 99.6 | 95.2 |
>
> The experiment results on Office-Home for universal UDA (Table 5) are given below:
>
> | Method | Ar->Cl | Ar->Pr | Ar->Rw | Cl->Ar | Cl->Rw | Cl->Rw | Pr->Ar | Pr->Cl | Pr->Rw | Rw->Ar | Rw->Cl | Rw->Pr | Avg |
> |-|-|-|-|-|-|-|-|-|-|-|-|-|-|
> | UniOT  | 67.27  | 80.54  | 86.03  | 73.51  | 77.33  | 84.28  | 75.54  | 63.33  | 85.99  | 77.77  | 65.37  | 81.92  | 76.57 |
> | UniOT + UOT (MMUOT)  | 67.45  | 80.69  | 86.32  | 73.84  | 77.86  | 84.68  | 75.91  | 64.62  | 86.13  | 77.95  | 68.06  | 82.25  | 77.15 |
> | UniOT + UOT (GEMUOT)  | 67.97  | 81.06  | 86.81  | 74.43  | 78.25  | 85.30   | 76.58  | 65.69  | 86.72  | 78.14  | 70.34  | 82.51  | 77.82 |
> | UniOT + UOT (Sparse Solver) | 67.71  | 80.95  | 86.64  | 74.22  | 78.13  | 84.89  | 76.37  | 65.11  | 86.48  | 78.06  | 69.53  | 82.44  | 77.54 |
> | UniOT + UOT (ETM-Refine + MROT-Ent) | 68.63  | 81.72  | 87.94  | 75.88  | 79.03  | 86.21  | 77.29  | 68.77  | 87.14  | 78.59  | 73.62  | 82.83  | 78.97 |
> | UniOT + UOT (ETM-Refine + MROT-Norm)| 69.02  | 81.95  | 88.36  | 76.12  | 79.36  | 86.49  | 77.03  | 69.25  | 87.30   | 78.93  | 74.18  | 82.96  | 79.25 |
>
> The experiment results on Office-31 for universal UDA (Table 6) are given below:
>
> | Method   | A->D  | A->W  | D->A  | D->W  | W->A  | W->D  | Avg |
> |-|-|-|-|-|-|-|-|
> | UniOT  | 86.97 | 88.48 | 88.35 | 98.83 | 87.60  | 96.57 | 91.13 |
> | UniOT + UOT (MMUOT)   | 87.12 | 88.79 | 88.54 | 98.95 | 88.08 | 96.62 | 91.35   |
> | UniOT + UOT (GEMUOT)  | 87.43 | 89.18 | 89.02 | 99.19 | 88.34 | 97.31 | 91.74  |
> | UniOT + UOT (Sparse Solver)  | 87.26 | 89.05 | 88.93 | 99.12 | 88.26 | 97.23 | 91.64 |
> | UniOT + UOT (ETM-Refine + MROT-Ent) | 88.25 | 89.62 | 89.47 | 99.48 | 89.10  | 97.94 | 92.31  |
> | UniOT + UOT (ETM-Refine + MROT-Norm)| 88.67 | 90.14 | 90.03 | 99.58 | 89.42 | 98.46 | 92.72 |
>
> From that, we can observe that our proposed method can still consistently achieve the best performance against some other solvers.
>
> - Q3.1: Time complexity analysis should be provided.
>
> QA3.1: Please kindly refer to A1.2.
>
> - Q3.2: How fast is your approach when $N = 1000$?
>
> QA3.2: We have done the time consumption experiments on SemiUOT/UOT in Fig.3 when $N = 1000$. The detailed experimental results are given below:
>
> | Method | Time (for UOT) | Time (for SemiUOT) |
> |-|-|-|
> | GEMUOT | 29.54s  | - |
> | ETM-Approx + MROT-Ent | 17.45s | 23.29s |
> | ETM-Refine + MROT-Ent  | 19.90s | 24.13s |
> | ETM-Refine + MROT-Norm | 22.78s | 28.62s |
>
> From the results, we can easily find that ETM+ MROT improves efficiency when the amount of points is large.
>
> - Q4: How about the results if the experiment were repeated?
>
> A4: We would like to highlight that our method is robust to repetition. We repeat for 20 experiments and report the mean and variance results below.
>
> (1) Results on Office-Home and VisDA (UDA)
>
> | Method for UDA  | Avg on Office-Home | Avg on VisDA  |
> |-|-|-|
> | JUMBOT + UOT (ETM-Refine + MROT-Ent) | 73.12 ± 0.16 | 73.59 ± 0.20 |
> | JUMBOT + UOT (ETM-Refine + MROT-Norm)| 73.43 ± 0.11   | 74.22 ± 0.15 |
>
> (2) Results on Office-Home and ImageCLEF (Partial UDA)
>
> | Method for Partial UDA  | Avg on Office-Home | Avg on ImageCLEF |
> |-|-|-|
> | MOT + UOT (ETM-Refine + MROT-Ent)  | 81.78 ± 0.22  | 94.03 ± 0.10  |
> | MOT + UOT (ETM-Refine + MROT-Norm)  | 82.11 ± 0.15 | 94.38 ± 0.08  |
> | MOT + SemiUOT (ETM-Refine + MROT-Ent) | 84.83 ± 0.13  | 94.82 ± 0.08 |
> | MOT + SemiUOT (ETM-Refine + MROT-Norm)  | 85.22 ± 0.09  | 95.19 ± 0.05 |
>
> (3) Results on Office-Home and Office-31 (Universal UDA)
>
> | Method for Universal UDA  | Avg on Office-Home | Avg on Office-31 |
> |-|-|-|
> | UniOT + UOT (ETM-Refine + MROT-Ent) | 78.95 ± 0.32  | 92.28 ± 0.27  |
> | UniOT + UOT (ETM-Refine + MROT-Norm)  | 79.22 ± 0.25  | 92.70 ± 0.22  |
>
> From that we can observe that our proposed method can still achieve stable and the best performance.
>
>
> - Q5.1: Some typos (e.g., charts/labels swap and color not consistent) here.
>
> QA5.1: We will fix them in the camera-ready version.
>
> - Q5.2: Running time versus accuracy should be discussed.
>
> A5.2: We mainly report the results on Office-Home with the UDA task.  In our experiments, we set the batch size to 512.
>
> |Method|Average Time (s) for one execution |Accuracy|
> |-|-|-|
> |JUMBOT with UOT|3.3s| 70.0  |
> |JUMBOT + GEMUOT|5.2s| 71.6 |
> |JUMBOT + ETM-Approx + MROT-Ent|3.7s| 72.3 |
> |JUMBOT + ETM-Refine + MROT-Ent |4.1s| 73.1 |
> |JUMBOT + ETM-Refine + MROT-Norm |4.6s| 73.4 |
>
> From this, we can conclude that ETM-Refine + MROT-Ent achieves the best performance with a slight increase of execution time.
>
> - Q5.3: Running time on different values of $\tau$ should be provided.
>
> A5.3: We conduct the experiments with $N=1000$ samples in synthetic data for $\tau = \{10, 100\}$ and report the computation time:
>
> | Method  | $\tau$ = 10 (UOT) | $\tau$ = 100 (UOT) | $\tau$ = 10 (SemiUOT) | $\tau$ = 100 (SemiUOT) |
> |-|-|-|-|-|
> | GEMUOT | 37.08s | 46.64s | -  | -  |
> | ETM-Approx + MROT-Ent  | 19.26s | 25.35s  | 24.04s  | 32.19s |
> | ETM-Refine + MROT-Ent  | 25.12s  | 31.75s | 28.78s  | 39.45s |
> | ETM-Refine + MROT-Norm | 32.94s  | 38.71s | 36.52s  | 47.36s |
>
> We can observe that our proposed ETM-based method can still run efficiently for larger $\tau$.
>
> - Comment: Some corrections.
>
> Response: (1) Line 34-35 should be “Meanwhile, adding additional entropy terms will lead to dense and inaccurate matching solutions.” (2) Line 58 should be “We first innovatively propose multiplier constraint terms to establish MROT for achieving more accurate results.”
>
> We hope that we addressed your concerns sufficiently, and if you agree, we would kindly request for updating the review in light of this response. If there is anything else we can answer or explain, or discuss further, kindly do let us know.
>
> Kind regards,
> Authors

---

### Note · Authors · 2025-08-12

Dear AC, Reviewers, and SAC,

Thank you for the time and expertise you invested in reviewing our paper. This note provides our final clarifications and commitments:

(1) **We appreciate the engagement and guidance of reviewers.** To start with, we must thank all the reviewers for realizing the good significance with good quality/novelty of our works. Following the rebuttal and discussion phases, the major concerns have been addressed; most reviewers acknowledged the clarifications and updated their assessments positively.

(2) **We will carefully incorporate all feedback in our carmera-ready version.** In particular, we will (i) refine exposition and notation; (ii) move key assumptions for the discrete setting (e.g., c-transform and regularity) upfront and formalize results as theorems/lemmas; (iii) highlight convergence and complexity, i.e., bounded-iteration ETM-Approx, super-linear local convergence of ETM-Refine, and MROT-Ent sharing Sinkhorn-type guarantees; (iv) bring Algorithms 1–2 into the main text and add an overview diagram; (v) correct typos and figure labeling; (vi) expand runtime-vs-accuracy and scalability analyses; (vii) integrate the additional ablations (“SOTA + alternative UOT solvers”) and repeated-run statistics introduced in the rebuttal; and so on.

(3) **We summarize the contribution is to study discrete SemiUOT and UOT through a unified algorithmic pipeline, i.e., finding marginal weights via ETM and resolving the matching by proposed MROT algorithm.**  Specifically, we propose ETM-Refine, which achieves the exact optimum efficiently and accurately by first generating a high-quality approximation via ETM-Approx and then refining it to optimality. We further introduce MROT, which leverages KKT multipliers to reweight costs for producing precise and accurate matching plans. Across synthetic data and multiple domain-adaptation settings (Office-Home, Office-31, VisDA, ImageCLEF; including partial and universal UDA and noisy/OOD settings), ETM(+MROT) consistently improves accuracy with competitive runtime and low variance over repeated runs. In noisy scenarios, the learned marginals reduce the weights of outliers, explaining the observed robustness.

We are grateful for the constructive guidance from the reviewers, AC, and SAC. We will implement the above revisions in the camera-ready version and respectfully ask the committee to consider these final remarks during decision-making.

Sincerely,
The Authors

---

### Decision · Program_Chairs · 2025-09-17

**Decision:**

Accept (poster)

**Comment:**

The paper presents a novel technique for solving unbalanced optimal transport and semi-unbalanced optimal transport.
Its main technical contribution is the Equivalent Transformation Mechanism (ETM), which converts UOT and SemiUOT problems (with KL divergence) into a classical Optimal Transport (OT) problem. This allows for the use of existing, efficient OT solvers.

Based on the  positive consensus among all reviewers, the  technical merit and novelty of the proposed  methods
and the responses to all reviewer questions, I recommend acceptance of this paper.